# Anchors Aweigh! Sail for Optimal Unified Multi-Modal Representations

## Abstract

Multimodal learning plays a crucial role in enabling machine learning models to fuse and utilize diverse data sources, such as text, images, and audio, to support a variety of downstream tasks. A unified representation across various modalities is particularly important for improving efficiency and performance. Recent binding methods, such as ImageBind (Girdhar et al., 2023), typically use a fixed anchor modality to align multimodal data in the anchor modal embedding space. In this paper, we mathematically analyze the *fixed anchor binding methods* and uncover notable limitations: (1) over-reliance on the choice of the anchor modality, (2) failure to capture intra-modal information, and (3) failure to account for inter-modal correlation among non-anchored modalities. To address these limitations, we propose CentroBind, a simple yet powerful approach that eliminates the need for a fixed anchor; instead, it employs adaptively adjustable centroid-based anchors generated from all available modalities, resulting in a balanced and rich representation space. We theoretically demonstrate that our method captures three crucial properties of multimodal learning: intra-modal learning, inter-modal learning, and multimodal alignment, while also constructing a robust unified representation across all modalities. Our experiments on both synthetic and real-world datasets demonstrate the superiority of the proposed method, showing that adaptive anchor methods outperform all fixed anchor binding methods as the former captures more nuanced multimodal interactions.

## 1 Introduction

Multimodal alignment is defined as identifying and exploiting relationships and correspondences between multiple modalities (e.g., text, image, audio) viewing common phenomena to establish meaningful connections between their representations (Baltrušaitis et al., 2018). This process allows machine learning models to analyze heterogeneous data holistically, facilitating comprehensive decision-making. A common approach is learning a shared embedding space (Tu et al., 2022; Girdhar et al., 2023; Liang et al., 2024b; Zhu et al., 2024), which aims to project data from multiple modalities into a common embedding space by clustering similar items together for direct comparison and linkage. This approach leverages well-trained single-modal embeddings, aligning them with auxiliary objective functions like contrastive loss (Oord et al., 2018) or triplet loss (Wang et al., 2020b) to minimize distances between similar items and maximize distances between dissimilar ones across modalities.

Instead of training separate models for each modality, ImageBind (Girdhar et al., 2023) pairs **images** with other modalities and projects them into a common image embedding space. Similarly, Zhu et al. (2024) shows that pairing **texts** with other modalities (LanguageBind) improves retrieval task performance when language is specified as the anchor modality. This approach has inspired various "-Bind" methods tailored to align different modalities for specific domains, such as molecular modeling (Xiao et al., 2024), medical imaging (Gao et al., 2024), brain signals (Yang et al., 2024b), and music selection for videos (Teng et al., 2024). These models commonly use image or text as the **anchor embedding** due to the abundance of data, with other modalities projected into this anchor representation.

We define the aforementioned approaches as **Fixed-Anchor-Bind** (FABIND) methods, where the embedding space of the primary anchor modality remains fixed during the alignment process. Many

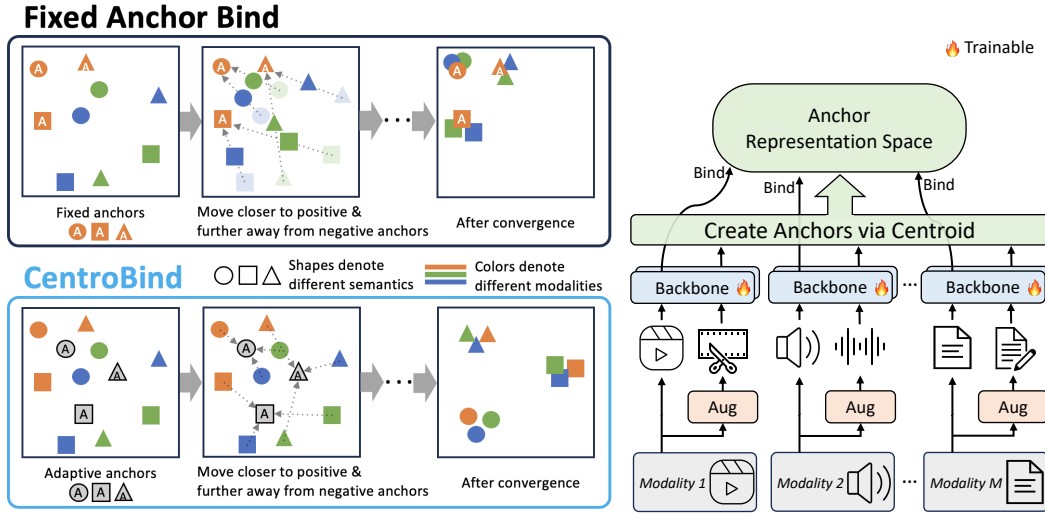

(a) FABIND vs. CENTROBIND.    (b) Graphical illustration of CENTROBIND.

Figure 1: Fixed anchor bind methods (FABIND) binds representations to the fixed anchor modality, while CENTROBIND uses adaptive anchors. (a) Colors and shapes represent different modalities and semantic information, respectively. $\mathcal{Z}$ denotes the unified representation space. (b) CENTROBIND forms adaptive anchors from the centroids of positive augmentation pairs.

"-Bind"-like approaches maximize mutual information $I(\mathbf{Z}_1; \mathbf{Z}_i)$ between the representation $\mathbf{Z}_1$ of the anchor modality and the representations $\mathbf{Z}_i, i \in \{2, \cdots, M\}$ of other modalities. Although these approaches are practically useful and widely adopted in learning unified multimodal representation, they have severe limitations, as demonstrated through theory and experiment in this paper.

**Issues with fixed anchor binding.**    First, selecting which modality to serve as an anchor is crucial but challenging, as it depends on both embedding quality and task suitability. Common choices like images or text can be suboptimal, especially when no single modality has the dominant information about the source. Second, fixing an anchor can result in loss of semantic information that is well represented only in other modalities. For instance, while *text* may describe 'a dog barks loudly,' *sound* could reveal mood, and an *image* could add facial expression, a holistic combination that a fixed alignment might miss. Third, optimizing only for anchor-to-other-modality overlooks information complementarity between non-anchored modalities. These issues are the primary motivation for the proposed CENTROBIND alternative to fixed modality anchoring methods. We formally analyze deficiencies of FABIND approaches in Section 2.

**Adaptive anchor alignment.**    We propose an alternative to fixed anchor alignment by replacing fixed anchors with *"adaptive"* anchors computed from paired samples. Our proposed method, CENTROBIND, described in Section 3, removes the need for selecting a fixed anchor modality, instead calculates the centroid over the aggregate of all modality's representations and generates an multimodal anchor representation, as shown in Figure 1a.[1] Encoders are then trained to minimize the ensemble of InfoNCE loss (Oord et al., 2018) between the representation of this adaptive anchor and the fixed modality representations. The main intuition is that *a desirable anchor should be representative of all modalities*, capturing the most comprehensive information, with well-trained encoders producing representations that naturally cluster around this shared centroid, reflecting their underlying semantic alignment.

Our theoretical analysis demonstrates that CENTROBIND effectively addresses three critical components of multimodal learning: 1) capturing intra-modal mutual information, 2) capturing inter-modal mutual information, and 3) performing multimodal alignment by maximizing embedding similarity measures. By incorporating these elements, CENTROBIND outperforms other multimodality

---

[1]While our focus is on centroid-based adaptive anchors, generalizing this approach to other methods, such as median or weighted average, is a possible extension. For further discussion, please refer to Appendix C.1.

alignment methods, as shown empirically on both synthetic and real-world datasets in retrieval and classification tasks. The proposed approach yields an unified representation space and, in the the perspective of Huh et al. (2024), who contend that multimodal representations better align as they move toward a platonic representation that captures the semantic information of all modalities simultaneously.

## 2 PROBLEM FORMULATION

In this section, we describe general representation learning and representation binding problems in multimodal learning. Then, we analyze fixed-anchor-bind (FABIND) methods such as Image-Bind (Girdhar et al., 2023), that bind multimodal representations to a user selected fixed modality.

### 2.1 REPRESENTATION LEARNING FRAMEWORK

**Notation.** Boldface upper case letters (e.g., $\mathbf{X}$) denote random vectors, and a realization is denoted by the boldface lower case letters (e.g., $\boldsymbol{x}$); For $n \in \mathbb{N}$, $[n] := \{1, 2, \cdots, n\}$; $P_{\mathbf{X}}$ and $P_{\mathbf{X},\mathbf{Y}}$ denote the marginal and the joint distributions of $\mathbf{X}$ and $(\mathbf{X}, \mathbf{Y})$, respectively.

Given $M$ datasets $\mathcal{D} = \{\mathcal{D}_i\}_{i=1}^M$, let $\mathcal{D}_i = \{(\boldsymbol{x}_{i,j}, \boldsymbol{y}_{i,j})\}_{j=1}^{N_i}$ be the dataset from the $i$-th modality, where $\boldsymbol{x}_{i,j} \in \mathcal{X}_i$ and $\boldsymbol{y}_{i,j} \in \mathcal{Y}_i$ are respectively the $j$-th input instance (e.g., feature vector) and the corresponding label in $i$-th modality, and we assume that $(\boldsymbol{x}_{i,j}, \boldsymbol{y}_{i,j}) \overset{\text{i.i.d.}}{\sim} P_{\mathbf{X}_i, \mathbf{Y}_i}$.[2] We assume that $j$ indexes paired samples among modalities. For instance, $\boldsymbol{x}_{1,c}$ and $\boldsymbol{x}_{2,c}$ are features having similar semantic information (e.g., dog image and dog sound) in $\mathcal{D}_1$ and $\mathcal{D}_2$. The goal of representation learning is to build $M$ encoders $f_i : \mathcal{X}_i \to \mathcal{Z}_i$ for each modality, which maps the input instances $\boldsymbol{x}_{i,j}$ to its embedding $\boldsymbol{z}_{i,j} = f_i(\boldsymbol{x}_{i,j})$, preserving as much information about $\boldsymbol{x}_{i,j}$ as possible.

For the uni-modal case ($M = 1$), keeping maximum information about $\boldsymbol{x}_{1,j}$ at its embedding $\boldsymbol{z}_{1,j}$ is generally preferred based on the "InfoMax" principle (Linsker, 1988), under which the objective is to maximize mutual information $I(\mathbf{X}_i; f(\mathbf{X}_i))$ between $\mathbf{X}_i$ and $f(\mathbf{X}_i)$. Throughout the paper, we call $I(\mathbf{X}_i; f(\mathbf{X}_i))$ *intra information* on $\mathbf{X}_i$. For the multimodal case ($M \geq 2$), on top of the InfoMax principle, "minimal sufficiency" is proposed in (Tian et al., 2020), which suggests maximizing *shared information* $I(f_i(\mathbf{X}_i); f_l(\mathbf{X}_l))$ between $f_i(\mathbf{X}_i)$ and $f_l(\mathbf{X}_l)$, while minimizing the *unique information* $I(\mathbf{X}_i; f_i(\mathbf{X}_i)|\{\mathbf{X}_l\}_{l \neq i})$. Although minimal sufficiency often leads to an efficient encoder with better performance in numerous multimodal downstream tasks, it is not always a good strategy as there exist exceptions where the unique information on an individual modality is crucial (Liang et al., 2024b; Wang et al., 2022). In other words, the optimality of minimal sufficiency is task-dependent. To avoid task dependency, we do not consider minimal sufficiency; instead, we maximize intra and shared information without reducing unique information. Next, we formalize the notion of sufficient embedding.

**Definition 1** ($\mathcal{Z}_i$-Sufficient embedding of $\mathbf{X}_i$ for $\mathbf{X}_l$). For an embedding space $\mathcal{Z}_i$, the embedding $f_i(\mathbf{X}_i)$ is $\mathcal{Z}_i$-sufficient for $\mathbf{X}_l$ if and only if the embedding $f_i(\mathbf{X}_i)$ achieves the maximum mutual information between $f_i(\mathbf{X}_i)$ and $\mathbf{X}_i$. Specifically,

$$f_i \in \arg \max_{f:\mathcal{X}_i \to \mathcal{Z}_i} I(f(\mathbf{X}_i); \mathbf{X}_l). \tag{1}$$

We call $f_i$ sufficient encoder of $\mathbf{X}_i$ for $\mathbf{X}_l$.

We note that if $i = l$, the sufficient encoder provides embeddings with maximum intra information, and if $i \neq l$, it gives embeddings with maximum shared information between $i$-th and $l$-th modalities.[3]

In the context of contrastive representation learning having the goal of attaining sufficient encoders in Definition 1, InfoNCE loss $I_{\text{NCE}}(X; Y)$ is often employed since it relates to mutual information. Specifically, InfoNCE provides a lower bound on mutual information, i.e., $I(\mathbf{X}; \mathbf{Y}) \geq -I_{\text{NCE}}(\mathbf{X}; \mathbf{Y})$ (Oord et al., 2018), and thus minimizing InfoNCE leads to an increase in

---

[2]In self-supervised learning, labels might not exist, which corresponds to the case that $\boldsymbol{y}_{i,j}$ are empty.

[3]With a proper choice of $\mathcal{Z}_i$ ensuring $\max_{f:\mathcal{X}_i \to \mathcal{Z}_i} I(f(\mathbf{X}_i); \mathbf{X}_l) = I(\mathbf{X}_i; \mathbf{X}_l)$, Definition 1 says that $\boldsymbol{z}_{i,j} = f_i(\boldsymbol{x}_{i,j})$ is a sufficient statistic (Polyanskiy & Wu, 2024) of $\boldsymbol{x}_{i,j}$ for $\boldsymbol{x}_{l,j}$ as the encoding entails no information loss.

mutual information. InfoNCE loss between embeddings $\mathbf{U}$ and $\mathbf{V}$ can be written as follows:

$$I_{\text{NCE}}(\mathbf{U}; \mathbf{V}|\tau) = \mathbb{E}_{P_{\mathbf{U},\mathbf{V}}, \prod_{i=1}^{N} P_{\mathbf{V}_i}} \left[ -\log \frac{\exp(\mathbf{U}^\top \mathbf{V}/\tau)}{\exp(\mathbf{U}^\top \mathbf{V}/\tau) + \sum_{i=1}^{N} \exp(\mathbf{U}^\top \mathbf{V}_i/\tau)} \right], \quad (2)$$

where the expectation is taken with respect to the distribution $P_{\mathbf{U},\mathbf{V}} \prod_{i=1}^{N} P_{\mathbf{V}_i}$. Here, we say $(\mathbf{U}, \mathbf{V})$ is a positive pair if $(\mathbf{U}, \mathbf{V}) \sim P_{\mathbf{U},\mathbf{V}}$ and $(\mathbf{U}, \mathbf{V})$ is a negative pair if $(\mathbf{U}, \mathbf{V}_i) \sim P_{\mathbf{U}} P_{\mathbf{V}_i}$. In (2), $N \geq 1$ and $\tau > 0$ are hyper-parameters, specifying the number of negative samples and the temperature parameter. For simplicity, in this paper we assume that embeddings are normalized (Wang & Isola, 2020) to unit vectors and are of the same dimensionality. Then, the exponent $\mathbf{U}^\top \mathbf{V}/\tau$ in (2) is proportional the cosine similarity score between $\mathbf{U}$ and $\mathbf{V}$.

## 2.2 BINDING REPRESENTATION SPACES

In addition to the objective of capturing intra and shared information, multimodal learning often takes into account multimodal alignment (Radford et al., 2021; Duan et al., 2022). Without multimodal alignment, each modality can only access its own embedding structure depending on its encoder. For example, embeddings of cat and dog images, respectively, locate around $(1, 0)$ and $(0, 2)$ in $\mathbb{R}^2$, whereas embeddings of cat and dog text can lie around $(0, 2)$ and $(1, 0)$. Such a misalignment can happen even for sufficient encoders (Definition 1), since the mutual information is invariant to one-to-one mappings (Polyanskiy & Wu, 2024).

To align multimodal embedding spaces, a unified representation space (Radford et al., 2021; Zhou et al., 2023) or multimodal alignment (Wang et al., 2023; Liang et al., 2024c) have been proposed for multimodal representation learning, in which embeddings of multimodal features having similar semantic should near each other in embedding space. Several FABIND methods have been proposed (see Appendix A for a summary of FABIND methods) that include ImageBind (Girdhar et al., 2023). ImageBind sets the image modality as the fixed anchor modality, and then InfoNCE loss is minimized between the embeddings of the anchor modality and the other modalities. FABIND (e.g., ImageBind) aims to find encoders $f_i^{\text{FB}}$ for all modalities, except the anchor modality, such that

$$f_i^{\text{FB}} = \arg\min_{f_i: \mathcal{X}_i \to \mathcal{Z}_i} I_{\text{NCE}}(f_1(\mathbf{X}_1); f_i(\mathbf{X}_i)), \ \forall i \in \{2, \cdots, M\}, \quad (3)$$

where $f_1$ is the encoder for the anchor modality (an image encoder in ImageBind). Note that FABIND freezes $f_1$, initialized by an existing pretrained model, during the optimization.

## 2.3 ANALYSIS OF FABIND

In this section, we characterize theoretical limitations of FABIND. To this end, we rewrite (3) as

$$f_i^{\text{FB}} = \arg\max_{f_i: \mathcal{X}_i \to \mathcal{Z}_i} I(f_1(\mathbf{X}_1); f_i(\mathbf{X}_i)), \ \forall i \in \{2, \cdots, M\}, \quad (4)$$

reflecting the fact that minimizing InfoNCE loss is equivalent to maximizing mutual information.[4] Let FABIND encoders from (4) for each modality be defined as $\mathcal{F}^{\text{FB}} = \{f_1, f_2^{\text{FB}}, \cdots, f_M^{\text{FB}}\}$. The anchor encoder $f_1$ is fixed during the entire FABIND procedure. Moreover, we assume that $I(f_1(\mathbf{X}_1); f_i^{\text{FB}}(\mathbf{X}_i)) = I(f_1(\mathbf{X}_1); \mathbf{X}_i)$ is the maximum value that can be achieved by (4) due to data processing inequality (Polyanskiy & Wu, 2024). We next demonstrate that the quality of anchor embedding $f_1(\mathbf{X}_1)$ significantly impacts the performance of $\mathcal{F}^{\text{FB}}$ in terms of shared information. The following propositions show the dependency of FABIND on anchor embedding quality.

**Proposition 1** (FABIND with sufficient anchor). *Let $f_1^{\text{suf}}(\mathbf{X}_1)$ be a sufficient embedding of the anchor $\mathbf{X}_1$, and let $\mathbf{X}_i, i \in [M]$, be a discrete random variable. Assume that $f_i^{\text{FB}}, i \in \{2, \cdots, M\}$ are obtained by (4) with a sufficient anchor encoder $f_1 = f_1^{\text{suf}}$, i.e., $I(f_1^{\text{suf}}(\mathbf{X}_1); f_i^{\text{FB}}(\mathbf{X}_i)) = I(f_1^{\text{suf}}(\mathbf{X}_1); \mathbf{X}_i)$. Then,*

$$I(f_1^{\text{suf}}(\mathbf{X}_1); f_i^{\text{FB}}(\mathbf{X}_i)) = I(\mathbf{X}_1; \mathbf{X}_i), \ \forall i \in \{2, \cdots, M\}. \quad (5)$$

---

[4]In contrast to (3), $f_i^{\text{FB}}$ in (4) might not be aligned with other modalities due to the one-to-one mapping invariant property of mutual information. However, we here do not analyze multimodal alignment of FABIND from (4), but rather investigate the performance of encoders in terms of the sufficiency in Definition 1.

*Proof.* The proof is in Appendix B.1. □

**Proposition 2** (FABIND with insufficient anchor). *Let $f_1^{\text{ins}}(\mathbf{X}_1)$ be an insufficient embedding of the anchor $\mathbf{X}_1$ for $\mathbf{X}_1$, in the sense that there exists some $\epsilon > 0$ such that $I(f_1^{\text{ins}}(\mathbf{X}_1); \mathbf{X}_1) < \epsilon \leq \max_f I(f(\mathbf{X}_1); \mathbf{X}_1)$. Assume that $f_i^{\text{FB}}, i \in \{2, \cdots, M\}$ are obtained by (4) with $f_1 = f_1^{\text{ins}}$, i.e., $I(f_1^{\text{ins}}(\mathbf{X}_1); f_i^{\text{FB}}(\mathbf{X}_i)) = I(f_1^{\text{ins}}(\mathbf{X}_1); \mathbf{X}_i)$. Then,*

$$I(f_1^{\text{ins}}(\mathbf{X}_1); f_i^{\text{FB}}(\mathbf{X}_i)) < \epsilon, \; \forall i \in \{2, \cdots, M\}. \tag{6}$$

*Proof.* The proof is in Appendix B.2. □

Proposition 1 shows that the FABIND encoders $\mathcal{F}^{\text{FB}}$ learned with a sufficient anchor embedding can achieve the maximum shared information between the anchor and the other modalities. However, it does not guarantee shared information between *non-anchored* modalities $I(f_i(\mathbf{X}_i); f_l(\mathbf{X}_l))$, $i, l \neq 1$, which can also be seen from (4). Proposition 2 establishes that an insufficient anchor may lead to a reduction of shared information between the anchor and the other modalities, implying that the performance of FABIND may overly depend on the quality of the arbitrarily selected anchor.

The above Propositions reveals several limitations in FABIND. Firstly, achieving maximum shared information requires sufficient anchor representation, which depends on having both an informative modality and a sufficient encoder. Without these conditions, FABIND may not effectively capture shared information. Secondly, even with sufficient anchor representation, FABIND may not provide encoders with maximum intra information. This is because its objective function (4) does not take into account intra information. Thirdly, the objective function of FABIND (4) focuses solely on learning shared information between pairs of anchor and non-anchored modalities, while disregarding shared information among non-anchor modalities. This implies that FABIND may not capture shared information among non-anchored modalities. This limitation renders FABIND less effective when crucial shared information exists among non-anchored modalities. Lastly, the representation produced by FABIND may not approximate an ideal representation, such as the "Platonic representation" of (Huh et al., 2024). While integrating all modalities is needed in order to effectively represent the information in all modalities, FABIND falls short of this ideal. In the sequel, we introduce an alternative multimodal representation that, fully leverages fine-grained, sample-level information in all modalities.

In summary, FABIND exhibits the following weaknesses:

**P1:** over-reliance on a single anchor modality;

**P2:** failure to capture intra information;

**P3:** absence of shared information among non-anchored modalities;

Next, we propose our method CENTROBIND, which does not require selecting a single anchor modality and is thus capable of capturing intra and shared information.

## 3 TOWARD A DESIRABLE UNIFIED REPRESENTATION SPACE

The main intuition deriving CENTROBIND is as follows: 1) A desirable multimodal embedding should not favor any specific modality; 2) A desirable unified embedding should attain the highest alignment in similarity. To this end, we generate an anchor representation that is the centroid of the multiple modality representations. Then, we train the encoders toward minimizing the InfoNCE loss between anchor and other modalities, similarly to FABIND. We note that other dynamic anchors, such as median and weighted average (refer to Appendix C.1 for further discussion), are also possible alternatives to the centroid. However, we focus on CENTROBIND, as the centroid represents the geometric center, aligning with the objective in achieving multimodal alignment in the embedding space (e.g., $\mathbb{R}^{d_z}$). Next, we formally define CENTROBIND, and show that the method aligns multimodal representations and simultaneously maximizes intra and shared information.

---

**Algorithm 1** CENTROBIND

---

1: Initialize encoders $f_1^{(0)}, f_2^{(0)}, \cdots, f_M^{(0)}$.
2: **for** $t = 0, 1, \ldots, t_{\max}$ **do**
3:      Sample a batch dataset $B$ from multimodal datasets $\{\mathcal{D}_i\}_i$.
4:      Generate anchor embeddings $\{\boldsymbol{a}_j\}_{j \in \mathcal{I}_B}$ using $\boldsymbol{a}_j = \text{mean}(\{f_i^{(t)}(\boldsymbol{x}'_{i,j})\}_i)$ in (7)
5:      **for** $i = 1, \ldots, M$ **do**
6:          Optimize $f_i^{(t+1)}$ toward minimizing $\mathcal{L}_{\text{CB}}(f_i^{(t+1)}|\tau)$ in (8)
7:      **end for**
8: **end for**

---

### 3.1 CENTROBIND

Consider $M$ modalities with corresponding encoders $\{f_i\}_{i=1}^M$. The CENTROBIND algorithm is presented in Algorithm 1, and a graphical illustration is given in Figure 1b. In the following, we describe each step of the algorithm.

**Initial encoders.** We initialize $M$ encoders $f_i : \mathcal{X}_i \to \mathcal{Z}$, $\forall i \in [M]$ for the $M$ modalities. These encoders can either be pretrained models (i.e., backbones) or parameterized models with random weights. The primary constraint for these initial encoders is that their output space must be the modality-independent embedding space $\mathcal{Z}$. When using pretrained encoders that produce embeddings in different output spaces, these are projected onto the common space $\mathcal{Z}$, ensuring consistency of output space across modalities.

**Anchor embedding.** Recall that $\boldsymbol{x}_{i,j} \in \mathcal{D}_i$ denotes the $j$-th feature in the $i$-th modality, where $j$ indexes positive pairs of features (e.g., different views of the same object). In each training iteration of CENTROBIND, we need to compute an anchor embedding $\boldsymbol{a}_j$ for the $j$-th multimodal positive features $\{\boldsymbol{x}_{i,j}\}_{i=1}^M$. This anchor $\boldsymbol{a}_j$ serves as a desirable aligned embedding for these features. The anchor $\boldsymbol{a}_j$ is calculated as follows:

$$\boldsymbol{a}_j = \text{mean}\left(\{f_i(\boldsymbol{x}'_{i,j})\}_i\right), \tag{7}$$

where $\text{mean}(\cdot)$ denotes the mean operator that computes the average of its input, and $\boldsymbol{x}'_{i,j}$ represents an augmented version of $\boldsymbol{x}_{i,j}$. If $\{\boldsymbol{x}_{i,j}\}_{i=1}^M$ are available in multimodal datasets, the anchor is given by $\boldsymbol{a}_j = \frac{1}{M} \sum_{i=1}^M f_i(\boldsymbol{x}'_{i,j})$. If only $m < M$ positive pairs are present among $M$ modalities, the anchor is given by $\boldsymbol{a}_j = \frac{1}{m} \sum_{i \in \mathcal{I}_j} f_i(\boldsymbol{x}'_{i,j})$, where $\mathcal{I}_j$ is the set of indices of modalities having the $m$ available features.

**Binding encoders to the anchor.** Once anchor embeddings $\{\boldsymbol{a}_j\}_j$ are derived from a batch of data $B = \{\boldsymbol{x}_{i,j}\}_{i,j}$, CENTROBIND aligns each modality-specific encoder embedding with the anchor embedding by minimizing the InfoNCE loss. Specifically, let $\mathbf{A} = \text{mean}(\{f_i(\mathbf{X}_i)\}_i)$ represent the anchor embedding variable. Then, CENTROBIND aims to minimize the InfoNCE loss $I_{\text{NCE}}(\mathbf{A}; f_i(\mathbf{X}_i))$ across all modalities $i \in [M]$. A detailed expression for this loss is provided in (8).

CENTROBIND optimizes the following symmetrized loss function:

$$\mathcal{L}_{\text{CB}}(f_i|\tau) = I_{\text{NCE}}(\mathbf{A}; f_i(\mathbf{X}_i)|\tau) + I_{\text{NCE}}(f_i(\mathbf{X}_i); \mathbf{A}|\tau), \tag{8}$$

where $\mathcal{L}_{\text{CB}}(f_i|\tau)$ denotes the loss function for the $i$-th modality. In particular, with a batch data $B = \{\boldsymbol{x}_{i,j} : i \in [M], j \in \mathcal{I}_B\}$, the loss can be computed as

$$I_{\text{NCE}}(\mathbf{A}; f_i(\mathbf{X}_i)|\tau) = -\frac{1}{|\mathcal{I}_B|} \sum_{k=1}^{|\mathcal{I}_B|} \log \frac{\exp(\boldsymbol{a}_k^\top f_i(\boldsymbol{x}_{i,k})/\tau)}{\sum_{j \in \mathcal{I}_B} \exp(\boldsymbol{a}_k^\top f_i(\boldsymbol{x}_{i,j})/\tau)} \quad \text{and} \tag{9a}$$

$$I_{\text{NCE}}(f_i(\mathbf{X}_i); \mathbf{A}|\tau) = -\frac{1}{|\mathcal{I}_B|} \sum_{k=1}^{|\mathcal{I}_B|} \log \frac{\exp(\boldsymbol{a}_k^\top f_i(\boldsymbol{x}_{i,k})/\tau)}{\sum_{j \in \mathcal{I}_B} \exp(f_i^\top(\boldsymbol{x}_{i,k})\boldsymbol{a}_j/\tau)}. \tag{9b}$$

## 3.2 THEORETICAL ANALYSIS OF CENTROBIND

We start by providing a lower bound on the objective function of CENTROBIND $\mathcal{L}_{\text{CB}}(f_i|\tau)$ (8) in Theorem 1, followed by an analysis of the minimizer of $\mathcal{L}_{\text{CB}}(f_i|\tau)$.

**Theorem 1.** *Consider $B = \{\boldsymbol{x}_{i,j} : i \in [M], j \in \mathcal{I}_B\}$ with a set of indices $\mathcal{I}_B$, where $\boldsymbol{x}_{i,j}$ is the $j$-th sample of $i$-th modality. Then, for any encoders $\{f_i\}_i$ and for any $\tau > 0$, (9a) is bounded as*

$$I_{\text{NCE}}\left(\mathbf{A}; f_i(\mathbf{X}_i) \mid \tau\right) \geq \frac{1}{|\mathcal{I}_B|} \sum_{l=1}^{M} I_{\text{NCE}}\left(f_l(\mathbf{X}'_l); f_i(\mathbf{X}_i) \,\Big|\, \frac{\tau M}{|\mathcal{I}_B|}\right) - \frac{1}{|\mathcal{I}_B|} \sum_{k=1}^{|\mathcal{I}_B|} \log C_{\mathcal{F},k,i}, \quad (10)$$

*where $C_{\mathcal{F},k,i} = \frac{(c_{\mathcal{F},k,i}^{\min} + c_{\mathcal{F},k,i}^{\max})^2}{4 c_{\mathcal{F},k,i}^{\min} c_{\mathcal{F},k,i}^{\max}}$ with $g(l,j|k,i) := \exp\left(\frac{|\mathcal{I}_B| f_l^{\top}(\boldsymbol{x}'_{l,k}) f_i(\boldsymbol{x}_{i,j})}{\tau M}\right)$,*

$$c_{\mathcal{F},k,i}^{\min} = \min_{l \in [M], j \in \mathcal{I}_B} g(l,j|k,i), \quad \text{and} \quad c_{\mathcal{F},k,i}^{\max} = \max_{l \in [M], j \in \mathcal{I}_B} g(l,j|k,i). \quad (11)$$

*Proof.* The proof is in Appendix B.3. $\qquad\square$

Theorem 1 provides a lower bound of $I_{\text{NCE}}\left(\mathbf{A}; f_i(\mathbf{X}_i) \mid \tau\right)$ in (9a), which is a part of the CENTROBIND objective function $\mathcal{L}_{\text{CB}}(f_i|\tau)$. Thus CENTROBIND minimizes a lower bound (10) that consists of two terms, $\sum_{l=1}^{M} I_{\text{NCE}}\left(f_l(\mathbf{X}'_l); f_i(\mathbf{X}_i) \,\Big|\, \frac{\tau M}{|\mathcal{I}_B|}\right)$ and $-\sum_{k=1}^{|\mathcal{I}_B|} \log C_{\mathcal{F},k,i}$. We next provide intuition on why a minimization of the lower bound is justified.

**The effect of minimizing $\sum_{l=1}^{M} I_{\text{NCE}}\left(f_l(\mathbf{X}'_l); f_i(\mathbf{X}_i) \,\Big|\, \frac{\tau M}{|\mathcal{I}_B|}\right)$.** The objective of minimizing $\sum_{l=1}^{M} I_{\text{NCE}}(f_l(\mathbf{X}'_l); f_i(\mathbf{X}_i) \mid \frac{\tau M}{|\mathcal{I}_B|})$ is to reduce several InfoNCE losses. Here, each term in the sum represents the InfoNCE loss between embeddings $f_l(\mathbf{X}'_l)$ from modality $l$ and $f_i(\mathbf{X}_i)$ from modality $i$, with $\frac{\tau M}{|\mathcal{I}_B|}$ being a temperature parameter for scaling the loss. This summation can be divided into two components: 1) Intra Information: When $l = i$, the term measures the similarity between embeddings within the same modality. Minimizing this loss enhances the representation of modality $i$, improving intra information; 2) Shared Information: When $l \neq i$, the term measures the similarity between embeddings from different modalities. Minimizing these losses helps in learning shared information between modalities, contributing to a more representative multimodal embedding.

By optimizing this summation, CENTROBIND effectively captures both intra and shared information. As shown below, this generally results in a more balanced representation for the modalities. In contrast, as noted in Section 2.3, FABIND does not adequately capture intra information and shared information between non-anchored modalities. This limitation highlights the advantage of CENTROBIND in achieving a more integrated multimodal representation than fixed anchor binding methods.

**The effect of minimizing $-\sum_{k=1}^{|\mathcal{I}_B|} \log C_{\mathcal{F},k,i}$.** We show the effect of growing $C_{\mathcal{F},k,i}$ in terms of cosine similarity score between embeddings. Since $C_{\mathcal{F},k,i} = \frac{1}{4}\left(\sqrt{\gamma} + \sqrt{\frac{1}{\gamma}}\right)^2$ with $\gamma = \frac{c_{\mathcal{F},k,i}^{\max}}{c_{\mathcal{F},k,i}^{\min}} \geq 1$, maximizing $C_{\mathcal{F},k,i}$ is equivalent to simultaneously maximizing $c_{\mathcal{F},k,i}^{\max}$ and minimizing $c_{\mathcal{F},k,i}^{\min}$. For ease of the analysis, we assume that the encoders are reasonably well-trained. Then, since a positive pair of embeddings normally yields higher similarity score, $c_{\mathcal{F},k,i}^{\max}$ is attained by choosing $l = i$ and $j = k$ in (11) as such choices make $\boldsymbol{x}'_{l,k}$ be positive pair with $\boldsymbol{x}_{i,j}$. Thus, $c_{\mathcal{F},k,i}^{\max}$ is roughly proportional to the similarity score of a positive pair of embeddings. Conversely, $c_{\mathcal{F},k,i}^{\min}$ corresponds to the similarity scores of negative pairs, which tend to be low. Hence, minimizing $-\sum_{k=1}^{|\mathcal{I}_B|} \log C_{\mathcal{F},k,i}$ enhances the similarity scores for positive pairs and reduces those for negative pairs, improving the overall multimodal alignment.

These comments suggest that CENTROBIND addresses the limitations **P1**, **P2**, and **P3** of FABIND identified in Section 2.3. We argue that the unified representation of CENTROBIND is closer to an ideal platonic representation (Huh et al., 2024) as compared to the representation used by FABIND. A platonic representation is defined as an ideal representation of the aggregate set of all modalities that maximally captures all multimodality information. From this perspective, a representation

derived solely from a single modality, without leveraging others, is not ideal. This suggests that CENTROBIND's unified space is likely to retain a more comprehensive representation of all modalities.

# 4 EXPERIMENT

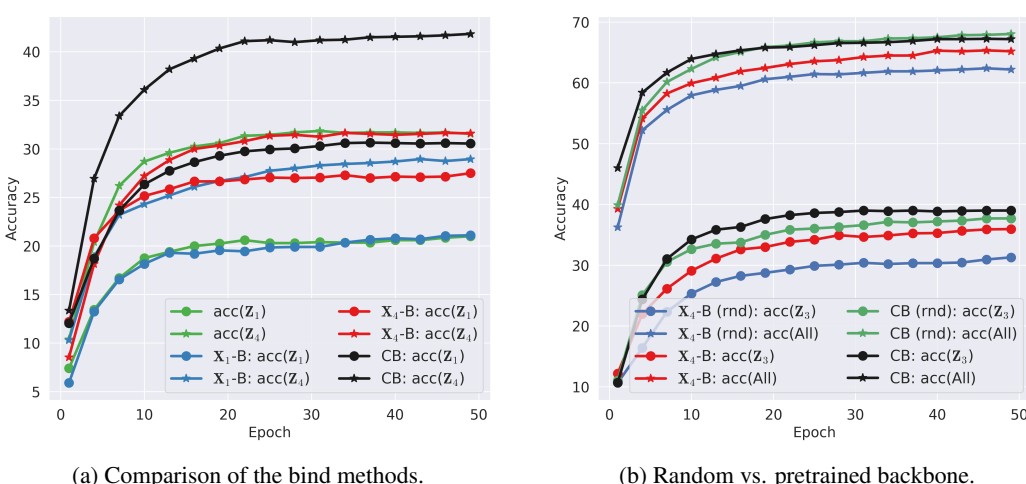

(a) Comparison of the bind methods.

(b) Random vs. pretrained backbone.

Figure 2: Accuracy as a measure of the representation space quality. Abbreviation: $\mathbf{X}_i$-B or CB: applying FABIND with anchor $\mathbf{X}_i$ or applying CentroBind; acc($\mathbf{Z}_i$) or acc(All): accuracy of $\mathbf{Z}_i$ or of concatenated embeddings $(\mathbf{Z}_1, \cdots, \mathbf{Z}_M)$; (rnd): if random backbones are used.

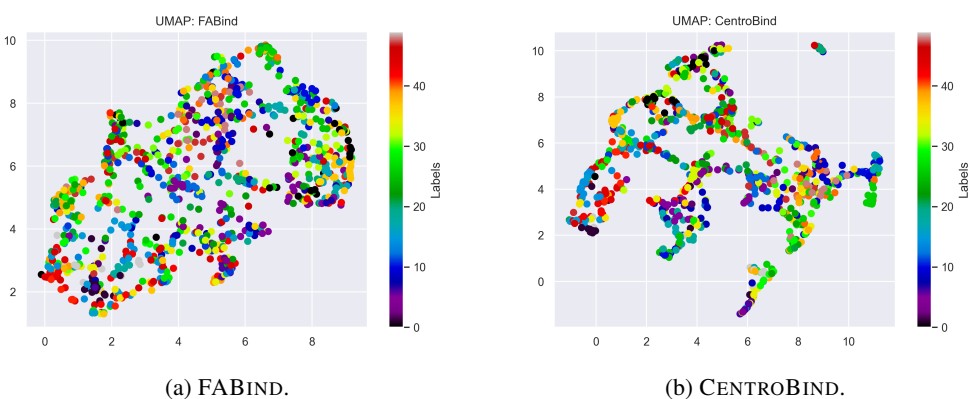

(a) FABIND.

(b) CENTROBIND.

Figure 3: Representation visualization via UMAP.

## 4.1 EXPERIMENTS WITH SYNTHETIC DATASET

**Synthetic datasets.** We employ a latent variable model (Bishop & Nasrabadi, 2006) for generating synthetic multimodal datasets. A latent variable model is a statistical model for data $\mathbf{X} \in \mathbb{R}^{d_x}$, under which $\mathbf{X}$ is generated according to a conditional probability distribution $P_{\mathbf{X}|\mathbf{Z}}$, where $\mathbf{Z} \in \mathbb{R}^{d_z}$ is the latent variable. In terms of the representation learning framework, $\mathbf{Z}$ can be seen as a low dimensional representation of $\mathbf{X}$. We assume that the class label $\mathbf{Y} \in [K]$ and the latent variable $\mathbf{Z}$ are jointly distributed according to $P_{\mathbf{Z},\mathbf{Y}}$. In our setting, we exploit Gaussian mixture model (GMM) (Bishop & Nasrabadi, 2006) for the latent variable $\mathbf{Z}$, and we generate $M$ modalities $\mathbf{X}_i = g_i(\mathbf{Z}) + \mathbf{N}, i \in [M]$ with random noise $\mathbf{N}$ and some non-linear projections $g_i : \mathbb{R}^{d_z} \to \mathbb{R}^{d_x}$. We choose the projections in a way such that each model can be ranked in ascending order, i.e., $\mathbf{X}_1$ is the worst, and $\mathbf{X}_4$ is the best modality in terms of their inherent correlation with the latent variable. The class label $\mathbf{Y}$ is set to the component id of GMM (for details, see Appendix C.1).

Table 1: Zero-shot one-to-one and two-to-one retrieval accuracy. ($\mathcal{V}$: video, $\mathcal{A}$: audio, $\mathcal{T}$: text)

| | One-to-One | | | | | Two-to-One | | | |
| Method | Retrieval | Top-1 | Top-5 | Top-10 | Method | Retrieval | Top-1 | Top-5 | Top-10 |
|---|---|---|---|---|---|---|---|---|---|
| FABIND | $\mathcal{V} \to \mathcal{T}$ | 0.446 | 0.719 | 0.822 | FABind | $\mathcal{V}, \mathcal{A} \to \mathcal{T}$ | 0.309 | 0.665 | 0.781 |
| CENTROBIND | | **0.483** | **0.764** | **0.850** | | | | | |
| FABIND | $\mathcal{A} \to \mathcal{T}$ | 0.077 | 0.238 | 0.367 | CENTROBIND | | **0.745** | **0.957** | **0.978** |
| CENTROBIND | | **0.233** | **0.517** | **0.678** | | | | | |
| FABIND | $\mathcal{T} \to \mathcal{V}$ | **0.812** | **0.946** | **0.978** | FABIND | $\mathcal{T}, \mathcal{A} \to \mathcal{V}$ | 0.180 | 0.401 | 0.513 |
| CENTROBIND | | 0.591 | 0.839 | 0.909 | | | | | |
| FABIND | $\mathcal{A} \to \mathcal{V}$ | **0.058** | 0.154 | 0.226 | CENTROBIND | | **0.388** | **0.646** | **0.768** |
| CENTROBIND | | 0.052 | **0.184** | **0.284** | | | | | |
| FABIND | $\mathcal{T} \to \mathcal{A}$ | 0.201 | 0.438 | 0.584 | FABIND | $\mathcal{T}, \mathcal{V} \to \mathcal{A}$ | 0.099 | 0.257 | 0.364 |
| CENTROBIND | | **0.290** | **0.572** | **0.706** | | | | | |
| FABIND | $\mathcal{V} \to \mathcal{A}$ | 0.051 | 0.155 | 0.223 | CENTROBIND | | **0.232** | **0.490** | **0.625** |
| CENTROBIND | | **0.054** | **0.175** | **0.258** | | | | | |

**Experiment results.** Figure 2 shows the classification accuracies with a synthetic dataset of $M = 4$ modalities. To obtain the results in Figure 2a, we initialize pretrained backbones for each modality, apply FABIND ($\mathbf{X}_i$-B) with anchor $\mathbf{X}_i$ or CENTROBIND (CB), and evaluate accuracy ($\mathrm{acc}(\mathbf{Z}_i)$) with embeddings from $i$-th modality. We provide $\mathrm{acc}(\mathbf{Z}_i)$, without any binding, for a reference. Figure 2a verifies our analysis of FABIND (Section 2.3) and CentroBind (Section 3.2): (1) the comparison between $\mathbf{X}_1$-B and $\mathbf{X}_4$-B shows the importance of choosing an anchor modality; (2) the comparison between $\mathrm{acc}(\mathbf{Z}_4)$ and $\mathbf{X}_1$-B: $\mathrm{acc}(\mathbf{Z}_4)$ shows a performance deterioration by FABIND, demonstrating the impact on performance of FABIND failure to capture intra information; (3) the proposed CB consistently outperforms FABIND, indicating that CB successfully captures elements that FABIND overlooks, including intra information and shared information among non-anchored modalities.

Figure 2b includes accuracies of FABIND and CB with random backbones. Similarly, CB outperforms all baselines. Somewhat surprisingly, CB with random backbones (green curves) also performs better than FABIND with pretrained backbones (red curves). This further supports our analysis that CENTROBIND is robust to backbone quality as it optimizes intra and shared information, whereas FABIND is sensitive to the backbone quality. Overall, these empirical results validate our findings. For clarity, we summarize the final accuracies in Table 3. We provide additional experimental results on synthetic datasets with $M = 6, 8$ in Appendix C.1. With the larger number of modalities, CB still outperforms the baselines, strengthening CB regarding scalability.

In addition, we visualize the embeddings learned by FABIND and CENTROBIND using UMAP (McInnes et al., 2018) in Figure 3 (for more details and additional visualizations using t-SNE (Van der Maaten & Hinton, 2008), see Appendix C.1). Figure 3 shows that CENTROBIND embeddings are better clustered, whereas FABIND embeddings appear more scattered, implying that CENTROBIND achieves a superior embedding structure compared to FABIND.

In terms of convergence, we empirically examine the convergence speed of both CENTROBIND and FABIND. In Figure 6, we plot the training loss curves, which show similar behavior, suggesting that the adaptive anchor does not introduce issues related to convergence or stability. Detailed discussions can be found in Appendix C.1.

### 4.2 EXPERIMENTS WITH REAL-WORLD DATASET

In this section, we provide experiment results with a real-world dataset. We compare CENTROBIND, FABIND anchored at text modality, UniBind (Lyu et al., 2024), AudioCLIP (Guzhov et al., 2022), and ViT-Lens (Lei et al., 2024) (see implementation detail in Appendix C.2). We utilize the MUStARD dataset (Castro et al., 2019) for its rich combination of multimodal data with more than two modalities. It consists of 690 video clips (including audio) and text for sarcasm detection with labels such as sarcasm indicators and speaker names. For the backbones in FABIND and CENTROBIND, we use the pretrained VideoMAE model (Tong et al., 2022) for video data, the pretrained WaveLM model (Chen et al., 2022) for audio data, and the pretrained BERT model (Devlin et al., 2019) for text data. A detailed description of the training setting is provided in Appendix C.2.

**Downstream tasks.** We perform evaluations in zero-shot binary and multi-class classification tasks, One-to-One cross-modal retrieval, and Two-to-One cross-modal retrieval. For classification tasks, we use a Multi-Layer Perceptron (MLP) to perform sarcasm detection as a binary classification and

speaker classification with 23 multi-class categories. In particular, MLP is trained on embeddings in a single modality (denoted by **Tr** in Table 2) and accuracy is evaluated on another modality (denoted by **Ev** in Table 2). In retrieval tasks, we measure the accuracy of correct retrieval. For One-to-One case, we retrieve data sample in different modality by choosing the closest embedding from a single input embedding, while for Two-to-One case we choose the closest embedding from the centroid of two input embeddings in two modalities. We denote input and target modalities with $\rightarrow$ in Table 1.

**Results on cross-modal retrieval.** Table 1 shows the performance for one-to-one and two-to-one retrieval tasks. CENTROBIND consistently excels in one-to-one retrieval for text and audio modalities, while FABIND performs better for video retrieval. This might be due to a power of text to describe, which may be suitable for FABIND anchored at text modality. A notable observation is that the centroid of video and audio embeddings achieves the best text retrieval performance. This implies complementary information exists and is captured by CENTROBIND.

**Results on sarcasm & speaker classification.**
Table 2 presents results for sarcasm detection and speaker classification tasks, where Sar-1 indicates Top-1 accuracy for sarcasm, and Spk-$k$, $k = 1, 3, 5$ represent Top-k accuracies for speaker classification. It is important to highlight that CENTROBIND and FABIND are trained on a single modality (**Tr**) and evaluated on a different modality (**Ev**) in a zero-shot setting, which can effectively measure ability of multmimodal alignment. In this experiment, CENTROBIND consistently outperforms FABIND and UniBind across all pairs of train and evaluation modalities, which can be distributed to CENTROBIND generally learning a better unified embedding space than FABIND. UniBind performs poorly in the zero-shot cross-modal experiment, which we believe is due to its insufficient multimodal alignment. Since UniBind utilizes LLM-augmented descriptions for each modality and binds other encoders to these descriptions, multimodal alignment may fail if the descriptions are dispersed across the embedding space. As analyzed in Section 2.3 and Section 3.2, these results highlight the CENTROBIND's ability to preserve intra and shared information among modalities, which are useful in unknown downstream tasks. Moreover, the zero-shot setting verifies the multimodal alignment of CENTROBIND.

Table 2: Accuracy results for Sarcasms and Speakers. ($\mathcal{V}$: video, $\mathcal{A}$: audio, $\mathcal{T}$: text). Asterisks* denote accuracy evaluated in different settings.

| Method | Tr, (Ev) | Sar-1 | Spk-1 | Spk-3 | Spk-5 |
|---|---|---|---|---|---|
| FABIND | | 0.706 | 0.378 | 0.614 | 0.730 |
| UniBind | | 0.544 | 0.170 | 0.328 | 0.478 |
| AudioCLIP* | $\mathcal{V}, (\mathcal{T})$ | 0.501 | 0.096 | 0.258 | 0.388 |
| ViT-Lens* | | 0.506 | 0.097 | 0.343 | 0.449 |
| CENTROBIND | | **0.716** | **0.474** | **0.736** | **0.836** |
| FABIND | | 0.648 | 0.186 | 0.455 | 0.577 |
| UniBind | | 0.628 | 0.220 | 0.399 | 0.501 |
| AudioCLIP* | $\mathcal{A}, (\mathcal{T})$ | 0.486 | 0.094 | 0.214 | 0.322 |
| ViT-Lens* | | 0.484 | 0.077 | 0.214 | 0.313 |
| CENTROBIND | | **0.691** | **0.290** | **0.546** | **0.714** |
| FABIND | | 0.572 | 0.243 | 0.445 | 0.630 |
| UniBind | | 0.484 | 0.129 | 0.262 | 0.404 |
| AudioCLIP* | $\mathcal{T}, (\mathcal{V})$ | 0.506 | 0.158 | 0.345 | 0.461 |
| ViT-Lens* | | 0.502 | 0.168 | 0.323 | 0.423 |
| CENTROBIND | | **0.694** | **0.368** | **0.670** | **0.791** |
| FABIND | | 0.623 | 0.228 | **0.484** | 0.628 |
| UniBind | | 0.567 | 0.199 | 0.367 | 0.514 |
| AudioCLIP* | $\mathcal{A}, (\mathcal{V})$ | 0.503 | 0.209 | 0.384 | 0.496 |
| ViT-Lens* | | 0.500 | 0.149 | 0.332 | 0.451 |
| CENTROBIND | | **0.683** | **0.243** | 0.475 | **0.632** |
| FABIND | | 0.604 | 0.255 | 0.472 | 0.636 |
| UniBind | | 0.506 | 0.126 | 0.280 | 0.429 |
| AudioCLIP* | $\mathcal{V}, (\mathcal{A})$ | 0.501 | 0.080 | 0.199 | 0.326 |
| ViT-Lens* | | 0.533 | 0.219 | 0.438 | 0.575 |
| CENTROBIND | | **0.626** | **0.326** | **0.548** | **0.703** |
| FABIND | | 0.534 | 0.241 | 0.509 | 0.635 |
| UniBind | | 0.514 | 0.091 | 0.248 | 0.365 |
| AudioCLIP* | $\mathcal{T}, (\mathcal{A})$ | 0.477 | 0.088 | 0.309 | 0.439 |
| ViT-Lens* | | 0.475 | 0.070 | 0.214 | 0.329 |
| CENTROBIND | | **0.655** | **0.346** | **0.610** | **0.741** |

## 5 CONCLUSIONS

In this paper, we analyze the limitations of fixed-anchor-bind methods (FABIND), including over-reliance on the choice of anchor modality, and failing to capture both intra and shared information among non-anchored modalities. To overcome such shortcomings, we propose CENTROBIND, which aligns multimodal embeddings to adaptive anchors constructed by centroids of the embeddings, hence removing the need for anchor modality. Moreover, we theoretically study CENTROBIND, showing that it captures intra- and shared information. Extensive experiments on both synthetic and real-world datasets show that CENTROBIND significantly outperforms FABIND, providing a robust unified representation space and validating our analysis on CENTROBIND and FABIND.

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

# A RELATED WORK

## A.1 MULTIMODAL LEARNING

Multimodal learning has gained significant attention in recent years due to its potential to enhance machine learning models by leveraging diverse data modalities, such as text, images, audio, and video. By combining these modalities, multimodal learning seeks to mimic human-like perception, thereby improving performance across a wide range of applications, from healthcare to natural language processing. Common supervised multimodal learning tasks include audio-visual classification (Peng et al., 2022; Feichtenhofer et al., 2019; Zhu & Rahtu, 2022), visual question answering (Antol et al., 2015; Guo et al., 2021), and vision-language tasks (Xu et al., 2015; Radford et al., 2021), as well as more complex vision-audio-language tasks (Aytar et al., 2017; Harwath et al., 2018).

Typically, these models integrate unimodal features extracted by modality-specific encoders (Seichter et al., 2021; Nagrani et al., 2021; Wu et al., 2022; Wang et al., 2020a; Peng et al., 2022). For instance, Madaan et al. (2024) introduce inter- and intra-modality modeling frameworks that treat the target as a composition of multiple modalities. Similarly, Du et al. (2023) propose a late-fusion approach for supervised multimodal tasks, demonstrating that insufficient feature extraction from individual modalities negatively affects the model's generalization ability. Additionally, Zhang et al. (2024) address joint optimization by alternating between unimodal learning scenarios and integrating modality-specific encoders with a unified head shared across all modalities.

## A.2 MULTIMODAL ALIGNMENT

Multimodal learning addresses four key challenges (Liang et al., 2024c; Baltrušaitis et al., 2018; Liang et al., 2024d): managing interactions among redundant, unique, and synergistic features (Dumas et al., 2017; Liang et al., 2024a;b), aligning fine-grained and coarse-grained information (Wang et al., 2023; 2024a), reasoning across diverse features (Yang et al., 2023), and integrating external knowledge (Shen et al., 2022; Lyu et al., 2024). Among these challenges, multimodal alignment is one of the core challenges that many researchers aim to solve.

A common method in multimodal alignment is using cross-modal alignment by using attention mechanisms between pairwise modalities, such as vision-language (Tan & Bansal, 2019) and vision-language-audio (Tsai et al., 2019). Another effective approach is leveraging graph neural networks to align multimodal datasets (Yang et al., 2021; Wilf et al., 2023). For instance, Yang et al. (2021) transforms unaligned multimodal sequence data into nodes, with edges capturing interactions across modalities over time. Wilf et al. (2023) build graph structures for each modality—visual, textual, and acoustic—and create edges to represent their interactions.

To enhance the generalizability of cross-modal representations, Xia et al. (2024) employ a unified codebook approach, facilitating a joint embedding space for visual and audio modalities. Another prominent method (Radford et al., 2021) achieves cross-modal alignment by leveraging large collections of image-text pairs, making it a widely adopted strategy in multimodal learning (Zhang et al., 2022; Guzhov et al., 2022; Zhou et al., 2023).

## A.3 BINDING METHODS

Recent studies have focused on aligning multimodal datasets by leveraging binding properties in various modalities. ImageBind (Girdhar et al., 2023) aligns multimodal data by using image representation as the anchor and aligning each modality embedding with the image embedding. Similarly, LanguageBind (Zhu et al., 2024) uses language representation as the anchor, aligning other modalities into the language space. PointBind (Guo et al., 2023) learns a joint embedding space across 3d point, language, image, and audio modalities by designating the point space as the central representation. Thanks to the efficacy of such a binding idea with a fixed anchor, several "-Bind" approaches have been studied in numerous domains (Teng et al., 2024; Xiao et al., 2024; Gao et al., 2024; Yang et al., 2024b; Balemans et al., 2024; Dhakal et al., 2024; Yang et al., 2024a) While these methods demonstrate strong performance in zero-shot cross-modality retrieval and classification tasks, they are constrained by their reliance on an existing single anchor modality.

Several approaches have integrated additional knowledge into multimodal representation spaces to address this limitation. Freebind (Wang et al., 2024a) introduces bi-modality spaces to enhance a pretrained image-paired unified space. It generates pseudo-embedding pairs across diverse modality pairs and aligns them with the pre-trained unified space using contrastive learning. Omnibind (Wang et al., 2024b) leverages multiple pretrained multimodal models to construct pseudo item-pair retrievals based on top-1 recall across various modality combinations using pairwise cross-modal alignment. Both methods show promising results in cross-modal retrieval by incorporating extra spaces into existing pairwise binding spaces. However, they still rely on fixed (pre-trained) representation spaces.

Unibind (Lyu et al., 2024) highlights the imbalanced representation when using image-centered representation spaces. To address this, Unibind employs large language models (LLMs) to create a unified and balanced representation space. It constructs a knowledge base with multimodal category descriptions, establishes LLM-augmented class-wise embedding centers, and aligns other modalities to these centers through contrastive learning. This approach attempts to balance representations across modalities but still depends heavily on large-scale pretrained LLMs and centers alignment around a single unified space, namely, text (language).

ViT-Lens (Lei et al., 2024) builds upon the Vision Transformer (ViT) (Dosovitskiy et al., 2021) and multimodal foundational models like CLIP (Radford et al., 2021) to align multiple modalities. It extends ViT by incorporating an additional embedding layer and attention layer for each modality, which are trained via contrastive learning involving embeddings generated by the CLIP and the ViT models. This approach generalizes FABIND by allowing more than one fixed anchor modality; specifically, image and text in this case. CENTROBIND could also adopt a similar strategy, leveraging the powerful ViT model for modality alignment while adaptively computing anchors based on their centroids.

## B  PROOFS

### B.1  PROOF OF PROPOSITION 1

Using the chain rule of the mutual information, we observe that

$$I(\mathbf{X}_1, f_1^{\mathrm{suf}}(\mathbf{X}_1); \mathbf{X}_i) = I(\mathbf{X}_1; \mathbf{X}_i) + I(f_1^{\mathrm{suf}}(\mathbf{X}_1); \mathbf{X}_i|\mathbf{X}_1)$$
$$= I(f_1^{\mathrm{suf}}(\mathbf{X}_1); \mathbf{X}_i) + I(\mathbf{X}_1; \mathbf{X}_i|f_1^{\mathrm{suf}}(\mathbf{X}_1)), \quad (12)$$

Since $f_1^{\mathrm{suf}}(\mathbf{X}_1)$ is a deterministic function of $\mathbf{X}_1$, we have

$$I(f_1^{\mathrm{suf}}(\mathbf{X}_1); \mathbf{X}_i|\mathbf{X}_1) = 0. \quad (13)$$

Moreover, $f_1^{\mathrm{suf}}$ obtained in Definition 1 with proper choice of $\mathcal{Z}$ achieves the maximum mutual information, implying together with $I(\mathbf{X}; \mathbf{Y}) \leq \min\{H(\mathbf{X}), H(\mathbf{Y})\}$ that $I(f_1^{\mathrm{suf}}(\mathbf{X}_1); \mathbf{X}_1) = H(\mathbf{X}_1)$, where $H(\mathbf{X}_1)$ is the entropy of $\mathbf{X}_1$ (Polyanskiy & Wu, 2024). In other words, we have $H(\mathbf{X}_1|f_1^{\mathrm{suf}}(\mathbf{X}_1)) = H(\mathbf{X}_1) - I(f_1^{\mathrm{suf}}(\mathbf{X}_1); \mathbf{X}_1) = 0$. This gives

$$I(\mathbf{X}_1; \mathbf{X}_i|f_1^{\mathrm{suf}}(\mathbf{X}_1)) = H(\mathbf{X}_1|f_1^{\mathrm{suf}}(\mathbf{X}_1)) - H(\mathbf{X}_1|f_1^{\mathrm{suf}}(\mathbf{X}_1), \mathbf{X}_i)$$
$$= 0 \quad (14)$$

Substituting (13) and (14) into (12) yields

$$I(\mathbf{X}_1; \mathbf{X}_i) = I(f_1^{\mathrm{suf}}(\mathbf{X}_1); \mathbf{X}_i). \quad (15)$$

We conclude the proof of Proposition 1 by noting that the optimality of FABIND (i.e., $I(f_1^{\mathrm{suf}}(\mathbf{X}_1); \mathbf{X}_i) = I(f_1^{\mathrm{suf}}(\mathbf{X}_1); f_i^{\mathrm{FB}}(\mathbf{X}_i)), \forall i \in \{2, \cdots, M\})$ yields

$$I(\mathbf{X}_1; \mathbf{X}_i) = I(f_1^{\mathrm{suf}}(\mathbf{X}_1); f_i^{\mathrm{FB}}(\mathbf{X}_i)). \quad (16)$$

### B.2  PROOF OF PROPOSITION 2

Using the chain rule of mutual information, we have

$$I(f_1^{\mathrm{ins}}(\mathbf{X}_1); \mathbf{X}_1, \mathbf{X}_i) = I(f_1^{\mathrm{ins}}(\mathbf{X}_1); \mathbf{X}_1) + I(f_1^{\mathrm{ins}}(\mathbf{X}_1); \mathbf{X}_i|\mathbf{X}_1)$$
$$= I(f_1^{\mathrm{ins}}(\mathbf{X}_1); \mathbf{X}_i) + I(f_1^{\mathrm{ins}}(\mathbf{X}_1); \mathbf{X}_1|\mathbf{X}_i). \quad (17)$$

Moreover, since $f_1^{\text{ins}}(\mathbf{X}_1)$ is a deterministic function of $\mathbf{X}_1$, we have $I(f_1^{\text{ins}}(\mathbf{X}_1); \mathbf{X}_i|\mathbf{X}_1) = 0$, leading to $I(f_1^{\text{ins}}(\mathbf{X}_1); \mathbf{X}_1) = I(f_1^{\text{ins}}(\mathbf{X}_1); \mathbf{X}_i) + I(f_1^{\text{ins}}(\mathbf{X}_1); \mathbf{X}_1|\mathbf{X}_i)$. Then, using the assumption $I(f_1^{\text{ins}}(\mathbf{X}_1); \mathbf{X}_1) < \epsilon$, it follows that

$$
\begin{aligned}
\epsilon &> I(f_1^{\text{ins}}(\mathbf{X}_1); \mathbf{X}_i) + I(f_1^{\text{ins}}(\mathbf{X}_1); \mathbf{X}_1|\mathbf{X}_i) \\
&\overset{(a)}{\geq} I(f_1^{\text{ins}}(\mathbf{X}_1); \mathbf{X}_i) \\
&\overset{(b)}{\geq} I(f_1^{\text{ins}}(\mathbf{X}_1); f_i^{\text{FB}}(\mathbf{X}_i)),
\end{aligned}
\tag{18}
$$

where the labeled inequalities follow from: (a) the non-negativity of mutual information; (b) the data processing inequality. This concludes the proof of Proposition 2.

### B.3 PROOF OF THEOREM 1

To prove Theorem 1, we leverage the reverse inequality of $M$-variable Hölder inequality (Seo, 2013, eq. (2.8)). For the sake of completeness, we state the inequality in Lemma 1.

**Lemma 1** (Reverse inequality of the $M$-variable Hölder inequality (Seo, 2013)). *Consider $M$ sequences $(x_{i,j})_{j \in [n]}$, $i \in [M]$ of $n$ positive scalars such that for some $0 < c_m \leq c_M < \infty$,*

$$
0 < c_m \leq x_{i,j} \leq c_M < \infty, \ \forall i, j.
\tag{19}
$$

*Then,*

$$
\prod_{i=1}^{M} \left( \sum_{j=1}^{n} x_{i,j} \right)^{\frac{1}{n}} \leq \frac{(c_m + c_M)^2}{4 c_m c_M} \sum_{j=1}^{n} \left( \prod_{i=1}^{M} x_{i,j} \right)^{\frac{1}{n}}.
\tag{20}
$$

Now we start by writing the summation of InfoNCE losses for each $f_l^{(t)}(\boldsymbol{x}_{l,k}'), l \in [M]$ to $f_i(\mathbf{X}_i)$ as

$$
\sum_{l=1}^{M} I_{\text{NCE}}(f_l(\mathbf{X}_l'); f_i(\mathbf{X}_i)|\tau) = -\frac{1}{|\mathcal{I}_B|} \sum_{k=1}^{|\mathcal{I}_B|} \sum_{l=1}^{M} \log \frac{\exp\left(\frac{f_l^\top(\boldsymbol{x}_{l,k}') f_i(\boldsymbol{x}_{i,k})}{\tau}\right)}{\sum_{j \in \mathcal{I}_B} \exp\left(\frac{f_l^\top(\boldsymbol{x}_{l,k}') f_i(\boldsymbol{x}_{i,j})}{\tau}\right)}.
\tag{21}
$$

Then, the inner summation in (21) is bounded as

$$
\begin{aligned}
&\sum_{l=1}^{M} \log \frac{\exp\left(\frac{f_l^\top(\boldsymbol{x}_{l,k}') f_i(\boldsymbol{x}_{i,k})}{\tau}\right)}{\sum_{j \in \mathcal{I}_B} \exp\left(\frac{f_l^\top(\boldsymbol{x}_{l,k}') f_i(\boldsymbol{x}_{i,j})}{\tau}\right)} \\
&= \frac{1}{\tau} \sum_{l=1}^{M} f_l^\top(\boldsymbol{x}_{l,k}') f_i(\boldsymbol{x}_{i,k}) - \log \prod_{l=1}^{M} \sum_{j \in \mathcal{I}_B} \exp\left(\frac{f_l^\top(\boldsymbol{x}_{l,k}') f_i(\boldsymbol{x}_{i,j})}{\tau}\right) \\
&\overset{(a)}{\geq} \frac{1}{\tau} \sum_{l=1}^{M} f_l^\top(\boldsymbol{x}_{l,k}') f_i(\boldsymbol{x}_{i,k}) - \log \left( C_{\mathcal{F},k,i} \sum_{j \in \mathcal{I}_B} \prod_{l=1}^{M} \exp\left(\frac{f_l^\top(\boldsymbol{x}_{l,k}') f_i(\boldsymbol{x}_{i,j})}{\tau |\mathcal{I}_B|}\right) \right)^{|\mathcal{I}_B|} \\
&\overset{(b)}{=} \frac{M}{\tau} \boldsymbol{a}_k^\top f_i(\boldsymbol{x}_{i,k}) - |\mathcal{I}_B| \log \sum_{j \in \mathcal{I}_B} \exp\left(\frac{M \boldsymbol{a}_k^\top f_i(\boldsymbol{x}_{i,j})}{\tau |\mathcal{I}_B|}\right) - |\mathcal{I}_B| \log C_{\mathcal{F},k,i} \\
&= |\mathcal{I}_B| \log \exp\left(\frac{M \boldsymbol{a}_k^\top f_i(\boldsymbol{x}_{i,k})}{\tau |\mathcal{I}_B|}\right) - |\mathcal{I}_B| \log \sum_{j \in \mathcal{I}_B} \exp\left(\frac{M \boldsymbol{a}_k^\top f_i(\boldsymbol{x}_{i,j})}{\tau |\mathcal{I}_B|}\right) - |\mathcal{I}_B| \log C_{\mathcal{F},k,i} \\
&= |\mathcal{I}_B| \log \frac{\exp\left(\frac{M \boldsymbol{a}_k^\top f_i(\boldsymbol{x}_{i,k})}{\tau |\mathcal{I}_B|}\right)}{\sum_{j \in \mathcal{I}_B} \exp\left(\frac{M \boldsymbol{a}_k^\top f_i(\boldsymbol{x}_{i,j})}{\tau |\mathcal{I}_B|}\right)} - |\mathcal{I}_B| \log C_{\mathcal{F},k,i},
\end{aligned}
\tag{22}
$$

where the labeled (in)equalities follow from: (a) Lemma 1 and $C_{\mathcal{F},k,i} = \frac{(c_{\mathcal{F},k,i}^{\min} + c_{\mathcal{F},k,i}^{\max})^2}{4 c_{\mathcal{F},k,i}^{\min} c_{\mathcal{F},k,i}^{\max}}$ with

$$c_{\mathcal{F},k,i}^{\min} = \min_{\ell \in [M], j \in \mathcal{I}_B} \exp\left( \frac{f_l^\top(\boldsymbol{x}_{l,k}') f_i(\boldsymbol{x}_{i,j})}{\tau} \right), \text{ and }$$

$$c_{\mathcal{F},k,i}^{\max} = \max_{\ell \in [M], j \in \mathcal{I}_B} \exp\left( \frac{f_l^\top(\boldsymbol{x}_{l,k}') f_i(\boldsymbol{x}_{i,j})}{\tau} \right); \tag{23}$$

and (b) the definition of anchor embedding (7). Substituting (22) into (21) gives

$$\sum_{l=1}^{M} I_{\text{NCE}}(f_l(\mathbf{X}_l'); f_i(\mathbf{X}_i)|\tau) \leq -\frac{1}{|\mathcal{I}_B|} \sum_{k=1}^{|\mathcal{I}_B|} \left[ |\mathcal{I}_B| \log \frac{\exp\left( \frac{M \boldsymbol{a}_k^\top f_i(\boldsymbol{x}_{i,k})}{\tau |\mathcal{I}_B|} \right)}{\sum_{j \in \mathcal{I}_B} \exp\left( \frac{M \boldsymbol{a}_k^\top f_i(\boldsymbol{x}_{i,j})}{\tau |\mathcal{I}_B|} \right)} - |\mathcal{I}_B| \log C_{\mathcal{F},k,i} \right]$$

$$= |\mathcal{I}_B| I_{\text{NCE}}\left( \mathbf{A}; f_i(\mathbf{X}_i) \,\bigg|\, \frac{\tau |\mathcal{I}_B|}{M} \right) + \sum_{k=1}^{|\mathcal{I}_B|} \log C_{\mathcal{F},k,i}. \tag{24}$$

Rearranging (24) and setting $\tilde{\tau} = \frac{\tau |\mathcal{I}_B|}{M}$ in (23) and (24) yield

$$I_{\text{NCE}}\left( \mathbf{A}; f_i(\mathbf{X}_i) \mid \tilde{\tau} \right) \geq \frac{1}{|\mathcal{I}_B|} \sum_{l=1}^{M} I_{\text{NCE}}\left( f_l(\mathbf{X}_l'); f_i(\mathbf{X}_i) \,\bigg|\, \frac{\tilde{\tau} M}{|\mathcal{I}_B|} \right) - \frac{1}{|\mathcal{I}_B|} \sum_{k=1}^{|\mathcal{I}_B|} \log C_{\mathcal{F},k,i}, \tag{25}$$

which concludes the proof of Theorem 1.

## C EXPERIMENT DETAILS

### C.1 EXPERIMENTS WITH SYNTHETIC DATASETS

**Synthetic datasets.** We employ a latent variable model (Bishop & Nasrabadi, 2006) for generating synthetic multimodal datasets. A latent variable model is a statistical model for data $\mathbf{X} \in \mathbb{R}^{d_x}$, under which $\mathbf{X}$ is generated according to a conditional probability distribution $P_{\mathbf{X}|\mathbf{Z}}$, where $\mathbf{Z} \in \mathbb{R}^{d_z}$ is the latent variable. In terms of the representation learning framework, $\mathbf{Z}$ can be seen as a true representation of $\mathbf{X}$. Moreover, we assume that the class label $\mathbf{Y} \in [K]$ and the latent variable $\mathbf{Z}$ are jointly distributed according to $P_{\mathbf{Z},\mathbf{Y}}$.

For the marginal distribution of $\mathbf{Z}$, we make use of a Gaussian mixture model (GMM) (Bishop & Nasrabadi, 2006), and hence the probability density function (PDF) of $\mathbf{Z}$ is a weighted sum of Gaussian densities. In particular, the PDF of $\mathbf{Z}$ is defined as follows:

$$p_{\mathbf{Z}}(\boldsymbol{z}) = \prod_{y=1}^{K} \pi_y \mathcal{N}(\boldsymbol{z}; \boldsymbol{\mu}_y, \boldsymbol{\Sigma}_y), \tag{26}$$

where $K$ is the number of mixture components, $\pi_y = \Pr(\mathbf{Y} = y)$ is the component prior probability, and $\mathcal{N}(\boldsymbol{z}; \boldsymbol{\mu}_y, \boldsymbol{\Sigma}_y)$ denotes Gaussian PDF with mean $\boldsymbol{\mu}_y \in \mathbb{R}^{d_z}$ and covariance matrix $\boldsymbol{\Sigma}_y \in \mathbb{R}^{d_z \times d_z}$. This leads to the conditional PDF of $\mathbf{Z}$ as $p_{\mathbf{Z}|\mathbf{Y}}(\boldsymbol{z}|y) = \mathcal{N}(\boldsymbol{z}; \boldsymbol{\mu}_y, \boldsymbol{\Sigma}_y)$.

Once a latent variable $\boldsymbol{z}$ is generated from GMM in (26), we generate data samples $(\boldsymbol{x}_{i,1}, \boldsymbol{x}_{i,2}, \cdots, \boldsymbol{x}_{i,N})$ for $i$-th modality using the conditional PDFs of $\mathbf{X}_i$ given $\boldsymbol{z}$, denoted by $p_{\mathbf{X}_i|\mathbf{Z}}(\boldsymbol{x}_i|\boldsymbol{z})$. Specifically, we use the model $\mathbf{X}_i = g_i(\mathbf{Z}_i) + \mathbf{N}$, where $g_i : \mathbb{R}^{d_z} \to \mathbb{R}^{d_x}$ is a non-linear projection from latent space to observation space, and $\mathbf{N} \sim \mathcal{N}(\mathbf{0}, I_{d_x})$ is Gaussian noise with zero-mean and identity covariance matrix. To make the inherent correlation between $\mathbf{X}_i$ and $\mathbf{Z}_i$ different among modalities, we choose $g_i$ such that

$$g_i(\mathbf{Z}) = \Theta_i^{(2)} \text{sigmoid}\left( \Theta_i^{(1)} \mathbf{Z} \right), \tag{27}$$

where $\text{sigmoid}(x) = \frac{1}{1 + e^{-x}}$ is applied element-wise, and $\Theta_i^{(1)} \in \mathbb{R}^{d_x \times d_z}$ and $\Theta_i^{(2)} \in \mathbb{R}^{d_x \times d_x}$ are matrices randomly generated from Gaussian distribution. Moreover, after $\Theta_i^{(1)}, i \in [M]$ are

Table 3: Classification accuracies presented in Figure 2.

| Backbone | Method | Unimodal | | | | Multimodal |
|---|---|---|---|---|---|---|
| | | $\mathbf{X}_1$ | $\mathbf{X}_2$ | $\mathbf{X}_3$ | $\mathbf{X}_4$ | $\mathbf{X}_1, \cdots, \mathbf{X}_4$ |
| Pre-trained | × | 0.2166 | 0.2878 | 0.3536 | 0.3923 | 0.6985 |
| | FABIND-$\mathbf{X}_1$ | 0.2180 | 0.2736 | 0.3210 | 0.2999 | 0.5541 |
| | FABIND-$\mathbf{X}_4$ | 0.2483 | 0.3349 | **0.4207** | 0.3896 | **0.7024** |
| | CENTROBIND | **0.2540** | **0.3433** | 0.4162 | **0.4559** | 0.6974 |
| Random | × | 0.2109 | 0.2472 | 0.2597 | 0.2815 | 0.6648 |
| | FABIND-$\mathbf{X}_1$ | 0.2119 | 0.2587 | 0.3034 | 0.3081 | 0.5502 |
| | FABIND-$\mathbf{X}_4$ | 0.2447 | 0.3076 | 0.3826 | 0.2813 | 0.6742 |
| | CENTROBIND | **0.2582** | **0.3392** | **0.4224** | **0.4649** | **0.7006** |

generated, we set arbitrary columns of them all zero, so that the number of all zero columns decreases in $i$. For example, $60\%$ of columns of $\Theta_1^{(1)}$ are all-zero, while only $10\%$ of columns of $\Theta_M^{(1)}$ are all-zero. This enables approximate control the correlation between $\mathbf{X}_i$ and $\mathbf{Z}$, providing estimates of best modality ($\mathbf{X}_M$) or worst modality ($\mathbf{X}_1$). To have meaningful labels for this latent model, which requires for downstream tasks, we set the labels $\mathbf{Y}$ being the component index in GMM. In particular, since there are $K$ components in GMM (26), there exists $K$ categories in $\mathbf{Y}$. We conduct experiments with three different synthetic datasets by setting $M = 4, 6, 8$. For all synthetic datasets, we fix $d_x = 16$, $d_z = 8$, and $K = 50$.

**Experiment details.** We initialize two different versions of backbones for all modalities, where the first is a random backbone (highlighted by (rnd) in figures), and the second is a backbone pretrained with InfoNCE loss. For each backbone, we use a simple multilayer perceptron (MLP). Comparing the results with these two versions of backbone provides how much both FABIND and CENTROBIND are robust to backbone quality. Given the backbones for $M$ modalities, we align the corresponding embedding spaces using either FABIND with anchor $\mathbf{X}_i$ (denoted by $\mathbf{X}_i$-B in figures) or CENTROBIND (denoted by CB in figures). Finally, with the encoders aligned by either FABIND or CENTROBIND, we evaluate classification accuracy as a measure of representation quality. We use a simple MLP for the classifier. To distinguish between accuracy with embeddings from a single modality and the one with concatenated embeddings from all modalities, we denote by acc($\mathbf{Z}_i$) the accuracy with embeddings from $i$-th modality and by acc(All) the accuracy with embeddings from all modalities. Specifically, for acc(All), we fuse the multimodal embeddings using MLP layers. Therefore, the accuracy of the multimodal case without binding methods (e.g., × method and the multimodal column in Table 3) can be considered a naive baseline for multimodal learning.

**Comparison with baseline methods.** Figure 2 shows the validation accuracy of each method (without binding, FABIND with anchor $\mathbf{X}_1$, FABIND with anchor $\mathbf{X}_4$, and CENTROBIND). For the same experimental setting, Figure 4 includes additional accuracy curves for $acc(\mathbf{Z}_1)$ and $acc(All)$. For better readability, the corresponding accuracy is provided in Table 3.

We conduct experiments with two types of backbone encoders: randomly initialized backbones and pre-trained backbones. For each type, we extract embeddings using four different methods: representations without binding (denoted by × in Table 5), FABIND with anchor modality $\mathbf{X}_1$ (denoted as FABIND-$\mathbf{X}_1$), FABIND with anchor modality $\mathbf{X}_4$ (denoted as FABIND-$\mathbf{X}_4$), and CENTROBIND. The embedding quality is then evaluated using classification accuracy. Specifically, we train five different decoders for each case: four unimodal decoders (one for each modality) and one multimodal decoder for the concatenated embeddings of all modalities. The results show that CENTROBIND outperforms the other baseline methods. Notably, CENTROBIND demonstrates superior performance in the case of randomly initialized backbones, indicating robustness to poor backbone quality.

Additional experimental results on synthetic datasets with $M = 6$ and $M = 8$ modalities are presented in Figure 7 and Figure 8, respectively. These results exhibit similar trends to those observed with $M = 4$ modalities. These experiments verify that CENTROBIND is capable of handling a large number of modalities effectively.

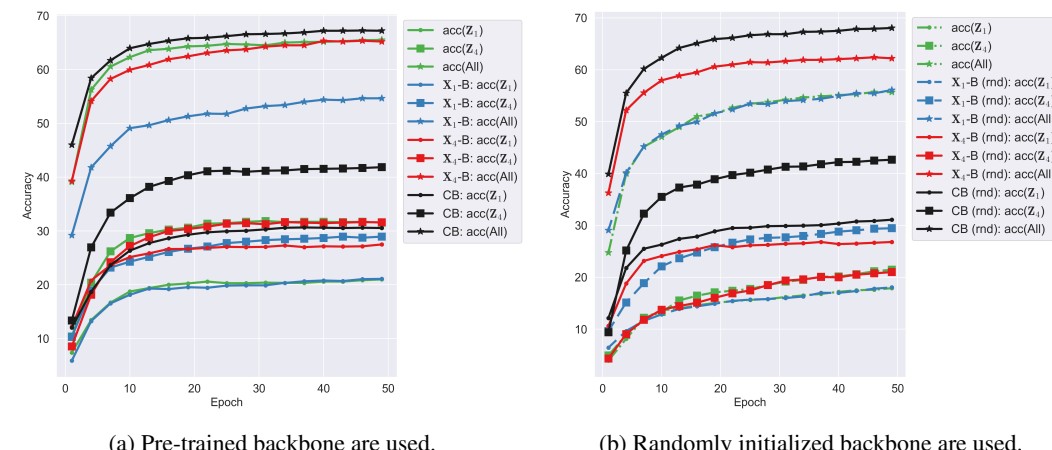

(a) Pre-trained backbone are used.

(b) Randomly initialized backbone are used.

Figure 4: Accuracy as a measure of the representation space quality. Abbreviation: $\mathbf{X}_i$-B or CB: applying FABIND with anchor $\mathbf{X}_i$ or applying CentroBind; acc($\mathbf{Z}_i$) or acc(All): accuracy of $\mathbf{Z}_i$ or of concatenated embeddings ($\mathbf{Z}_1, \cdots, \mathbf{Z}_M$); (rnd): if random backbones are used.

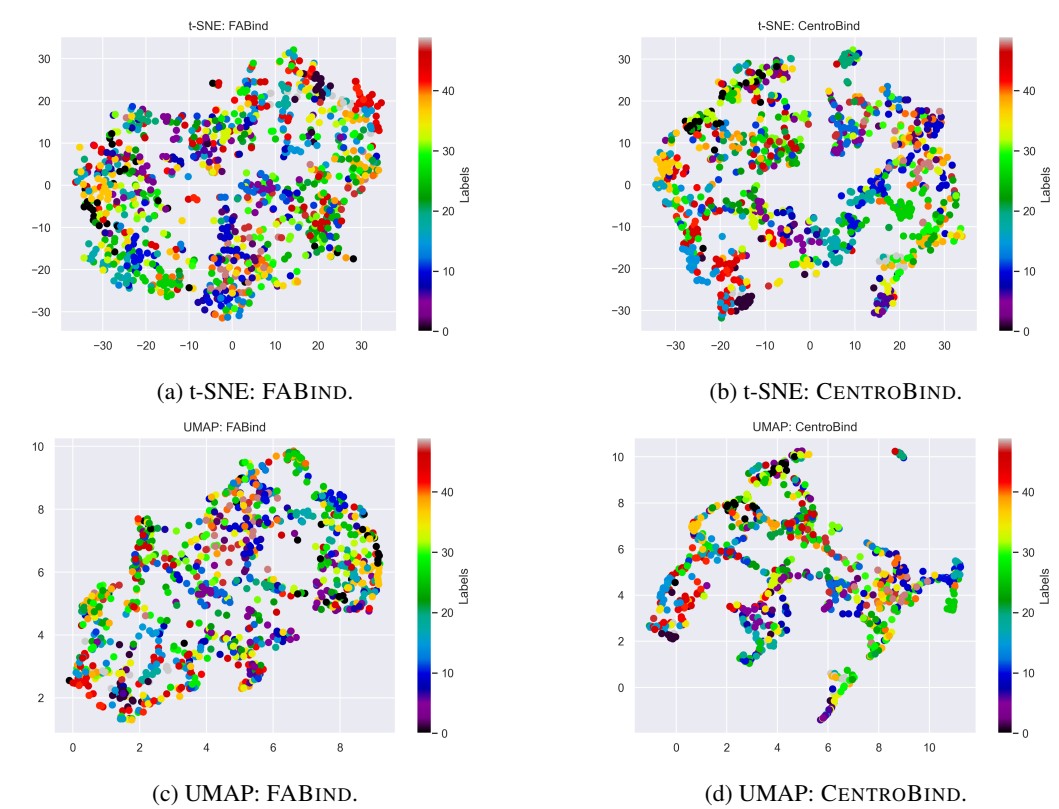

(a) t-SNE: FABIND.

(b) t-SNE: CENTROBIND.

(c) UMAP: FABIND.

(d) UMAP: CENTROBIND.

Figure 5: Representation visualization via t-SNE and UMAP.

**Representation visualization.** Figure 5 presents t-SNE (Van der Maaten & Hinton, 2008) and UMAP (McInnes et al., 2018) visualizations of embeddings generated by FABIND and CENTROBIND. For this visualization, we use synthetic datasets with 4 modalities, ensuring that each modality is equally informative, and plot the embeddings for $\mathbf{X}_1$. FABIND is anchored at $\mathbf{X}_4$, and both binding methods utilize pre-trained backbones.

In both t-SNE and UMAP visualizations, CENTROBIND produces more clustered representations, whereas FABIND results in more scattered embeddings. These findings validate our analysis that CENTROBIND creates a superior representation space by effectively learning both intra- and shared information.

**Convergence and stability analysis.** The convergence rate of CENTROBIND may differ from that of FABIND due to the replacement of the fixed anchor with a dynamic anchor. In Figure 6, we plot the loss curves of CENTROBIND and FABIND during training. The results show that the loss of CENTROBIND saturates earlier than that of FABIND. We attribute this to the fact that the centroid serves as a minimizer of embeddings in terms of Euclidean distance, making it easier to converge embeddings to their centroid compared to converging them to one specific embedding.

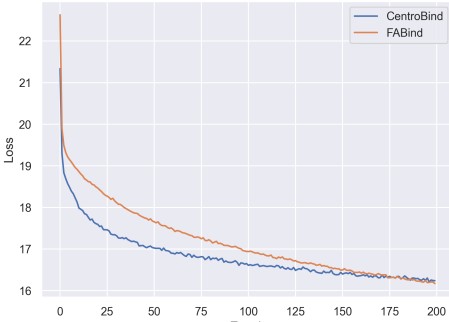

Figure 6: Training loss.

The plot also reveals a crossover point where the loss curves intersect. We believe this occurs due to the number of InfoNCE losses optimized by CENTROBIND and FABIND. Specifically, with $M$ modalities, CENTROBIND minimizes $M$ InfoNCE losses, while FABIND minimizes $M - 1$ InfoNCE losses. This results in a smaller loss for FABIND when the encoders are well-trained, which explains the crossover point observed in Figure 6.

Table 4: Classification accuracies presented in Figure 9. In the experiment in Figure 9a and 9b, $\mathbf{X}_1, \mathbf{X}_2,$ and $\mathbf{X}_3$ are very noisy, and $\mathbf{X}_4$ is highly informative. In the experiment in Figure 9c and 9d, $\mathbf{X}_1$ and $\mathbf{X}_2$ are very noisy, and $\mathbf{X}_3$ and $\mathbf{X}_4$ are highly informative. We choose $\mathbf{X}_4$ for FABIND for the best fixed anchor modality for both cases. Weighted average method uses prior knowledge of modality quality to determine the weights for each modality. Random anchor method without intra learning uses randomly chosen modality as an anchor for each iteration under fixed anchor encoder, while with intra learning we train intra modal learning by not freezing the anchor encoder.

| Backbone | Method | Figure 9a and 9b | | | Figure 9c and 9d | | |
|---|---|---|---|---|---|---|---|
| | | $\mathbf{X}_2$ | $\mathbf{X}_4$ | $\mathbf{X}_1, \cdots, \mathbf{X}_4$ | $\mathbf{X}_2$ | $\mathbf{X}_4$ | $\mathbf{X}_1, \cdots, \mathbf{X}_4$ |
| Pre-trained | × | 0.115 | 0.296 | 0.566 | 0.099 | 0.256 | 0.537 |
| | FABIND-$\mathbf{X}_4$ | 0.124 | 0.297 | **0.639** | 0.115 | 0.263 | 0.540 |
| | CENTROBIND | 0.131 | **0.363** | 0.618 | **0.116** | 0.336 | 0.563 |
| | Weighted average | 0.133 | 0.342 | 0.609 | 0.102 | 0.338 | 0.574 |
| | Random + intra learning | 0.131 | 0.347 | 0.613 | 0.105 | 0.353 | 0.554 |
| | Random anchor | **0.147** | 0.359 | 0.619 | 0.097 | 0.327 | 0.579 |
| | Median (coordinate-wise) | 0.134 | **0.363** | 0.634 | 0.112 | **0.375** | **0.582** |
| Random | × | 0.092 | 0.114 | 0.487 | 0.067 | 0.176 | 0.465 |
| | FABIND-$\mathbf{X}_4$ | 0.131 | 0.143 | 0.575 | 0.112 | 0.153 | 0.523 |
| | CENTROBIND | 0.132 | **0.355** | **0.626** | **0.113** | **0.336** | 0.559 |
| | Weighted average | 0.115 | 0.347 | 0.619 | 0.109 | **0.336** | 0.556 |
| | Random + intra learning | 0.132 | 0.333 | 0.602 | 0.104 | 0.309 | 0.562 |
| | Random anchor | **0.145** | 0.354 | 0.618 | 0.097 | 0.324 | 0.552 |
| | Median (coordinate-wise) | 0.137 | 0.347 | 0.612 | 0.112 | 0.330 | **0.565** |

**Comparison with other adaptive anchor generation.** We compare the centroid-based adaptive anchor method with other potential approaches, such as weighted average, random anchor fixing, and component-wise median. Figure 9 illustrates the accuracies of each method under scenarios where modalities are unevenly distributed. Specifically, we create 4 modalities with differing quality levels. In experiments (a) and (b) of Figure 9, $\mathbf{X}_1, \mathbf{X}_2,$ and $\mathbf{X}_3$ are set as highly uninformative, while $\mathbf{X}_4$ represents a high-quality dataset. Conversely, experiments (c) and (d) use $\mathbf{X}_1$ and $\mathbf{X}_2$ as poor-quality datasets, while $\mathbf{X}_3$ and $\mathbf{X}_4$ are high-quality datasets.

For the weighted average method (denoted as WAB in Figure 9), we assign weights based on modality quality: $(0.2, 0.2, 0.2, 1)$ for experiments (a) and (b), and $(0.2, 0.2, 0.8, 0.8)$ for experiments (c) and (d). These weights correspond to the information rate of each modality.

For the random modality dynamic anchor method (denoted as RB in Figure 9), we randomly select one modality as the dynamic anchor at each iteration, with the anchor encoder frozen. To investigate the impact of intra-modal learning, we also conduct experiments with a random anchor that includes intra-information learning (denoted as RB+Intra). In this case, the anchor modality is randomly selected at each iteration, and the anchor encoder is not frozen, allowing all encoders to be trained.

Since the median is more robust to outliers than the average (Lopuhaä & Rousseeuw, 1991), we additionally evaluate the case of a median-based dynamic anchor. In high-dimensional spaces, rather than in the univariate case, a coordinate-wise median can be used as a naive generalization of the univariate median to the multivariate setting, preserving its robustness to outliers. We assess the dynamic anchor binding method using the coordinate-wise median approach (denoted as MB in Figure 9). Specifically, for the median anchor, we compute the $j$th coordinate of the $i$th anchor as $\boldsymbol{a}_{i,j} = \mathrm{Median}(\boldsymbol{z}_{1,i,j}, \boldsymbol{z}_{2,i,j}, \cdots, \boldsymbol{z}_{M,i,j})$, where $\boldsymbol{z}_{m,i,j}$ denotes the $j$th coordinate of the embedding for the $i$th sample in modality $m$. For improved readability, we summarize the final accuracies for each method and modality in Table 4.

This scenario, where modal distributions are uneven, is commonly referred to as the *modality imbalance* problem (Du et al., 2023; Peng et al., 2022; Zhang et al., 2024). Intuitively, in the presence of modality imbalance, the centroid may produce suboptimal dynamic anchor constructions, and other methods, such as weighted average, might yield better results. Nevertheless, CENTROBIND consistently performs better or comparably to weighted average methods, demonstrating its robustness to the modality imbalance problem.

From these experiments, we conjecture that the specific dynamic anchor generation method may not significantly impact final performance, provided that all encoders are well-trained during the process.

Addressing the modality imbalance problem typically requires additional information, such as domain knowledge, labels, or downstream task insights. Since this work focuses on multimodal alignment under contrastive learning, we do not assume such information is available. We therefore leave the exploration of the modality imbalance problem for dynamic anchor generation as a direction for future work.

## C.2 Experiments with real-world datasets

**Training details.** We utilize Low-Rank Adaptation (Hu et al., 2022) for training CENTROBIND and FABIND, enhancing training efficiency and achieving impressive results with fewer iterations. For parameter settings, we set a learning rate of $0.001$, the AdamW optimizer (Loshchilov & Hutter, 2019) with a batch size of $16$, and a temperature of $0.3$ for InfoNCE. Training CENTROBIND requires augmentation. We augment video frames with various transformations, including random perspective shifts, random flips and rotation, color jitter, Gaussian blur, and auto-contrast adjustment. For the audio modality, we apply a low-pass filter, speed changes, echo effect, room impulse response convolution, and background noise. For the text modality, we generate paraphrased sentences using the Phi-3 language model served using Ollama [5].

**UniBind** We evaluate UniBind as a baseline method, using LLM-generated descriptions as the anchor modality. Specifically, UniBind generates descriptions for each modality using a large language model (LLM), ensuring that every modality is paired with corresponding descriptions. These descriptions collectively form a knowledge base, and UniBind optimizes the InfoNCE loss between each modality and its paired description from the knowledge base. In this framework, the anchor modality is the LLM-augmented representation. It is important to note that the LLM-generated descriptions for different modality pairs can vary, which may hinder effective multimodal alignment (see Table 2). In our experiments, we generate descriptions for video and audio modalities using the VideoLLaMA2.1-7B-AV audio-visual model from VideoLLaMA2 (Cheng et al., 2024), and for the text modality, we use the Qwen2.5-32B-Instruct model from Qwen2.5 (Team, 2024). We evaluate

---

[5]https://ollama.com/library/phi3

Table 5: Classification accuracy evaluated on each modality (training and evaluation modalities are the same) with MUStARD dataset. Asterisk* denotes different backbone encoders and pretraining settings.

| Method | Modality | Sar-1 | Spk-1 | Spk-3 | Spk-5 |
|---|---|---|---|---|---|
| FABIND | | 0.606 | 0.219 | 0.458 | 0.632 |
| UniBind | | 0.600 | 0.214 | 0.412 | 0.569 |
| AudioCLIP* | $\mathcal{T}$ | 0.488 | 0.155 | 0.280 | 0.388 |
| ViT-Lens* | | 0.543 | 0.172 | 0.342 | 0.472 |
| CENTROBIND | | **0.667** | **0.287** | **0.507** | **0.642** |
| FABIND | | 0.668 | 0.375 | 0.587 | 0.691 |
| UniBind | | 0.658 | 0.381 | 0.641 | 0.770 |
| AudioCLIP* | $\mathcal{V}$ | 0.504 | 0.110 | 0.275 | 0.414 |
| ViT-Lens* | | **0.697** | **0.586** | **0.738** | **0.797** |
| CENTROBIND | | 0.670 | 0.380 | 0.609 | 0.726 |
| FABIND | | 0.639 | 0.201 | 0.457 | 0.599 |
| UniBind | | 0.633 | 0.272 | 0.528 | 0.691 |
| AudioCLIP* | $\mathcal{A}$ | 0.525 | 0.158 | 0.343 | 0.454 |
| ViT-Lens* | | **0.686** | **0.396** | **0.664** | **0.8** |
| CENTROBIND | | 0.616 | 0.234 | 0.461 | 0.609 |
| FactorCL* | | 0.699 | - | - | - |
| SimMMDG* | | 0.725 | - | - | - |
| FABIND | $\mathcal{V},\mathcal{A},\mathcal{T}$ | 0.678 | 0.343 | 0.554 | 0.677 |
| UniBind | $(\mathcal{V},\mathcal{A},\mathcal{T})$ | 0.646 | 0.383 | 0.622 | 0.764 |
| AudioCLIP* | | 0.530 | 0.119 | 0.261 | 0.378 |
| ViT-Lens* | | **0.731** | **0.506** | **0.736** | **0.812** |
| CENTROBIND | | 0.704 | 0.346 | 0.594 | 0.733 |

UniBind's performance in two settings: standard classification accuracy (Table 5) and zero-shot cross-modal classification (Table 2).

**AudioCLIP**   We employ AudioCLIP (Guzhov et al., 2022), which aligns image, text, and audio representations into a unified multimodal space. To extend its capabilities to the video modality in our experiments, we adapt AudioCLIP to extract embeddings for video, audio, and text modalities using a pretrained model. For audio, we follow AudioCLIP's approach, padding audio samples to ensure uniform input sizes. For text, we utilize its pretrained settings, truncating tokenized text to 77 tokens, which only occurs in one instance. For the video modality, we use the center frame as a representative image sample. Finally, embeddings from all three modalities are concatenated for downstream tasks.

**ViT-Lens**   In our experiments, we leverage the pretrained models from ViT-Lens to extract embeddings for audio, text, and video modalities. We generally follow the example code[6] provided by the authors. Note that we select the center frame image from the video to extract the embedding.

**Classification results.**   In contrast to the cross-modal retrieval results in Table 1 and zero-shot cross-modal classification in Table 2, Table 5 presents the classification accuracy of FABIND, UniBind, and CENTROBIND for each modality as well as for multimodal scenarios. Specifically, embeddings are extracted using the binding methods, and a simple decoder is trained to classify the embeddings. In Table 5, we report the sarcasm and speaker classification accuracies of decoders trained and evaluated on the same modality.

For sarcasm detection, CENTROBIND generally outperforms other baseline methods. While UniBind performs poorly in cross-modal classification, it achieves better performance in speaker classification compared to others. This improvement is due to the LLM-augmented descriptions, which provide additional knowledge (from LLMs) to the embeddings. Notably, UniBind utilizes 4 modalities, whereas FABIND and CENTROBIND only use 3, which could penalize the performance of FABIND and CENTROBIND . Nevertheless, CENTROBIND consistently outperforms FABIND. Moreover,

---
[6]https://github.com/TencentARC/ViT-Lens

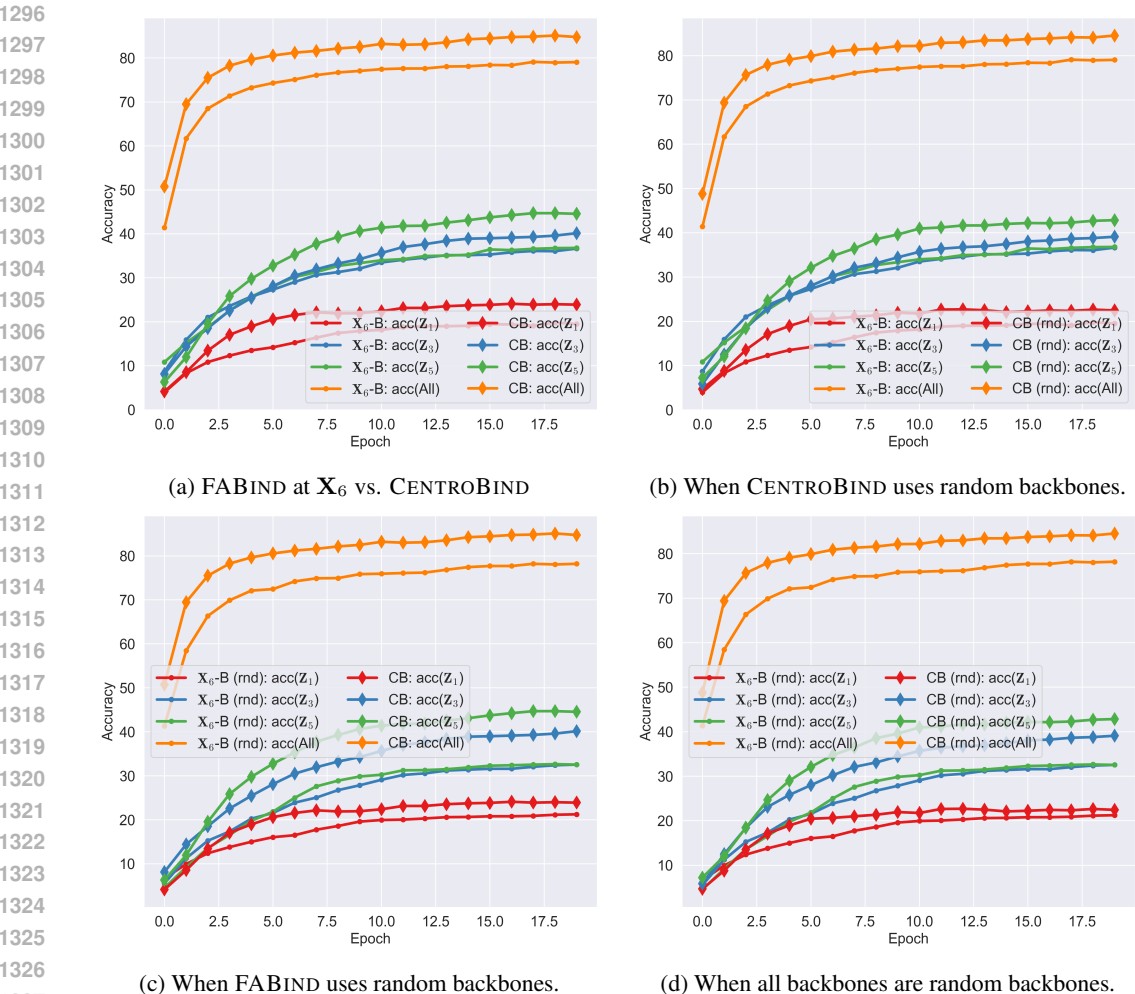

(a) FABIND at $\mathbf{X}_6$ vs. CENTROBIND

(b) When CENTROBIND uses random backbones.

(c) When FABIND uses random backbones.

(d) When all backbones are random backbones.

Figure 7: Experiment results with synthetic dataset of $M = 6$ modalities. Abbreviation: $\mathbf{X}_i$-B or CB: applying FABIND method to backbones with anchor $\mathbf{X}_i$ or applying CENTROBIND; acc($\mathbf{Z}_i$) or acc(All): accuracy of $\mathbf{Z}_i$ or of concatenated embeddings $(\mathbf{Z}_1, \cdots, \mathbf{Z}_M)$; (rnd): if random backbones are used for $\mathbf{X}_i$-B or CB.

our method can also incorporate LLM-augmented descriptions as an additional modality, potentially improving its performance further.

Although a direct comparison is not feasible, we also include the sarcasm detection accuracy of FactorCL (Liang et al., 2024b), SimMMDG (Dong et al., 2023), AudioCLIP (Guzhov et al., 2022), and ViT-Lens (Lei et al., 2024) for reference. ViT-Lens, in particular, achieves higher performance than CENTROBIND due to its use of larger backbone encoders, such as Vision Transformer (ViT) (Khan et al., 2022) and pretraining on extremely large-scale datasets. However, since ViT-Lens can be considered a variant of FABind, applying our dynamic anchor method could further improve its performance. Specifically, ViT-Lens uses a pretrained CLIP model as the anchor encoder, while the other non-anchored modalities use pretrained ViT models with modality adaptation layers. Within our framework, CENTROBIND could adopt the pretrained Vision Transformer as backbone encoders, potentially enhancing its performance further.

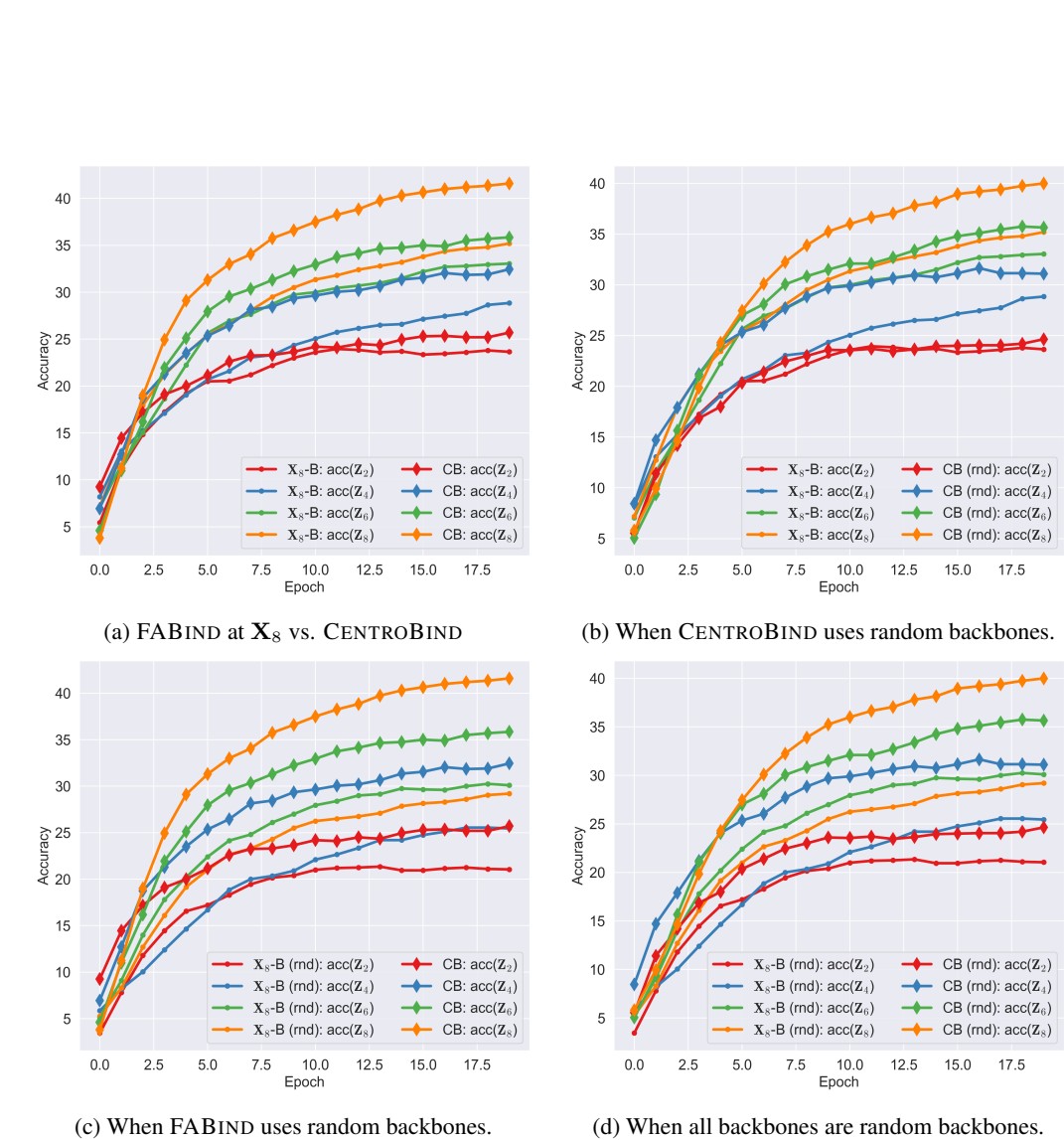

(a) FABIND at $\mathbf{X}_8$ vs. CENTROBIND

(b) When CENTROBIND uses random backbones.

(c) When FABIND uses random backbones.

(d) When all backbones are random backbones.

Figure 8: Experiment results with synthetic dataset of $M = 8$ modalities. Abbreviation: $\mathbf{X}_i$-B or CB: applying FABIND method to backbones with anchor $\mathbf{X}_i$ or applying CENTROBIND; acc($\mathbf{Z}_i$) or acc(All): accuracy of $\mathbf{Z}_i$ or of concatenated embeddings $(\mathbf{Z}_1, \cdots, \mathbf{Z}_M)$; (rnd): if random backbones are used for $\mathbf{X}_i$-B or CB.

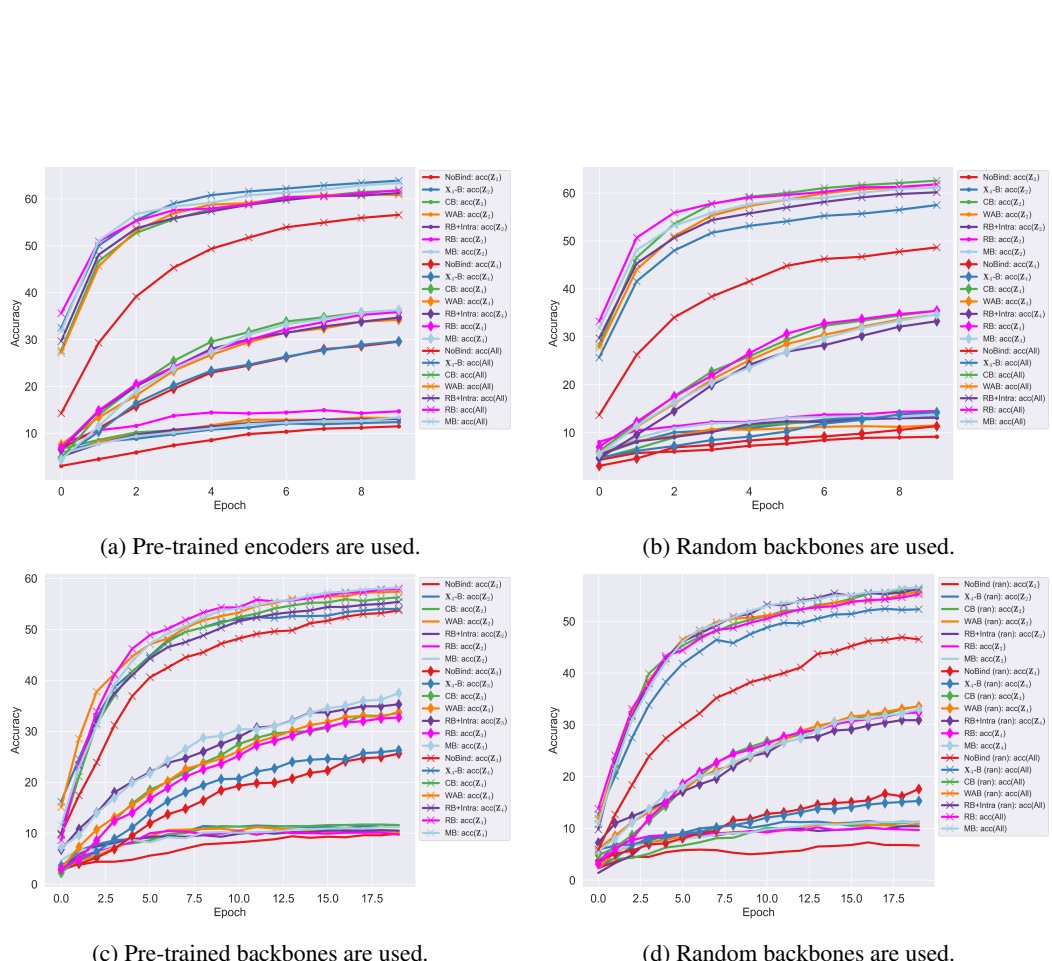

(a) Pre-trained encoders are used.

(b) Random backbones are used.

(c) Pre-trained backbones are used.

(d) Random backbones are used.

Figure 9: Comparison of other dynamic anchor generation methods. (a) and (b): Modal qualities are set to $(0.2, 0.2, 0.2, 1)$. (c) and (d): Modal qualities are set to $(0.2, 0.2, 0.8, 0.8)$. Abbreviation: $\mathbf{X}_i$-B or CB: applying FABIND method to backbones with anchor $\mathbf{X}_i$ or applying CENTROBIND; WAB: weighted average for dynamic anchor with weight identical to the predefined quality for each modality; RB+Intra: randomly choosing a modality for dynamic anchor in every iteration and intra information learning; RB: randomly choosing a modality for dynamic anchor in every iteration; MB: coordinate-wise median for dynamic anchors; acc($\mathbf{Z}_i$) or acc(All): accuracy of $\mathbf{Z}_i$ or of concatenated embeddings $(\mathbf{Z}_1, \cdots, \mathbf{Z}_M)$; (ran): if random backbones are used.

