# OpenReview forum: "Anchors Aweigh! Sail for Optimal Unified Multi-Modal Representations"
_ICLR.cc/2025/Conference — Submitted to ICLR 2025_

### Official Review · Reviewer_ioPy · 2024-10-20

**Soundness:** 2
**Presentation:** 3
**Contribution:** 2
**Rating:** 5
**Confidence:** 3

**Summary:**

This paper proposes CentroBind, which employs a dynamically adjustable centroid-based anchors generated from all available modalities. The experiments are conducted on synthetic and real-world datasets.

**Strengths:**

1. The paper is easy-to-follow.
2. The paper provides sufficient analysis of the proposed method.

**Weaknesses:**

1. CentroBind computes the average of the inputs in the batch as the anchors. However, means are sensitive to outliers. When there are outliers in a batch, the performance will be influenced. Particularly, when the batch size is small, it will also influence the performance.
2. The experiments are not sufficient. There are many unified multimodal representation methods. Why do the authors only compare FABind? More recent baselines should be included.
3. I consider this paper to be incremental work, as it makes only very minor changes to FABind.
4. There are no visualizations in the experiments. Features before and after CentroBind should be visualized to better demonstrate its effectiveness.

**Questions:**

See Weaknesses.

---

> ### Author Response · Authors · 2024-11-23
>
> We thank the reviewer for the comments and suggestions. Please find the following answers to each comments.
>
> > **Weaknesses 1.** CentroBind computes the average of the inputs in the batch as the anchors. However, means are sensitive to outliers. When there are outliers in a batch, the performance will be influenced. Particularly, when the batch size is small, it will also influence the performance.
>
> Thank you for pointing this out. We have also considered this perspective, recognizing that the mean is not robust to outliers. One alternative approach is using the medoid, which is more robust to outliers.
>
> However, despite the limitations of the centroid, we have reasons for continuing to use it:
> 1) The centroid is the minimizer of average L2 distances to the embeddings, making it suitable for the goal of multimodal alignment.
> 2) Identifying outliers in positive pairs is challenging without prior domain knowledge (e.g., data distribution). Otherwise, detecting outliers would require numerous samples for a specific semantic representation, which is impractical.
>
> To address this concern, we conducted additional experiments on synthetic datasets with manually controlled quality (providing prior knowledge on modalities) in Figure 2, 6, 7, and [Figure 8](https://anonymous.4open.science/api/repo/temp-C948/file/Comparison_dynamic_anchor.png?v=4844fb0a). Even in scenarios with very low-quality modalities ([Figure 8](https://anonymous.4open.science/api/repo/temp-C948/file/Comparison_dynamic_anchor.png?v=4844fb0a)) that can be considered outliers, CentroBind performs comparably to the weighted average method that rely on prior knowledge of modality quality, implying robustness of modality imbalance problem.
>
> > **Weaknesses 2.** The experiments are not sufficient. There are many unified multimodal representation methods. Why do the authors only compare FABind? More recent baselines should be included.
>
> We have noted that in both academia and industry, fixed-anchor-bind models such as ImageBind and LanguageBind are the de facto standard used by many researchers and practitioners. Therefore, we focused our analysis and refinements on these approaches. However, we agree that including more baseline comparisons is always helpful, and we have provided additional baseline results incorporating other alignment methods. Please refer to our response to Reviewer YKP7 for further details.
>
> > **Weaknesses 3.** I consider this paper to be incremental work, as it makes only very minor changes to FABind.
>
> In this paper, we first identify the fundamental limitations of alignment methods such as ImageBind and LanguageBind. We generalize these methods under the category of fixed-anchor-bind methods and provide rigorous theoretical analyses of their limitations. Based on these analyses, we propose a simple yet effective modification to the approach, allowing the anchors to be dynamic rather than fixed, which alleviates many of the identified limitations. This also opens the door to further refinements, such as weighted or attention-based averaging. We believe that modeling does not have to be complex; in fact, simplicity is best, especially when grounded in sound rationale and backed by empirical evidence. Therefore, we respectfully disagree with the notion that our contribution is incremental.
>
> > **Weaknesses 4.** There are no visualizations in the experiments. Features before and after CentroBind should be visualized to better demonstrate its effectiveness.
>
> Thank you for your comment. Please refer to our response to Reviewer BBKI for the visualizations.

---

> ### Author Response · Authors · 2024-11-26
> **Additional experiment results regarding weakness 1: means are sensitive to outliers.**
>
> To address weakness 1, we include Table 4 to compare our approach with other methods along with [Figure 8](https://anonymous.4open.science/api/repo/temp-C948/file/Comparison_dynamic_anchor.png?v=4844fb0a) (weighted average, random anchors, and prior knowledge of the given modality).
>
>
>
> ## Table 4
> | **Backbone**     | **Method**              |    [Fig. 8a |        and             |   8b]                                    | [Fig. 8c | and  | 8d]|
> |:------------------:|:-------------------------:|:-------------:|:-------------:|:--------------------------:|:-------------:|:-------------:|:--------------------------:|
> |                              |                                   | **$X_2$**            | **$X_4$** | **$X_1,\cdots,X_4$** | **$X_2$** | **$X_4$** | **$X_1,\cdots,X_4$** |
> | **Pre-trained**  | $\times$               | 0.115       | 0.296       | 0.566                    | 0.099       | 0.256       | 0.537                    |
> |                  | FABind- $X_4$         | 0.124       | 0.297       | **0.639**                | 0.115       | 0.263       | 0.540                    |
> |                  | CentroBind             | 0.131       | **0.363**   | 0.618                    | **0.116**   | 0.336       | 0.563                    |
> |                  | Weighted average       | 0.133       | 0.342       | 0.609                    | 0.102       | 0.338       | 0.574                    |
> |                  | Random + intra learning| 0.131       | 0.347       | 0.613                    | 0.105       | **0.353**   | 0.554                    |
> |                  | Random anchor          | **0.147**   | 0.359       | 0.619                    | 0.097       | 0.327       | **0.579**                |
> | **Random**       | $\times$               | 0.092       | 0.114       | 0.487                    | 0.067       | 0.176       | 0.465                    |
> |                  | FABind- $X_4$         | 0.131       | 0.143       | 0.575                    | 0.112       | 0.153       | 0.523                    |
> |                  | CentroBind             | 0.132       | **0.355**   | **0.626**                | **0.113**   | **0.336**   | **0.559**                |
> |                  | Weighted average       | 0.115       | 0.347       | 0.619                    | 0.109       | **0.336**   | 0.556                    |
> |                  | Random + intra learning| 0.132       | 0.333       | 0.602                    | 0.104       | 0.309       | 0.562                    |
> |                  | Random anchor          | **0.145**   | 0.354       | 0.618                    | 0.097       | 0.324       | 0.552                    |
>
> Table 4 presents the classification accuracy corresponding to the experiments illustrated in [Figure 8](https://anonymous.4open.science/api/repo/temp-C948/file/Comparison_dynamic_anchor.png?v=4844fb0a). In the experiments depicted in Figures 8a and 8b, the information level varies such that $X_1$, $X_2$, and $X_3$ are highly noisy, while $X_4$ is highly informative. Conversely, in Figures 8c and 8d, $X_1$ and $X_2$ are uninformative, whereas $X_3$ and $X_4$ are high-quality modalities. Each modality is selected as the fixed anchor modality in turn, and we compare CentroBind with FABind, weighted average, random anchor, and random anchor with intra-learning methods.
>
> When modality quality is imbalanced, the centroid-based approach may produce suboptimal dynamic anchor constructions, allowing other methods, such as weighted averages, to achieve better results in specific scenarios. Despite this, CentroBind consistently performs either better or on par with weighted average methods, showcasing its robustness to modality imbalance. Furthermore, the random anchor method also achieves competitive performance. These observations suggest that the choice of dynamic anchor generation method may have limited impact on final performance, as long as all encoders are appropriately trained during the process.

---

> ### Author Response · Authors · 2024-11-26
> **Additional experiment results addressing Weaknesses 2**
>
> To address weakness 2, we have included UniBind, AudioCLIP, ViT-Lens as the baseline approaches.
>
> ____
> >## Table 2
> | **Method**  | **Tr, (Ev)**   | **Sar-1** | **Spk-1** | **Spk-3** | **Spk-5** |
> |:-------------:|:----------------:|:-----------:|:-----------:|:-----------:|:-----------:|
> | FABind      | $\mathcal{V}$, ($\mathcal{T}$) | 0.706     | 0.378     | 0.614     | 0.730     |
> | UniBind     |                | 0.544     | 0.170     | 0.328     | 0.478     |
> | AudioCLIP*  |                | 0.501     | 0.096     | 0.258     | 0.388     |
> | ViT-Lens*   |                | 0.506     | 0.097     | 0.343     | 0.449     |
> | CentroBind  |                | **0.716** | **0.474** | **0.736** | **0.836** |
> |-------------|----------------|-----------|-----------|-----------|-----------|
> | FABind      | $\mathcal{A}$, ($\mathcal{T}$) | 0.648     | 0.186     | 0.455     | 0.577     |
> | UniBind     |                | 0.628     | 0.220     | 0.399     | 0.501     |
> | AudioCLIP*  |                | 0.486     | 0.094     | 0.214     | 0.322     |
> | ViT-Lens*   |                | 0.484     | 0.077     | 0.214     | 0.313     |
> | CentroBind  |                | **0.691** | **0.290** | **0.546** | **0.714** |
> |-------------|----------------|-----------|-----------|-----------|-----------|
> | FABind      | $\mathcal{T}$, ($\mathcal{V}$) | 0.572     | 0.243     | 0.445     | 0.630     |
> | UniBind     |                | 0.484     | 0.129     | 0.262     | 0.404     |
> | AudioCLIP*  |                | 0.506     | 0.158     | 0.345     | 0.461     |
> | ViT-Lens*   |                | 0.502     | 0.168     | 0.323     | 0.423     |
> | CentroBind  |                | **0.694** | **0.368** | **0.670** | **0.791** |
> |-------------|----------------|-----------|-----------|-----------|-----------|
> | FABind      | $\mathcal{A}$, ($\mathcal{V}$) | 0.623     | 0.228     | **0.484** | 0.628     |
> | UniBind     |                | 0.567     | 0.199     | 0.367     | 0.514     |
> | AudioCLIP*  |                | 0.503     | 0.209     | 0.384     | 0.496     |
> | ViT-Lens*   |                | 0.500     | 0.149     | 0.332     | 0.451     |
> | CentroBind  |                | **0.683** | **0.243** | 0.475     | **0.632** |
> |-------------|----------------|-----------|-----------|-----------|-----------|
> | FABind      | $\mathcal{V}$, ($\mathcal{A}$) | 0.604     | 0.255     | 0.472     | 0.636     |
> | UniBind     |                | 0.506     | 0.126     | 0.280     | 0.429     |
> | AudioCLIP*  |                | 0.501     | 0.080     | 0.199     | 0.326     |
> | ViT-Lens*   |                | 0.533     | 0.219     | 0.438     | 0.575     |
> | CentroBind  |                | **0.626** | **0.326** | **0.548** | **0.703** |
> |-------------|----------------|-----------|-----------|-----------|-----------|
> | FABind      | $\mathcal{T}$, ($\mathcal{A}$) | 0.534     | 0.241     | 0.509     | 0.635     |
> | UniBind     |                | 0.514     | 0.091     | 0.248     | 0.365     |
> | AudioCLIP*  |                | 0.477     | 0.088     | 0.309     | 0.439     |
> | ViT-Lens*   |                | 0.475     | 0.070     | 0.214     | 0.329     |
> | CentroBind  |                | **0.655** | **0.346** | **0.610** | **0.741** |
>
>
> Table 2 presents the classification accuracy for zero-shot cross-modal sarcasm detection and speaker classification tasks. CentroBind achieves significantly outperform top-1 accuracy compared to all baseline methods in both tasks. Our additional baselines, UniBind, AudioCLIP, and ViT-Lens, perform poorly in the zero-shot cross-modal experiment.

---

> > ### Author Response · Authors · 2024-11-26
> > **Additional visualization update regarding weaknesses 4. There are no visualizations in the experiments.**
> >
> > To address weakness 4, we have attached the t-SNE and UMAP visualizations of the learned feature spaces of the synthetic dataset for FABind and CentroBind [here](https://anonymous.4open.science/api/repo/temp-C948/file/tsne_umap.png). We have included the plots in the revised manuscript.

---

> > > ### Comment · Reviewer_ioPy · 2024-11-26
> > >
> > > Thank you for your response and the additional experiments. The authors mention that using the median is more robust; however, I would like to ask why the results of "MedianBind" are not reported. Does the median yield better results than the mean on common datasets? Additionally, the authors utilized a batch size of 16, which is relatively small. In the context of a large dataset, could the mean of each batch vary significantly, potentially leading to instable training? Addressing these points would be valuable in illustrating why using the centroid is preferable to other metrics.

---

> ### Author Response · Authors · 2024-11-27
> **Answers to the questions regarding median and batch size**
>
> - We sincerely thank the reviewer for their comments on the use of medians and stability in training. We have updated [Figure 8]((https://anonymous.4open.science/r/temp-C948/Comparison_dynamic_anchor.png)) and Table 4 to include results for a median-based dynamic anchor approach.
> - Based on valuable feedback from the reviewers, we now explore a variety of dynamic anchor computation methods, including weighted averaging, random choice (with and without intra-learning), as presented in Table 4. We specifically thank the reviewer for suggesting the inclusion of the median-based approach, making the paper more comprehensive.
> - To reflect the diversity of anchor computation methods, we have revised the manuscript to **explicitly acknowledge alternative strategies**. Additionally, we provide a clear rationale for selecting the centroid-based approach before introducing it as our primary method.
> - Finally, we reiterate that a key contribution of our work lies in **the theoretical analysis of fixed anchor limitations and their replacement with dynamic anchors**, facilitating intra- and shared information learning across modalities. The median-based approach can be an extension of this analysis, and we thank the reviewer for highlighting this point.
>
> ### Discussion on the Median-Based Results
> - The median-based anchor was computed coordinate-wise. Specifically, for a median-based anchor, we compute the $j$th coordinate of the $i$th anchor as $a_{i,j} = {\rm Median}(z_{1,i,j}, z_{2,i,j}, \cdots, z_{M,i,j})$, where ${z}_{m,i,j}$ denotes the $j$th coordinate of the embedding for the $i$th sample in modality $m$.
> - Comparative results of the median-based method with other approaches are presented in Table 4. For these experiments, we intentionally introduced extreme outliers, as detailed in Appendix C.
> - As expected, the median approach occasionally outperforms other methods in such extreme cases. However, the differences are not significant, and CentroBind often achieves better overall performance.
> - In this paper, we chose to focus on centroid-based anchors, as the centroid represents the geometric center, which aligns closely with the objective of this paper: achieving effective multimodal alignment in embedding spaces.
> - Furthermore, we think that outliers are uncommon in embedding spaces when the encoders are well-trained. We include an extended discussion of this topic in Appendix C.

---

> ### Author Response · Authors · 2024-11-27
> **Discussion on the median-based dynamic anchors (cont.)**
>
> #### Table 4
> | **Backbone**     | **Method**              |    [Fig. 8a |        and             |   8b]                                    | [Fig. 8c | and  | 8d]|
> |:------------------:|:-------------------------:|:-------------:|:-------------:|:--------------------------:|:-------------:|:-------------:|:--------------------------:|
> |                              |                                   | $X_2$            | $X_4$ | $X_1$, $\cdots$, $X_4$ | $X_2$ | $X_4$ | $X_1$, $\cdots$, $X_4$ |
> | **Pre-trained**  | $\times$               | 0.115       | 0.296       | 0.566                    | 0.099       | 0.256       | 0.537                    |
> |                  | FABind- $X_4$         | 0.124       | 0.297       | **0.639**                | 0.115       | 0.263       | 0.540                    |
> |                  | CentroBind             | 0.131       | **0.363**   | 0.618                    | **0.116**   | 0.336       | 0.563                    |
> |                  | Weighted average       | 0.133       | 0.342       | 0.609                    | 0.102       | 0.338       | 0.574                    |
> |                  | Random + intra learning| 0.131       | 0.347       | 0.613                    | 0.105       | 0.353   | 0.554                    |
> |                  | Random anchor          | **0.147**   | 0.359       | 0.619                    | 0.097       | 0.327       | 0.579                |
> |                  |Median          | 0.134   | **0.363**       | 0.634                    | 0.112       | **0.375**       | **0.582**                |
> | **Random**       | $\times$               | 0.092       | 0.114       | 0.487                    | 0.067       | 0.176       | 0.465                    |
> |                  | FABind- $X_4$         | 0.131       | 0.143       | 0.575                    | 0.112       | 0.153       | 0.523                    |
> |                  | CentroBind             | 0.132       | **0.355**   | **0.626**                | **0.113**   | **0.336**   | 0.559                |
> |                  | Weighted average       | 0.115       | 0.347       | 0.619                    | 0.109       | **0.336**   | 0.556                    |
> |                  | Random + intra learning| 0.132       | 0.333       | 0.602                    | 0.104       | 0.309       | 0.562                    |
> |                  | Random anchor          | **0.145**   | 0.354       | 0.618                    | 0.097       | 0.324       | 0.552                    |
> |                  |Median          | 0.137   | 0.347       | 0.612                    | 0.112       | 0.330       | **0.565**                |

---

> > ### Author Response · Authors · 2024-11-27
> > **Stability in training with small batch size**
> >
> > - We would like to clarify that the centroid dynamic anchor is derived from modalities, not from samples within a batch. Specifically, for a scenario with 4 modalities, the $i$th anchor is computed as the centroid of 4 embeddings, each corresponding to the $i$th sample from a different modality. This process is entirely independent of the batch size.
> > - As such, we believe that the batch size does not significantly influence the behavior of the centroid dynamic anchor. Instead, the batch size primarily affects the number of negative samples in the InfoNCE loss, which is outside the scope of our study.
> > - Given limited computational resources (esp. when dealing with video modal samples), we conducted our experiments using a batch size of 16. Despite this constraint, we ensured that FABind, UniBind, and CentroBind were trained fairly under identical parameter settings, including the batch size. We hope this explanation addresses the reviewer’s concern.

---

> ### Author Response · Authors · 2024-12-02
> **Thank you for your review!**
>
> Dear Reviewer ioPy,
>
> Thank you for your effort in reviewing our work. We hope you have had the chance to review our responses and new updates on experiments. As the discussion period is drawing to a close, we would greatly appreciate it if you could confirm whether our updates have fully addressed your concerns.
>
> Thank you again for your time and thoughtful review.
>
> Best regards,
> The Authors

---

### Official Review · Reviewer_NCrk · 2024-10-28

**Soundness:** 3
**Presentation:** 4
**Contribution:** 4
**Rating:** 8
**Confidence:** 3

**Summary:**

The paper addressed the issues related to fixing a particular modality as an anchor in multi-modal representations. They argue that the anchor should not be fixed since the semantic relationships among the non-anchors may be lost and also since intra-modality information  may not be captured if the anchor is fixed. Instead, they propose choosing centroids of positive augmentation pairs among multiple modalities. The paper presents theoretical analyses demonstrating the drawbacks of fixed anchors and the advantages of dynamic anchors. Experimental results show the superiority of the proposed strategy on a synthetic dataset and a multimodal dataset for cross-modal retrieval and, sarcasm and speaker classification.

**Strengths:**

1. A strong theoretical paper that shows that fixing an image or text as an anchor for multimodal representation loses information contained in other modalities that are not selected as anchors.
2. The proposed strategy of using dynamic anchors is also supported theoretically by deriving a lower bound on CentroBind's objective function and minimizing the objective function.
3. Very good interpretations based on theoretical analysis of why FABind is insufficient and how CentroBind overcomes the former's limitations.

**Weaknesses:**

There are no major weaknesses to speak of.
1. To me, the claim of a closer representation to Huh et al's "Platonic representation" is not  quite evident. I tend to believe the authors are also not very positive about this aspect from their statement in line 361 that the representation is "likely closer" to the Platonic representation.
2. This is not a weakness of the paper per se. Audio, video and text are the usual modalities that are discussed in multimodal representation. It would be interesting to study datasets with additional modalities such as VIDIMU: Multimodal video and IMU kinematic dataset on daily life activities using affordable devices (https://zenodo.org/record/8210563), which has inertial sensors.

**Questions:**

1. Line 220: The authors may clarify why equation (4) shows that FABind does not guarantee shared information fbetwen non-anchor modalities.
2. Are there any constraints on the batch B from which the anchor embeddings are derived?
3. It is not clear which part of the analysis specifically shows that using a fixed anchor affects the representativeness of all modalities (P4).
4. How would acc(All) perform in Fig 2 (a)?

---

> ### Author Response · Authors · 2024-11-23
>
> We would like to thank the reviewer for their time and effort in reviewing our paper and valuable comments. Please find the following answers to the comments and questions.
>
> > **Weaknesses 1.** To me, the claim of a closer representation to Huh et al's "Platonic representation" is not quite evident. I tend to believe the authors are also not very positive about this aspect from their statement in line 361 that the representation is "likely closer" to the Platonic representation.
>
> Huh et al.’s “Platonic Representation” hypothesis posits that “neural networks, trained with different objectives on different data and modalities, are converging to a shared statistical model of reality in their representation spaces.” While it is difficult to prove this directly, we believe that dynamic computation of anchors based on sample-wise mutual information is more likely to achieve such a representation than fixed anchor-based methods. This is because, CentroBind, by utilizing dynamic anchors, is likely to retain a more comprehensive representation of all modalities.
>
> > **Weaknesses 2.** This is not a weakness of the paper per se. Audio, video and text are the usual modalities that are discussed in multimodal representation. It would be interesting to study datasets with additional modalities such as VIDIMU: Multimodal video and IMU kinematic dataset on daily life activities using affordable devices (https://zenodo.org/record/8210563), which has inertial sensors.
>
> Thank you for suggesting this dataset. Incorporating different modalities, such as kinematics, would indeed be an interesting avenue for future research.
>
>
> > **Question 1.** Line 220: The authors may clarify why equation (4) shows that FABind does not guarantee shared information between non-anchor modalities.
>
> Solving Equation (4) requires maximizing the mutual information between the anchor modality and other modalities, while the mutual information between non-anchored modalities is not explicitly considered in the objective function. Consequently, there is no guarantee that mutual information among non-anchor modalities will increase. In other words, the solution to Equation (4) only needs to maximize the shared information between the anchor and the other modalities.
>
> > **Question 2.** Are there any constraints on the batch B from which the anchor embeddings are derived?
>
> Similar to any other fixed-anchor-bind method, CentroBind also needs each sample in a batch to have paired representation (embeddings) among different modalities. However, it does not need to be fully paired across all modalities.
>
> > **Question 3.** It is not clear which part of the analysis specifically shows that using a fixed anchor affects the representativeness of all modalities (P4).
>
> Thank you for pointing out this potentially confusing part. Our intention in claiming P4 is to emphasize that a single modality cannot capture the unique information provided by other modalities. To be representative of all modalities, one must leverage all modalities collectively, which is precisely the motivation behind multimodal learning. Otherwise, unique information present in other modalities may be overlooked. We have updated the paper to include this reasoning on page 5.
>
> > **Question 4.** How would acc(All) perform in Fig 2 (a)?
>
> acc(All) refers to the classification accuracy achieved when embeddings from all modalities are utilized. Specifically, this involves training and evaluating a decoder using the concatenated embeddings of all modalities as the input to the decoder.

---

> > ### Comment · Reviewer_NCrk · 2024-11-26
> >
> > Thanks to the authors for their response.
> > With reference to Question 4, I do understand that acc(all) uses embeddings from all modalities. I was wondering where it would be placed in the graph of Fig 2(a).
> > The visualizations are helpful in contrasting the embeddings of Fabind and Centrobind.

---

> > > ### Author Response · Authors · 2024-11-27
> > > **Additional visualization**
> > >
> > > We thank the reviewer for their clarification and follow-up comment. Regarding the visualization of acc(All) performance in Figure 2(a), please refer to this [figure](https://anonymous.4open.science/r/temp-C948/fig_2_with_all.png). The same figure has also been included as Figure 4 in Appendix C.

---

> > > > ### Author Response · Authors · 2024-12-02
> > > > **Thanks for your review!**
> > > >
> > > > Dear Reviewer NCrk,
> > > >
> > > > Thank you very much for your positive feedback on our work. We truly appreciate your time and thoughtful review.
> > > >
> > > > Best regards,
> > > > The Authors

---

### Official Review · Reviewer_aFcb · 2024-11-01

**Soundness:** 3
**Presentation:** 3
**Contribution:** 3
**Rating:** 6
**Confidence:** 4

**Summary:**

**Summary of this paper:**

This work introduces a new task: CentroBind, a novel approach in multimodal learning that addresses limitations found in fixed-anchor binding methods(FABIND). FABIND typically uses a single anchor modality to align representations across various modalities. However, this method has critical drawbacks, such as dependence on the anchor choice and lack of intra- and inter-modal information sharing. CentroBind proposes to replace fixed anchors with dynamic centroid-based anchors derived from all available modalities, creating a more balanced representation space. Theoretical analysis and experiments on synthetic and real-world datasets demonstrate that CentroBind enhances multimodal alignment and performs better in tasks requiring nuanced multimodal interactions.

**Strengths:**

The paper is well-organized and written. The paper proposes an innovative solution to the limitations inherent in fixed-anchor binding methods by introducing CentroBind, which leverages centroid-based dynamic anchors. CentroBind's design is rooted in solid theoretical analysis, addressing crucial aspects of multimodal learning, including intra-modal, inter-modal, and multimodal alignment. The paper conducted extensive experiments on both synthetic and real-world datasets, and the results confirmed that CentroBind outperformed FABIND. The research content is innovative and holds potential application value.

**Weakness:**

As for method. The theoretical advantages of dynamic centroid-based anchors are well-articulated, the practical implementation details are somewhat limited.As for the experimental results. (1) I believe that the comparative experiments and ablation studies are not sufficiently. The paper only compares CentroBind and FABIND methods, lacking additional comparisons with other recent multi-modal alignment frameworks, which leads me to question the validity of this comparison. (2) There could be more exploration into how CentroBind performs under different data distributions and noise levels, which could provide further insights into its robustness.

**Weakness Summary.**

CentroBind's scalability and practicality for large datasets require further consideration. In addition, broadening experimental comparisons and analysis would help address potential limitations and position the method within the broader multimodal research.

**Strengths:**

Refer to the summary

**Weaknesses:**

Refer to the summary

**Questions:**

Refer to the summary

---

> ### Author Response · Authors · 2024-11-23
>
> We thank the reviewer for their time and effort in reviewing our paper and comments. We answer each weaknesses as follows.
>
> > **Weaknesses 1.**  I believe that the comparative experiments and ablation studies are not sufficiently. The paper only compares CentroBind and FABIND methods, lacking additional comparisons with other recent multi-modal alignment frameworks, which leads me to question the validity of this comparison.
>
> We would like to point out that FABind methods are among the most widely used approaches and represent the current state-of-the-art in multi-modal representation learning. As an additional experiment, we are working to include the most recent approach, UniBind, as an additional baseline.
>
> > **Weaknesses 2.** There could be more exploration into how CentroBind performs under different data distributions and noise levels, which could provide further insights into its robustness.
>
> Thank you for your comment. We have included additional experiments where we vary the information density of different modalities to the extreme and compare three settings: weighted averaging of each modality, random anchors, and equal averaging (i.e., vanilla CentroBind). Please refer to our response to Reviewer 2XLP for more details.
>
> > **Weakness summary.**  CentroBind's scalability and practicality for large datasets require further consideration. In addition, broadening experimental comparisons and analysis would help address potential limitations and position the method within the broader multimodal research.
>
> Our set of experiments includes:
> 1. Experiments involving multiple modalities with varying levels of quality and informativeness (Figure 2)
> 2.  Experiments involving many more modalities (Figures 6 and 7)
> 3.  Additional experiments using more extreme cases of modality qualities to test the limits of the approach ([Figure 8](https://anonymous.4open.science/api/repo/temp-C948/file/Comparison_dynamic_anchor.png?v=4844fb0a))
> 4.  Experiments on real-world benchmark datasets and subsequent downstream task evaluations (Table 2)
>
> Through these experiments, we believe we have sufficiently demonstrated the scalability and practicality of our approach across various settings and scenarios. We are also working on adding another baseline result using the UniBind [1], which is one of the latest methods with an available codebase, and AudioCLIP [2].
>
> [1] Lyu et al., UniBind: LLM-Augmented Unified and Balanced Representation Space to Bind Them All
>
> [2] Guzhov at el., Audioclip: Extending clip to image, text and audio.

---

> > ### Author Response · Authors · 2024-11-26
> > **Results regarding Weaknesses 1 insufficient experiments. [1/2]**
> >
> > We have conducted additional experimental results with other baseline methods, such as AudioCLIP, UniBind, and ViT-Lens.
> >
> > ____
> > >## Table 2
> > | **Method**  | **Tr, (Ev)**   | **Sar-1** | **Spk-1** | **Spk-3** | **Spk-5** |
> > |:-------------:|:----------------:|:-----------:|:-----------:|:-----------:|:-----------:|
> > | FABind      | $\mathcal{V}$, ($\mathcal{T}$) | 0.706     | 0.378     | 0.614     | 0.730     |
> > | UniBind     |                | 0.544     | 0.170     | 0.328     | 0.478     |
> > | AudioCLIP*  |                | 0.501     | 0.096     | 0.258     | 0.388     |
> > | ViT-Lens*   |                | 0.506     | 0.097     | 0.343     | 0.449     |
> > | CentroBind  |                | **0.716** | **0.474** | **0.736** | **0.836** |
> > |-------------|----------------|-----------|-----------|-----------|-----------|
> > | FABind      | $\mathcal{A}$, ($\mathcal{T}$) | 0.648     | 0.186     | 0.455     | 0.577     |
> > | UniBind     |                | 0.628     | 0.220     | 0.399     | 0.501     |
> > | AudioCLIP*  |                | 0.486     | 0.094     | 0.214     | 0.322     |
> > | ViT-Lens*   |                | 0.484     | 0.077     | 0.214     | 0.313     |
> > | CentroBind  |                | **0.691** | **0.290** | **0.546** | **0.714** |
> > |-------------|----------------|-----------|-----------|-----------|-----------|
> > | FABind      | $\mathcal{T}$, ($\mathcal{V}$) | 0.572     | 0.243     | 0.445     | 0.630     |
> > | UniBind     |                | 0.484     | 0.129     | 0.262     | 0.404     |
> > | AudioCLIP*  |                | 0.506     | 0.158     | 0.345     | 0.461     |
> > | ViT-Lens*   |                | 0.502     | 0.168     | 0.323     | 0.423     |
> > | CentroBind  |                | **0.694** | **0.368** | **0.670** | **0.791** |
> > |-------------|----------------|-----------|-----------|-----------|-----------|
> > | FABind      | $\mathcal{A}$, ($\mathcal{V}$) | 0.623     | 0.228     | **0.484** | 0.628     |
> > | UniBind     |                | 0.567     | 0.199     | 0.367     | 0.514     |
> > | AudioCLIP*  |                | 0.503     | 0.209     | 0.384     | 0.496     |
> > | ViT-Lens*   |                | 0.500     | 0.149     | 0.332     | 0.451     |
> > | CentroBind  |                | **0.683** | **0.243** | 0.475     | **0.632** |
> > |-------------|----------------|-----------|-----------|-----------|-----------|
> > | FABind      | $\mathcal{V}$, ($\mathcal{A}$) | 0.604     | 0.255     | 0.472     | 0.636     |
> > | UniBind     |                | 0.506     | 0.126     | 0.280     | 0.429     |
> > | AudioCLIP*  |                | 0.501     | 0.080     | 0.199     | 0.326     |
> > | ViT-Lens*   |                | 0.533     | 0.219     | 0.438     | 0.575     |
> > | CentroBind  |                | **0.626** | **0.326** | **0.548** | **0.703** |
> > |-------------|----------------|-----------|-----------|-----------|-----------|
> > | FABind      | $\mathcal{T}$, ($\mathcal{A}$) | 0.534     | 0.241     | 0.509     | 0.635     |
> > | UniBind     |                | 0.514     | 0.091     | 0.248     | 0.365     |
> > | AudioCLIP*  |                | 0.477     | 0.088     | 0.309     | 0.439     |
> > | ViT-Lens*   |                | 0.475     | 0.070     | 0.214     | 0.329     |
> > | CentroBind  |                | **0.655** | **0.346** | **0.610** | **0.741** |
> >
> >
> > Table 2 presents the classification accuracy for zero-shot cross-modal sarcasm detection and speaker classification tasks. CentroBIND achieves significantly outperform top-1 accuracy compared to all baseline methods in both tasks. Our additional baselines, UniBind, AudioCLIP, and ViT-Lens, perform poorly in the zero-shot cross-modal experiment.

---

> > > ### Author Response · Authors · 2024-11-26
> > > **Results regarding Weaknesses 1 insufficient experiments. [2/2]**
> > >
> > > ### Table 5
> > > | **Method**       | **Modality**                          | **Sar-1**  | **Spk-1**  | **Spk-3**  | **Spk-5**  |
> > > |:------------------:|:----------------------------------:|:------------:|:------------:|:------------:|:------------:|
> > > | FABind           | $\mathcal{T}$                                | 0.606      | 0.219      | 0.458      | 0.632      |
> > > | UniBind          |                                       | 0.600      | 0.214      | 0.412      | 0.569      |
> > > | AudioCLIP*       |                                       | 0.488      | 0.155      | 0.280      | 0.388      |
> > > | ViT-Lens*        |                                       | 0.543      | 0.172      | 0.342      | 0.472      |
> > > | CentroBind       |                                       | **0.667**  | **0.287**  | **0.507**  | **0.642**  |
> > > |----------------|-------------------------------|------------|------------|------------|------------|
> > > | FABind           | $\mathcal{V}$                                | 0.668      | 0.375      | 0.587      | 0.691      |
> > > | UniBind          |                                       | 0.658      | 0.381      | 0.641      | 0.770      |
> > > | AudioCLIP*       |                                       | 0.504      | 0.110      | 0.275      | 0.414      |
> > > | ViT-Lens*        |                                       | **0.697**  | **0.586**  | **0.738**  | **0.797**  |
> > > | CentroBind       |                                       | 0.670      | 0.380      | 0.609      | 0.726      |
> > > |----------------|-------------------------------|------------|------------|------------|------------|
> > > | FABind           | $\mathcal{A}$                                | 0.639      | 0.201      | 0.457      | 0.599      |
> > > | UniBind          |                                       | 0.633      | 0.272      | 0.528      | 0.691      |
> > > | AudioCLIP*       |                                       | 0.525      | 0.158      | 0.343      | 0.454      |
> > > | ViT-Lens*        |                                       | **0.686**  | **0.396**  | **0.664**  | **0.8**    |
> > > | CentroBind       |                                       | 0.616      | 0.234      | 0.461      | 0.609      |
> > > |----------------|-------------------------------|------------|------------|------------|------------|
> > > | FactorCL*        | $\substack{\mathcal{V}, \mathcal{A}, \mathcal{T} \\ (\mathcal{V}, \mathcal{A}, \mathcal{T})}$ | 0.699      | -          | -          | -          |
> > > | SimMMDG*         |                                       | 0.725      | -          | -          | -          |
> > > | FABind           |                                       | 0.678      | 0.343      | 0.554      | 0.677      |
> > > | UniBind          |                                       | 0.646      | 0.383      | 0.622      | 0.764      |
> > > | AudioCLIP*       |                                       | 0.530      | 0.119      | 0.261      | 0.378      |
> > > | ViT-Lens*        |                                       | **0.731**  | **0.506**  | **0.736**  | **0.812**  |
> > > | CentroBind       |                                       | 0.704      | 0.346      | 0.594      | 0.733      |
> > >
> > > Models with asterisk are evaluated in different experimental settings (encoder backbone, pretraining datasets).
> > >
> > > Table 5 summarizes the classification accuracy of FABind, UniBind, and CentroBind for both single-modal and multimodal settings, using embeddings extracted by the binding methods and a simple decoder for classification. CentroBind generally outperforms other methods in sarcasm detection, while UniBind shows better performance in speaker classification, likely due to its use of LLM-augmented descriptions as additional modality. Notably, CentroBind consistently surpasses FABind. Additionally, CentroBind can incorporate LLM-augmented descriptions as an extra modality, which could further enhance its performance.
> > >
> > > For reference, the table also includes sarcasm detection results from FactorCL, SimMMDG, AudioCLIP, and ViT-Lens. While ViT-Lens achieves higher accuracy due to larger backbone encoders and extensive pretraining, it can be considered a variant of FABind. Applying CentroBind's dynamic anchor method to ViT-Lens or integrating Vision Transformers into CentroBind could further boost performance.

---

> ### Author Response · Authors · 2024-11-26
> **Results regarding Weaknesses 2 different data distributions.**
>
> We have conducted experiments on synthetic dataset generated according to certain data distribution to address Weaknesses 2.
>
>
> ## Table 4
> | **Backbone**     | **Method**              |    [Fig. 8a |        and             |   8b]                                    | [Fig. 8c | and  | 8d]|
> |:------------------:|:-------------------------:|:-------------:|:-------------:|:--------------------------:|:-------------:|:-------------:|:--------------------------:|
> |                              |                                   | **$X_2$**            | **$X_4$** | **$X_1,\cdots,X_4$** | **$X_2$** | **$X_4$** | **$X_1,\cdots,X_4$** |
> | **Pre-trained**  | $\times$               | 0.115       | 0.296       | 0.566                    | 0.099       | 0.256       | 0.537                    |
> |                  | FABind- $X_4$         | 0.124       | 0.297       | **0.639**                | 0.115       | 0.263       | 0.540                    |
> |                  | CentroBind             | 0.131       | **0.363**   | 0.618                    | **0.116**   | 0.336       | 0.563                    |
> |                  | Weighted average       | 0.133       | 0.342       | 0.609                    | 0.102       | 0.338       | 0.574                    |
> |                  | Random + intra learning| 0.131       | 0.347       | 0.613                    | 0.105       | **0.353**   | 0.554                    |
> |                  | Random anchor          | **0.147**   | 0.359       | 0.619                    | 0.097       | 0.327       | **0.579**                |
> | **Random**       | $\times$               | 0.092       | 0.114       | 0.487                    | 0.067       | 0.176       | 0.465                    |
> |                  | FABind- $X_4$         | 0.131       | 0.143       | 0.575                    | 0.112       | 0.153       | 0.523                    |
> |                  | CentroBind             | 0.132       | **0.355**   | **0.626**                | **0.113**   | **0.336**   | **0.559**                |
> |                  | Weighted average       | 0.115       | 0.347       | 0.619                    | 0.109       | **0.336**   | 0.556                    |
> |                  | Random + intra learning| 0.132       | 0.333       | 0.602                    | 0.104       | 0.309       | 0.562                    |
> |                  | Random anchor          | **0.145**   | 0.354       | 0.618                    | 0.097       | 0.324       | 0.552                    |
>
>
>
> Table 4 presents the classification accuracy corresponding to the experiments illustrated in [Figure 8](https://anonymous.4open.science/api/repo/temp-C948/file/Comparison_dynamic_anchor.png?v=4844fb0a). In the experiments depicted in Figures 8a and 8b, the information level varies such that $X_1$, $X_2$, and $X_3$ are highly noisy, while $X_4$ is highly informative. Conversely, in Figures 8c and 8d, $X_1$ and $X_2$ are uninformative, whereas $X_3$ and $X_4$ are high-quality modalities. Each modality is selected as the fixed anchor modality in turn, and we compare CentroBind with FABind, weighted average, random anchor, and random anchor with intra-learning methods.
>
> When modality quality is imbalanced, the centroid-based approach may produce suboptimal dynamic anchor constructions, allowing other methods, such as weighted averages, to achieve better results in specific scenarios. Despite this, CentroBind consistently performs either better or on par with weighted average methods, showcasing its robustness to modality imbalance. Furthermore, the random anchor method also achieves competitive performance. These observations suggest that the choice of dynamic anchor generation method may have limited impact on final performance, as long as all encoders are appropriately trained during the process.

---

### Official Review · Reviewer_YKP7 · 2024-11-01

**Soundness:** 2
**Presentation:** 3
**Contribution:** 2
**Rating:** 3
**Confidence:** 4

**Summary:**

The paper removes the need for selecting a fixed anchor modality in Fixed-Anchor-Bind (FABIND) and uses the centroid of all modality representations as an anchor representation. The paper study their method theoretically, showing that it captures intra- and inter information and shows empirical improvements over various synthetic and real-world datasets.

**Strengths:**

The paper addresses challenges with Fixed-Anchor Binding (FABIND) and introduces a centroid-based anchor approach as an effective alternative. This method, Centro BIND, demonstrates superior performance over FABIND across both synthetic and real-world datasets in retrieval and classification tasks, which is likely to capture the interest of the community. Additionally, the paper is concise, well-written, and easy to follow.

**Weaknesses:**

The paper addresses challenges with Fixed-Anchor Binding (FABIND) and introduces a centroid-based anchor approach as an effective alternative. This method, Centro BIND, demonstrates superior performance over FABIND across both synthetic and real-world datasets in retrieval and classification tasks, which is likely to capture the interest of the community. Additionally, the paper is concise, well-written, and easy to follow.

* **Limited Baseline Comparison**: While the limitations of fixed anchors are well-articulated, additional baselines would help to further justify the centroid-based approach. For example, sampling a modality as an anchor randomly or choosing an anchor based on the minimum or maximum similarity score could provide useful comparisons to strengthen the motivation for centroid-based anchoring.

* **Positioning Among Related Works**: Although the paper’s primary focus is to demonstrate improvement over FABIND, it would benefit from a broader positioning among recent multimodal works [1, 2, 3]. Several studies have explored inter- and intra-modality information; a comparative discussion here would contextualize the contributions more effectively. Including a simple baseline of early or late fusion across modalities for both synthetic and real-world datasets would be insightful. Additionally, it’s unclear why Centro BIND does not utilize all modalities in Table 2 for a more comprehensive comparison.

* **Comparison with Individual Modalities**: Since the work emphasizes intra-modality information, a comparison with individual modality baselines and an ensemble of modalities, especially in Table 2 for evaluation, is essential. This would underscore how effectively Centro BIND leverages intra-modality information compared to single-modality models.

* **Clarification on FABIND Setup**: Could the authors clarify if FABIND uses pre-trained encoders in the experiments, or are the encoders randomly initialized? This detail would help to understand the experimental setup more clearly.

[1] Du et al., On Uni-Modal Feature Learning in Supervised Multi-Modal Learning.
[2] Madaan et al., A Framework for Multi-modal Learning: Jointly Modeling Inter- & Intra-Modality Dependencies.
[3] Zhang et al., Multimodal Representation Learning by Alternating Unimodal Adaptation

**Questions:**

Please refer to the Weaknesses section.

---

> ### Author Response · Authors · 2024-11-23
>
> We would like to thank the reviewer for their comments and suggestions. Please find the following responses to the comments.
>
>
> > **Limited Baseline Comparison.** While the limitations of fixed anchors are well-articulated, additional baselines would help to further justify the centroid-based approach. For example, sampling a modality as an anchor randomly or choosing an anchor based on the minimum or maximum similarity score could provide useful comparisons to strengthen the motivation for centroid-based anchoring.
>
> Thank you for your comment on the different dynamic anchor generation methods. In the revised version, we have compared various baseline methods for dynamic anchors, such as the weighted average (which requires additional knowledge about the datasets) and random anchors. We briefly include these details in Section 4.1 and provide a more thorough explanation in Appendix C.1. The results indicate that, despite not requiring additional information (e.g., quality or distribution) about the dataset and downstream tasks, CentroBind performs comparably to the weighted average, which we consider a "genie case" assuming prior knowledge of data distributions for each modality.
>
> We also refer to our responses to Reviewer 2XLP for further clarification.
>
> > **Positioning Among Related Works.** Although the paper’s primary focus is to demonstrate improvement over FABIND, it would benefit from a broader positioning among recent multimodal works [1, 2, 3]. Several studies have explored inter- and intra-modality information; a comparative discussion here would contextualize the contributions more effectively. Including a simple baseline of early or late fusion across modalities for both synthetic and real-world datasets would be insightful. Additionally, it’s unclear why Centro BIND does not utilize all modalities in Table 2 for a more comprehensive comparison.
>
> Thank you for the suggestion. We have revised the related work section to include the suggested references, as well as additional relevant works detailed in Appendix A. Specifically, we have expanded the discussion to introduce literature on multimodal learning, including supervised multimodal learning and self-supervised learning, extending beyond the previous focus on multimodal alignment perspectives.
>
> > **Comparison with Individual Modalities.** Since the work emphasizes intra-modality information, a comparison with individual modality baselines and an ensemble of modalities, especially in Table 2 for evaluation, is essential. This would underscore how effectively Centro BIND leverages intra-modality information compared to single-modality models.
>
> Thank you for your comment on the experiment. Table 2 presents the cross-modal zero-shot classification accuracy, which is a standard task for evaluating unified representation spaces. Since the primary objective of our work is to build a unified space for multiple modalities, we focused on comparing CentroBind with the state-of-the-art method FABind, which is specifically designed for multimodal learning in unified spaces. Therefore, we believe that adding unimodal cases to Table 2 would diverge from its intended purpose. Instead, we have included a comparison with unimodal learning in Figure 2, represented by acc($Z_1$). We have analyzed this synthetic experiment in terms of intra-modal information in Section 4.1.
>
> > **Clarification on FABIND Setup.** Could the authors clarify if FABIND uses pre-trained encoders in the experiments, or are the encoders randomly initialized? This detail would help to understand the experimental setup more clearly.
>
> We utilized pre-trained encoders for the real-world dataset experiments, as noted in the first paragraph of Section 4.2. For the synthetic experiments, we conducted extensive comparisons between randomly initialized encoders and pre-trained encoders, as shown in Figures 2, 6, 7, and 8.

---

> > ### Comment · Reviewer_YKP7 · 2024-11-24
> > **Thank you for your response**
> >
> > Thank you for your response and for including additional experiments. I appreciate the effort, but several concerns remain unaddressed. Below are my detailed comments:
> >
> > ---
> >
> > > Limited Baseline Comparison:
> >
> > Thank you for including the results. However, the current presentation is challenging to interpret, as the plot combines $Acc(z_2)$, $Acc(z_1)$, and $Acc(all)$ into a single visualization, which makes the results less clear. I recommend separating these for better readability. Additionally, I suggest extending the comparisons to datasets beyond synthetic ones to demonstrate the scalability and applicability of the approach in real-world scenarios.
> >
> > ---
> >
> > > Comparison with Related Work:
> >
> > I appreciate the inclusion of a discussion on related works. However, as mentioned in my initial feedback, incorporating a straightforward baseline (e.g., early or late fusion across modalities) would provide more meaningful insights. This should be applied to both synthetic and real-world datasets. Furthermore, it is unclear why Centro BIND does not leverage all modalities in Table 2 for a more comprehensive comparison. Including experimental results against the compared methods in related work will further strengthen the paper's positioning for future works.
> >
> > ---
> >
> > > Comparison with Individual Modalities:
> >
> > I understand the primary objective is to develop unified representations. However, my request for experiments with individual modalities was motivated by the challenge of over-reliance on unimodal biases, which could undermine the efficacy of the proposed approach. Running these experiments would serve as a useful sanity check to ensure that the model does not depend disproportionately on single modalities and could be valuable for future references.
> >
> > ---
> >
> > In summary, while the added experiments are appreciated, addressing the above concerns would improve the clarity, scalability, and comprehensiveness of the work.

---

> > > ### Author Response · Authors · 2024-11-25
> > > **Thank you for your feedback!**
> > >
> > > We sincerely appreciate the detailed comments and constructive suggestions. Based on your feedback, we have made significant revisions to the paper.
> > > - Specifically, we have newly added or revised Figure 3, 4, 5, and 8; and Table 2, 3, 4, and 5.
> > > - Additionally, we have analyzed new experimental results for the single-modality case and included comparisons with other baseline methods, such as AudioCLIP [1], UniBind [2], and ViT-Lens [3].
> > >
> > > Detailed responses to each comment are provided below.
> > >
> > > > Limited Baseline Comparison. Thank you for including the results. However, the current presentation is challenging to interpret, as the plot combines acc(Z2), acc(Z1), and acc(All) into a single visualization, which makes the results less clear. I recommend separating these for better readability. Additionally, I suggest extending the comparisons to datasets beyond synthetic ones to demonstrate the scalability and applicability of the approach in real-world scenarios.
> > >
> > > Thank you for your valuable feedback. In response to your comment, we have added Table 3 to enhance the readability of the results originally presented in Figure 2. Similarly, Table 4 has been included to summarize the results shown in Figure 8.
> > >
> > > Additionally, as per your suggestion, we have incorporated three additional baseline results: AudioCLIP [1], UniBind [2], and ViT-Lens [3]. The experimental results have been updated accordingly in the manuscript.
> > >
> > > > Comparison with Related Work. I appreciate the inclusion of a discussion on related works. However, as mentioned in my initial feedback, incorporating a straightforward baseline (e.g., early or late fusion across modalities) would provide more meaningful insights. This should be applied to both synthetic and real-world datasets. Furthermore, it is unclear why CentroBIND does not leverage all modalities in Table 2 for a more comprehensive comparison. Including experimental results against the compared methods in related work will further strengthen the paper’s positioning for future works.
> > >
> > > We appreciate the reviewer’s valuable suggestions. Regarding the inclusion of early or late-fusion baselines, our synthetic experiments already include late-fusion methods, such as acc(All), presented in Figures 2(b), 6, and 8. Here, acc(All) refers to a late-fusion approach that employs MLP layers for combining multimodal embeddings. To address the reviewer’s comment, we have explicitly highlighted this point in the Appendix. Additionally, for the real-world experiments, we have added Table 5, which reports the conventional classification accuracy for each unimodal case as well as the multimodal case. In accordance with the reviewer’s suggestion, the decoder for the multimodal case was trained using a late-fusion method with MLP layers.
> > >
> > > In response to the comment on comparing with other methods, we have incorporated three additional multimodal models that utilize unified representation spaces: AudioCLIP [1], UniBind [2], and ViT-Lens [3]. While we appreciate the suggestion to include additional multimodal learning models, we believe these suggested models fall outside the scope of our paper. This is because the suggested models rely on supervised multimodal learning, which requires labeled data. In contrast, our method—along with the newly included models (AudioCLIP, UniBind, and ViT-Lens)—operates without labels during the training phase, aligning more closely with the objectives of our work.

---

> > > > ### Author Response · Authors · 2024-11-25
> > > > **Thank you for your feedback! [cont.]**
> > > >
> > > > > Comparison with Individual Modalities. I understand the primary objective is to develop unified representations. However, my request for experiments with individual modalities was motivated by the challenge of over-reliance on unimodal biases, which could undermine the efficacy of the proposed approach. Running these experiments would serve as a useful sanity check to ensure that the model does not depend disproportionately on single modalities and could be valuable for future references.
> > > >
> > > > Thank you for your valuable feedback. We agree that this is a valid concern and have conducted experiments involving single modalities. The results are presented in Table 5 of the manuscript. These results demonstrate that CentroBind outperforms FABind and UniBind in most cases. Both approaches are directly comparable, as they utilize the same backbone encoder models and are fine-tuned on the same dataset.
> > > >
> > > > For ViT-Lens, the results were computed using the provided pretrained models, which leverage a significantly larger Vision Transformer and are aligned using a substantially larger set of datasets. Due to its more powerful backbone, ViT-Lens outperforms CentroBind on video, audio, and combined modalities.
> > > >
> > > > However, CentroBind achieves superior performance for text. It is worth noting that in the future, CentroBind could adopt a similar approach by employing ViT as the backbone model while maintaining its dynamic anchor computation using centroids. In contrast, ViT-Lens relies on two fixed anchors derived from the CLIP model.
> > > >
> > > > We hope these additional results effectively address your feedback and concerns. Thank you once again for your constructive and insightful suggestions.

---

> ### Author Response · Authors · 2024-11-26
> **Results regarding Limited Baseline Comparison and Comparison with Individual Modalities. [1/3]**
>
> We have compared several dynamic anchor generation methods in the following Table 4.
>
> >## Table 2
> | **Method**  | **Tr, (Ev)**   | **Sar-1** | **Spk-1** | **Spk-3** | **Spk-5** |
> |:-------------:|:----------------:|:-----------:|:-----------:|:-----------:|:-----------:|
> | FABind      | $\mathcal{V}$, ($\mathcal{T}$) | 0.706     | 0.378     | 0.614     | 0.730     |
> | UniBind     |                | 0.544     | 0.170     | 0.328     | 0.478     |
> | AudioCLIP*  |                | 0.501     | 0.096     | 0.258     | 0.388     |
> | ViT-Lens*   |                | 0.506     | 0.097     | 0.343     | 0.449     |
> | CentroBind  |                | **0.716** | **0.474** | **0.736** | **0.836** |
> |-------------|----------------|-----------|-----------|-----------|-----------|
> | FABind      | $\mathcal{A}$, ($\mathcal{T}$) | 0.648     | 0.186     | 0.455     | 0.577     |
> | UniBind     |                | 0.628     | 0.220     | 0.399     | 0.501     |
> | AudioCLIP*  |                | 0.486     | 0.094     | 0.214     | 0.322     |
> | ViT-Lens*   |                | 0.484     | 0.077     | 0.214     | 0.313     |
> | CentroBind  |                | **0.691** | **0.290** | **0.546** | **0.714** |
> |-------------|----------------|-----------|-----------|-----------|-----------|
> | FABind      | $\mathcal{T}$, ($\mathcal{V}$) | 0.572     | 0.243     | 0.445     | 0.630     |
> | UniBind     |                | 0.484     | 0.129     | 0.262     | 0.404     |
> | AudioCLIP*  |                | 0.506     | 0.158     | 0.345     | 0.461     |
> | ViT-Lens*   |                | 0.502     | 0.168     | 0.323     | 0.423     |
> | CentroBind  |                | **0.694** | **0.368** | **0.670** | **0.791** |
> |-------------|----------------|-----------|-----------|-----------|-----------|
> | FABind      | $\mathcal{A}$, ($\mathcal{V}$) | 0.623     | 0.228     | **0.484** | 0.628     |
> | UniBind     |                | 0.567     | 0.199     | 0.367     | 0.514     |
> | AudioCLIP*  |                | 0.503     | 0.209     | 0.384     | 0.496     |
> | ViT-Lens*   |                | 0.500     | 0.149     | 0.332     | 0.451     |
> | CentroBind  |                | **0.683** | **0.243** | 0.475     | **0.632** |
> |-------------|----------------|-----------|-----------|-----------|-----------|
> | FABind      | $\mathcal{V}$, ($\mathcal{A}$) | 0.604     | 0.255     | 0.472     | 0.636     |
> | UniBind     |                | 0.506     | 0.126     | 0.280     | 0.429     |
> | AudioCLIP*  |                | 0.501     | 0.080     | 0.199     | 0.326     |
> | ViT-Lens*   |                | 0.533     | 0.219     | 0.438     | 0.575     |
> | CentroBind  |                | **0.626** | **0.326** | **0.548** | **0.703** |
> |-------------|----------------|-----------|-----------|-----------|-----------|
> | FABind      | $\mathcal{T}$, ($\mathcal{A}$) | 0.534     | 0.241     | 0.509     | 0.635     |
> | UniBind     |                | 0.514     | 0.091     | 0.248     | 0.365     |
> | AudioCLIP*  |                | 0.477     | 0.088     | 0.309     | 0.439     |
> | ViT-Lens*   |                | 0.475     | 0.070     | 0.214     | 0.329     |
> | CentroBind  |                | **0.655** | **0.346** | **0.610** | **0.741** |
>
> Models with asterisk are evaluated in different experimental settings (encoder backbone, pretraining datasets).
>
> Table 2 presents the classification accuracy for zero-shot cross-modal sarcasm detection and speaker classification tasks. CentroBind achieves significantly outperform top-1 accuracy compared to all baseline methods in both tasks. Our additional baselines, UniBind, AudioCLIP, and ViT-Lens, perform poorly in the zero-shot cross-modal experiment.

---

> ### Author Response · Authors · 2024-11-26
> **Results regarding Limited Baseline Comparison and Comparison with Individual Modalities. [2/3]**
>
> ___
> ## Table 4
> | **Backbone**     | **Method**              |    [Fig. 8a |        and             |   8b]                                    | [Fig. 8c | and  | 8d]|
> |:------------------:|:-------------------------:|:-------------:|:-------------:|:--------------------------:|:-------------:|:-------------:|:--------------------------:|
> |                              |                                   | **$X_2$**            | **$X_4$** | **$X_1,\cdots,X_4$** | **$X_2$** | **$X_4$** | **$X_1,\cdots,X_4$** |
> | **Pre-trained**  | $ \times$               | 0.115       | 0.296       | 0.566                    | 0.099       | 0.256       | 0.537                    |
> |                  | FABind- $X_4$         | 0.124       | 0.297       | **0.639**                | 0.115       | 0.263       | 0.540                    |
> |                  | CentroBind             | 0.131       | **0.363**   | 0.618                    | **0.116**   | 0.336       | 0.563                    |
> |                  | Weighted average       | 0.133       | 0.342       | 0.609                    | 0.102       | 0.338       | 0.574                    |
> |                  | Random + intra learning| 0.131       | 0.347       | 0.613                    | 0.105       | **0.353**   | 0.554                    |
> |                  | Random anchor          | **0.147**   | 0.359       | 0.619                    | 0.097       | 0.327       | **0.579**                |
> | **Random**       | $\times$               | 0.092       | 0.114       | 0.487                    | 0.067       | 0.176       | 0.465                    |
> |                  | FABind- $X_4$         | 0.131       | 0.143       | 0.575                    | 0.112       | 0.153       | 0.523                    |
> |                  | CentroBind             | 0.132       | **0.355**   | **0.626**                | **0.113**   | **0.336**   | **0.559**                |
> |                  | Weighted average       | 0.115       | 0.347       | 0.619                    | 0.109       | **0.336**   | 0.556                    |
> |                  | Random + intra learning| 0.132       | 0.333       | 0.602                    | 0.104       | 0.309       | 0.562                    |
> |                  | Random anchor          | **0.145**   | 0.354       | 0.618                    | 0.097       | 0.324       | 0.552                    |
>
>
>
> Table 4 presents the classification accuracy corresponding to the experiments illustrated in [Figure 8](https://anonymous.4open.science/api/repo/temp-C948/file/Comparison_dynamic_anchor.png?v=4844fb0a). In the experiments depicted in Figures 8a and 8b, the information level varies such that $X_1$, $X_2$, and $X_3$ are highly noisy, while $X_4$ is highly informative. Conversely, in Figures 8c and 8d, $X_1$ and $X_2$ are uninformative, whereas $X_3$ and $X_4$ are high-quality modalities. Each modality is selected as the fixed anchor modality in turn, and we compare CentroBind with FABind, weighted average, random anchor, and random anchor with intra-learning methods.
>
> When modality quality is imbalanced, the centroid-based approach may produce suboptimal dynamic anchor constructions, allowing other methods, such as weighted averages, to achieve better results in specific scenarios. Despite this, CentroBind consistently performs either better or on par with weighted average methods, showcasing its robustness to modality imbalance. Furthermore, the random anchor method also achieves competitive performance. These observations suggest that the choice of dynamic anchor generation method may have limited impact on final performance, as long as all encoders are appropriately trained during the process.

---

> > ### Author Response · Authors · 2024-11-26
> > **Results regarding Limited Baseline Comparison and Comparison with Individual Modalities. [3/3]**
> >
> > ### Table 5
> > | **Method**       | **Modality**                          | **Sar-1**  | **Spk-1**  | **Spk-3**  | **Spk-5**  |
> > |:------------------:|:----------------------------------:|:------------:|:------------:|:------------:|:------------:|
> > | FABind           | $\mathcal{T}$                                | 0.606      | 0.219      | 0.458      | 0.632      |
> > | UniBind          |                                       | 0.600      | 0.214      | 0.412      | 0.569      |
> > | AudioCLIP*       |                                       | 0.488      | 0.155      | 0.280      | 0.388      |
> > | ViT-Lens*        |                                       | 0.543      | 0.172      | 0.342      | 0.472      |
> > | CentroBind       |                                       | **0.667**  | **0.287**  | **0.507**  | **0.642**  |
> > |----------------|-------------------------------|------------|------------|------------|------------|
> > | FABind           | $\mathcal{V}$                                | 0.668      | 0.375      | 0.587      | 0.691      |
> > | UniBind          |                                       | 0.658      | 0.381      | 0.641      | 0.770      |
> > | AudioCLIP*       |                                       | 0.504      | 0.110      | 0.275      | 0.414      |
> > | ViT-Lens*        |                                       | **0.697**  | **0.586**  | **0.738**  | **0.797**  |
> > | CentroBind       |                                       | 0.670      | 0.380      | 0.609      | 0.726      |
> > |----------------|-------------------------------|------------|------------|------------|------------|
> > | FABind           | $\mathcal{A}$                                | 0.639      | 0.201      | 0.457      | 0.599      |
> > | UniBind          |                                       | 0.633      | 0.272      | 0.528      | 0.691      |
> > | AudioCLIP*       |                                       | 0.525      | 0.158      | 0.343      | 0.454      |
> > | ViT-Lens*        |                                       | **0.686**  | **0.396**  | **0.664**  | **0.8**    |
> > | CentroBind       |                                       | 0.616      | 0.234      | 0.461      | 0.609      |
> > |----------------|-------------------------------|------------|------------|------------|------------|
> > | FactorCL*        | $\substack{\mathcal{V}, \mathcal{A}, \mathcal{T} \\ (\mathcal{V}, \mathcal{A}, \mathcal{T})}$ | 0.699      | -          | -          | -          |
> > | SimMMDG*         |                                       | 0.725      | -          | -          | -          |
> > | FABind           |                                       | 0.678      | 0.343      | 0.554      | 0.677      |
> > | UniBind          |                                       | 0.646      | 0.383      | 0.622      | 0.764      |
> > | AudioCLIP*       |                                       | 0.530      | 0.119      | 0.261      | 0.378      |
> > | ViT-Lens*        |                                       | **0.731**  | **0.506**  | **0.736**  | **0.812**  |
> > | CentroBind       |                                       | 0.704      | 0.346      | 0.594      | 0.733      |
> >
> > Models with asterisk are evaluated in different experimental settings (encoder backbone, pretraining datasets).
> >
> > Table 5 summarizes the classification accuracy of FABind, UniBind, and CentroBind for both single-modal and multimodal settings, using embeddings extracted by the binding methods and a simple decoder for classification. CentroBind generally outperforms other methods in sarcasm detection, while UniBind shows better performance in speaker classification, likely due to its use of LLM-augmented descriptions as additional modality. Notably, CentroBind consistently surpasses FABind. Additionally, CentroBind can incorporate LLM-augmented descriptions as an extra modality, which could further enhance its performance.
> >
> > For reference, the table also includes sarcasm detection results from FactorCL, SimMMDG, AudioCLIP, and ViT-Lens. While ViT-Lens achieves higher accuracy due to larger backbone encoders and extensive pretraining, it can be considered a variant of FABind. Applying CentroBind's dynamic anchor method to ViT-Lens or integrating Vision Transformers into CentroBind could further boost performance.

---

> ### Author Response · Authors · 2024-12-02
> **Thank you for your review!**
>
> Dear Reviewer YKP7,
>
> Thank you for your time and effort in reviewing our work. We hope you have had the chance to review our responses and clarifications. As the discussion period is drawing to a close, we would greatly appreciate it if you could confirm whether our updates have fully addressed your concerns.
>
> Thanks!
>
> Best regards,
> The Authors

---

### Official Review · Reviewer_BBKi · 2024-11-02

**Soundness:** 4
**Presentation:** 4
**Contribution:** 4
**Rating:** 8
**Confidence:** 4

**Summary:**

This paper proposes a novel method to eliminate the need for an anchor in multimodal representation. Such method is based on centroid anchors computed from the modalities. The method faces a real problem in the largest part of current multimodal models that mainly rely on text anchors. The proposed method is novel and of interest to the community.

**Strengths:**

1) The paper faces the problem of misalignment among non-anchor modalities, which is a real problem in multimodal alignment.
2) The mathematical solutions and proofs to the problem, as well as the mutual information perspective, make the paper valuable.
3) Although simple, the experiments confirm the theoretical intuitions.
4) The paper is well-written and well-presented.

**Weaknesses:**

1) A tSNE visualization of the learned space could be strengthen the claims.
2) Maybe more detailed experiments with existing embedding models could be conducted to generalize the results. But the paper is fine also as it is now. :)


Minor comments:

1) Supplementary materials should have been submitted together with the paper.
2) The Saxon genitive should be avoided in scientific writing, although I know that both ChatGPT and Grammarly suggest to use it. However, I suggest the authors to remove all the Saxon genitives from the paper.

**Questions:**

1) Could the authors better explain how they dynamically adjust the centroids? I mean, after an iteration, or step by step also inside the iteration?
2) How would this model scale up to multiple (more than three) modalities?

---

> ### Author Response · Authors · 2024-11-23
>
> We would like to thank the reviewer for their comments and suggestions. Please find the following responses to the questions and comments.
>
> > **Weaknesses 1.** A tSNE visualization of the learned space could be strengthen the claims.
>
> Thank you for your suggestion. We have attached the t-SNE and UMAP visualizations of the learned feature spaces of the synthetic dataset for FABind and CentroBind [here](https://anonymous.4open.science/api/repo/temp-C948/file/tsne_umap.png). We have included the plots in the revised manuscript.
>
> > **Weaknesses 2.** Maybe more detailed experiments with existing embedding models could be conducted to generalize the results. But the paper is fine also as it is now. :)
>
> Thank you for the suggestion. We strongly agree with this comment, and we will add more baseline results, such as UniBind [1] and AudioCLIP [2]. Once the results are ready (in a few days), we will update the paper and response to the reviewers.
>
> [1] Lyu et al., UniBind: LLM-Augmented Unified and Balanced Representation Space to Bind Them All
>
> [2] Guzhov at el., Audioclip: Extending clip to image, text and audio.
>
> > **Minor comment 1.** Supplementary materials should have been submitted together with the paper.
>
> Thank you for pointing this out. The revised version now includes appendix.
>
> > **Minor comment 2.** The Saxon genitive should be avoided in scientific writing, although I know that both ChatGPT and Grammarly suggest to use it. However, I suggest the authors to remove all the Saxon genitives from the paper.
>
> Thank you for the comment. We have revised the paper accordingly.
>
> > **Question 1.** Could the authors better explain how they dynamically adjust the centroids? I mean, after an iteration, or step by step also inside the iteration?
>
> The dynamic anchor is generated for every pair of samples in each iteration. Specifically, in an iteration with a minibatch dataset $B$, we generate $|B|$ dynamic anchors for each pair of samples.
> After the iteration is completed and the encoders are updated, the dynamic anchors for the subsequent iteration will differ due to these updates. As a result, the dynamic anchors are continually adjusted throughout the training process, adapting to the evolving embeddings generated by the encoders.
>
> > **Question 2.** How would this model scale up to multiple (more than three) modalities?
>
> Similar to FABind, CentroBind is capable of learning from multiple modalities, including more than three modalities. Due to our limited computational resources, we chose to conduct experiments with three modalities for the real-world dataset case. However, we also explored scenarios involving more than three modalities through synthetic experiments. Figure 2 presents the results for four modalities, while Figures 6 and 7 in the appendix illustrate cases involving six and eight modalities, respectively. These experiments demonstrate that CentroBind successfully constructs a unified representation space for more than three modalities.

---

> ### Author Response · Authors · 2024-11-26
> **Additional experimental results regarding Weaknesses 2. [1/2]**
>
> We have updated our paper based on the reviewer's comment in Weaknesses 1,2, and 4 in the following Table. Table 2 and 5 summarize comparison of baseline methods and CentroBind on a real-world dataset.
>
>
> ____
> >## Table 2
> | **Method**  | **Tr, (Ev)**   | **Sar-1** | **Spk-1** | **Spk-3** | **Spk-5** |
> |:-------------:|:----------------:|:-----------:|:-----------:|:-----------:|:-----------:|
> | FABind      | $\mathcal{V}$, ($\mathcal{T}$) | 0.706     | 0.378     | 0.614     | 0.730     |
> | UniBind     |                | 0.544     | 0.170     | 0.328     | 0.478     |
> | AudioCLIP*  |                | 0.501     | 0.096     | 0.258     | 0.388     |
> | ViT-Lens*   |                | 0.506     | 0.097     | 0.343     | 0.449     |
> | CentroBind  |                | **0.716** | **0.474** | **0.736** | **0.836** |
> |-------------|----------------|-----------|-----------|-----------|-----------|
> | FABind      | $\mathcal{A}$, ($\mathcal{T}$) | 0.648     | 0.186     | 0.455     | 0.577     |
> | UniBind     |                | 0.628     | 0.220     | 0.399     | 0.501     |
> | AudioCLIP*  |                | 0.486     | 0.094     | 0.214     | 0.322     |
> | ViT-Lens*   |                | 0.484     | 0.077     | 0.214     | 0.313     |
> | CentroBind  |                | **0.691** | **0.290** | **0.546** | **0.714** |
> |-------------|----------------|-----------|-----------|-----------|-----------|
> | FABind      | $\mathcal{T}$, ($\mathcal{V}$) | 0.572     | 0.243     | 0.445     | 0.630     |
> | UniBind     |                | 0.484     | 0.129     | 0.262     | 0.404     |
> | AudioCLIP*  |                | 0.506     | 0.158     | 0.345     | 0.461     |
> | ViT-Lens*   |                | 0.502     | 0.168     | 0.323     | 0.423     |
> | CentroBind  |                | **0.694** | **0.368** | **0.670** | **0.791** |
> |-------------|----------------|-----------|-----------|-----------|-----------|
> | FABind      | $\mathcal{A}$, ($\mathcal{V}$) | 0.623     | 0.228     | **0.484** | 0.628     |
> | UniBind     |                | 0.567     | 0.199     | 0.367     | 0.514     |
> | AudioCLIP*  |                | 0.503     | 0.209     | 0.384     | 0.496     |
> | ViT-Lens*   |                | 0.500     | 0.149     | 0.332     | 0.451     |
> | CentroBind  |                | **0.683** | **0.243** | 0.475     | **0.632** |
> |-------------|----------------|-----------|-----------|-----------|-----------|
> | FABind      | $\mathcal{V}$, ($\mathcal{A}$) | 0.604     | 0.255     | 0.472     | 0.636     |
> | UniBind     |                | 0.506     | 0.126     | 0.280     | 0.429     |
> | AudioCLIP*  |                | 0.501     | 0.080     | 0.199     | 0.326     |
> | ViT-Lens*   |                | 0.533     | 0.219     | 0.438     | 0.575     |
> | CentroBind  |                | **0.626** | **0.326** | **0.548** | **0.703** |
> |-------------|----------------|-----------|-----------|-----------|-----------|
> | FABind      | $\mathcal{T}$, ($\mathcal{A}$) | 0.534     | 0.241     | 0.509     | 0.635     |
> | UniBind     |                | 0.514     | 0.091     | 0.248     | 0.365     |
> | AudioCLIP*  |                | 0.477     | 0.088     | 0.309     | 0.439     |
> | ViT-Lens*   |                | 0.475     | 0.070     | 0.214     | 0.329     |
> | CentroBind  |                | **0.655** | **0.346** | **0.610** | **0.741** |
>
> Models with asterisk are evaluated in different experimental settings (encoder backbone, pretraining datasets).
>
> Table 2 presents the classification accuracy for zero-shot cross-modal sarcasm detection and speaker classification tasks. CentroBind achieves significantly outperform top-1 accuracy compared to all baseline methods in both tasks. Our additional baselines, UniBind, AudioCLIP, and ViT-Lens, perform poorly in the zero-shot cross-modal experiment.

---

> > ### Author Response · Authors · 2024-11-26
> > **Additional experimental results regarding Weaknesses 2. [2/2]**
> >
> > ### Table 5
> > | **Method**       | **Modality**                          | **Sar-1**  | **Spk-1**  | **Spk-3**  | **Spk-5**  |
> > |:------------------:|:----------------------------------:|:------------:|:------------:|:------------:|:------------:|
> > | FABind           | $\mathcal{T}$                                | 0.606      | 0.219      | 0.458      | 0.632      |
> > | UniBind          |                                       | 0.600      | 0.214      | 0.412      | 0.569      |
> > | AudioCLIP*       |                                       | 0.488      | 0.155      | 0.280      | 0.388      |
> > | ViT-Lens*        |                                       | 0.543      | 0.172      | 0.342      | 0.472      |
> > | CentroBind       |                                       | **0.667**  | **0.287**  | **0.507**  | **0.642**  |
> > |----------------|-------------------------------|------------|------------|------------|------------|
> > | FABind           | $\mathcal{V}$                                | 0.668      | 0.375      | 0.587      | 0.691      |
> > | UniBind          |                                       | 0.658      | 0.381      | 0.641      | 0.770      |
> > | AudioCLIP*       |                                       | 0.504      | 0.110      | 0.275      | 0.414      |
> > | ViT-Lens*        |                                       | **0.697**  | **0.586**  | **0.738**  | **0.797**  |
> > | CentroBind       |                                       | 0.670      | 0.380      | 0.609      | 0.726      |
> > |----------------|-------------------------------|------------|------------|------------|------------|
> > | FABind           | $\mathcal{A}$                                | 0.639      | 0.201      | 0.457      | 0.599      |
> > | UniBind          |                                       | 0.633      | 0.272      | 0.528      | 0.691      |
> > | AudioCLIP*       |                                       | 0.525      | 0.158      | 0.343      | 0.454      |
> > | ViT-Lens*        |                                       | **0.686**  | **0.396**  | **0.664**  | **0.8**    |
> > | CentroBind       |                                       | 0.616      | 0.234      | 0.461      | 0.609      |
> > |----------------|-------------------------------|------------|------------|------------|------------|
> > | FactorCL*        | $\substack{\mathcal{V}, \mathcal{A}, \mathcal{T} \\ (\mathcal{V}, \mathcal{A}, \mathcal{T})}$ | 0.699      | -          | -          | -          |
> > | SimMMDG*         |                                       | 0.725      | -          | -          | -          |
> > | FABind           |                                       | 0.678      | 0.343      | 0.554      | 0.677      |
> > | UniBind          |                                       | 0.646      | 0.383      | 0.622      | 0.764      |
> > | AudioCLIP*       |                                       | 0.530      | 0.119      | 0.261      | 0.378      |
> > | ViT-Lens*        |                                       | **0.731**  | **0.506**  | **0.736**  | **0.812**  |
> > | CentroBind       |                                       | 0.704      | 0.346      | 0.594      | 0.733      |
> >
> >
> > Table 5 summarizes the classification accuracy of FABind, UniBind, and CentroBind for both single-modal and multimodal settings, using embeddings extracted by the binding methods and a simple decoder for classification. CentroBind generally outperforms other methods in sarcasm detection, while UniBind shows better performance in speaker classification, likely due to its use of LLM-augmented descriptions as additional modality. Notably, CentroBind consistently surpasses FABind. Additionally, CentroBind can incorporate LLM-augmented descriptions as an extra modality, which could further enhance its performance.
> >
> > For reference, the table also includes sarcasm detection results from FactorCL, SimMMDG, AudioCLIP, and ViT-Lens. While ViT-Lens achieves higher accuracy due to larger backbone encoders and extensive pretraining, it can be considered a variant of FABind. Applying CentroBind's dynamic anchor method to ViT-Lens or integrating Vision Transformers into CentroBind could further boost performance.

---

> > > ### Comment · Reviewer_BBKi · 2024-11-30
> > > **Thanks for reply**
> > >
> > > I would like to thank the authors for their reply and their clarifications.
> > >
> > > Good luck!

---

> ### Author Response · Authors · 2024-12-02
> **Thank you for reviewing our work!**
>
> Dear Reviewer BBKi,
>
> Thank you very much for your positive feedback on our work!
> We greatly appreciate your time and thoughtful review.
>
> Best regards,
> The Authors

---

### Official Review · Reviewer_2XLP · 2024-11-04

**Soundness:** 3
**Presentation:** 3
**Contribution:** 3
**Rating:** 5
**Confidence:** 4

**Summary:**

The paper explores limitations in current Fixed-Anchor-Bind (FABIND) methods used for multi-modal learning, which commonly align various modalities using a fixed anchor like images or text.  These methods struggle to capture essential intra-modal and shared information among non-anchored modalities.  To address these shortcomings, the authors propose a novel approach, CentroBind, which dynamically computes centroid-based anchors from all modalities, enhancing representation alignment.  Their experimental results demonstrate that CentroBind outperforms traditional FABIND methods on synthetic and real-world datasets.

**Strengths:**

1、Innovative Approach: The introduction of CentroBind addresses key limitations in existing multi-modal learning methods by using dynamic anchors, offering a fresh perspective and improving multi-modal alignment.


2、Theoretical Rigor: The paper provides a comprehensive mathematical analysis to support the efficacy of CentroBind, ensuring a solid foundation for the proposed approach.

**Weaknesses:**

1、Robustness of anchor generation: CentroBind takes the centroids of all modalities to generate anchors, how does this approach perform in the face of modalities with significantly different information densities? For example, if some modalities contain more or less useful information than others, does the centroid bias towards these modalities, resulting in an imbalance in the representation space?

2、Effect of different modal weights: All modes are treated as equally weighted when calculating the centroid anchor. Have methods been investigated for assigning different weights to each modality to better capture inter-modality differences? Does this weight assignment improve or optimize the overall performance of the model?

3、CentroBind aims to maximize both intra-and inter-modality information, but is there an information bottleneck effect (e.g. information may be compressed in high-dimensional embedding Spaces)? How does this bottleneck effect affect the representation power of the model, especially in tasks where multimodal interactions are very complex? Have experiments been conducted to verify the balance between information maximization and representation compactness?

4、Does CentroBind suffer from potential optimization pitfalls when multiple modalities are strongly correlated or there is obvious synergy information between modalities? For example, when two modes are highly dependent, is the centroid overly pulled towards a particular mode? In this case, how can the optimization process be adjusted or improved to capture real multimodal relationships?

5、Boundary conditions for theoretical analysis: The theoretical analysis in the paper explores the advantages of CentroBind, but are there certain boundary conditions or assumptions under which CentroBind might lose its advantage? For example, does this approach still work when the data is unevenly distributed or the amount of data is severely imbalanced?

6、Loss function Optimization: CentroBind's loss function combines multiple InfoNCE loss terms, but does this design affect the convergence speed or stability of the model? Has the effect of different temperature parameters or weight of the loss term been explored to optimize the convergence of the model?

**Questions:**

See Weaknesses

---

> ### Author Response · Authors · 2024-11-23
> **Response to Reviewer 2XLP [1/2]**
>
> We would like to thank you for your comments and suggestions. We have improved the paper based on the comments and questions. We response to each weaknesses as follows.
>
> > **Weaknesses 1.** Robustness of anchor generation.
>
> Thank you for highlighting the modality imbalance problem. You are correct that CentroBind does not explicitly account for the quality of each modality, which could, in theory, lead to worse performance compared to other dynamic anchor generation methods. To address this concern, we conducted the synthetic experiment shown in Figure 2 under a modality imbalance scenario, where the first modality is the worst, and the fourth modality is the best. In this case, CentroBind outperformed other baseline methods.
>
> In response to your comments regarding weighted averages and other dynamic anchor generation methods, we performed additional experiments with extremely unbalanced information densities across modalities. Details of these experiments ([Figure 8](https://anonymous.4open.science/api/repo/temp-C948/file/Comparison_dynamic_anchor.png?v=4844fb0a)) have been added to Appendix C.1. In such imbalanced scenarios, weighted average and CentroBind exhibit similar performance, suggesting that CentroBind is robust to modality imbalance. We hypothesize that this robustness arises because dynamic anchor generation has limited influence on the final result, provided there is no fixed anchor or a frozen encoder, as in FABind.
>
> Although CentroBind demonstrates robustness to modality imbalance, we believe addressing this issue is beyond the scope of our paper. Solving the modality imbalance problem often requires additional knowledge, such as information about downstream tasks, which contrasts with the representation learning setting.
>
> Finally, we note that in practice, binding methods are often trained on diverse datasets containing varying information densities within the same modality. Assuming the information density is unknown, CentroBind provides a reasonable approach for anchor generation, as it represents an optimal point for multimodal alignment in Euclidean space.
>
> > **Weaknesses 2.** Effect of different modal weights.
>
> Please find the above response to weaknesses 1.
>
> > **Weaknesses 3.** CentroBind aims to maximize both intra-and inter-modality information, but is there an information bottleneck effect (e.g. information may be compressed in high-dimensional embedding Spaces).
>
> Thank you for the interesting question. We acknowledge that we have not explored the information bottleneck problem in the context of representation learning and have not conducted related experiments. While this is indeed an interesting direction for future research, we believe it lies outside the scope of our current paper.
> This issue is not specific to our method but is a broader concern that applies to all representation learning approaches. Addressing the information bottleneck problem would require a more generalized study that goes beyond the focus of our work.
>
> > **Weaknesses 4.**  Does CentroBind suffer from potential optimization pitfalls when multiple modalities are strongly correlated or there is obvious synergy information between modalities?
>
> Thank you for the question regarding dependency among modalities. We believe that strong correlations between modalities are always beneficial for CentroBind in extracting representations from multi-modal samples. Our reasoning is as follows:
> 1) The fundamental motivation for using multi-modal data is to capture a comprehensive representation shared among multiple modalities. This implies that samples from each modality should share some part of the true underlying representation, such as the concept of a Platonic representation [1]. We think that strong correlation implies that modalities share the core representation strongly, which would be helpful for multimodal learning;
> 2) When modalities are correlated, they naturally contribute more weight to the centroid during dynamic anchor generation. Intuitively, making the anchor embedding align more closely with highly correlated modalities is advantageous, as their shared information is more likely to reflect the true representation compared to the information from uncorrelated modalities.
>
> To demonstrate this further, in line with Weakness 1, we conducted experiments where modality quality varies. Specifically, sub-figures (c) and (d) in [Figure 8](https://anonymous.4open.science/api/repo/temp-C948/file/Comparison_dynamic_anchor.png?v=4844fb0a) of the Appendix depict cases where two modalities are highly correlated while others are not. In these scenarios, CentroBind performs well and is even competitive with the weighted average method, which relies on additional prior knowledge of modality quality.
>
> [1] Huh et al., The Platonic Representation Hypothesis.

---

> ### Author Response · Authors · 2024-11-23
> **Response to Reviewer 2XLP [2/2]**
>
> > **Weaknesses 5.**  Boundary conditions for theoretical analysis: The theoretical analysis in the paper explores the advantages of CentroBind, but are there certain boundary conditions or assumptions under which CentroBind might lose its advantage? For example, does this approach still work when the data is unevenly distributed or the amount of data is severely imbalanced?
>
> The theoretical analysis of CentroBind is based on the lower bound of its objective function, as established in Theorem 1. Notably, this bound is derived in a general setting without making any assumptions about the data distribution. In fact, the bound is sample-wise valid, meaning our analysis holds for any data distribution.
>
> While the bound itself is distribution-independent, we acknowledge that the gap between the objective function and its lower bound likely depends on the underlying data distribution. A theoretical characterization of this gap (e.g., assessing its tightness under specific distributional assumptions) would indeed be an interesting direction for future work. However, we consider this to be beyond the scope of our current study, as our focus is on scenarios where the quality of modalities and data distribution are unknown.
>
> > **Weaknesses 6.** Loss function Optimization: CentroBind's loss function combines multiple InfoNCE loss terms, but does this design affect the convergence speed or stability of the model?
>
> When a dataset contains $M$ modalities, the loss function of CentroBind comprises $M$ InfoNCE losses, whereas FABind (e.g., ImageBind) uses $M-1$ InfoNCE losses. Thus, the number of loss functions is comparable between the two methods. The primary distinction lies in the anchor generation strategy: CentroBind employs dynamic anchors, while FABind uses fixed anchors. We believe dynamic anchor generation enables faster convergence since aligning embeddings with their centroid is geometrically easier (in terms of distance) compared to aligning them with fixed embeddings.
>
> To support this claim, we have included a plot of the training loss curves for CentroBind and FABind in Appendix C.1 (page 19). This plot illustrates that CentroBind's loss converges to its saturation point faster than FABind's loss.
>
> Regarding the temperature parameter in the InfoNCE loss, we have not explicitly explored its effect on the convergence rate, as this was deemed beyond the scope of our study. However, to the best of our knowledge, previous works such as [2], [3], and [4] have investigated the role of the temperature parameter in InfoNCE loss. These studies demonstrate that the temperature influences the local separation and uniformity of the embedding distributions.
>
> [2] Wang et al., Understanding the Behaviour of Contrastive Loss
>
> [3] Zhang et al., How Does SimSiam Avoid Collapse Without Negative Samples? A Unified Understanding with Self-supervised Contrastive Learning
>
> [4] Koromilas et al., Bridging Mini-Batch and Asymptotic Analysis in Contrastive Learning: From InfoNCE to Kernel-Based Losses

---

> ### Author Response · Authors · 2024-11-26
> **Additional experiment results regarding Weaknesses 1, 2, and 4**
>
> We have updated our paper based on your comment on Weaknesses 1,2, and 4 in the following table.
> Table 4 summarizes the comparison of dynamic anchor generation methods shown in Figure 8.
>
> ## Table 4
> | **Backbone**     | **Method**              |    [Fig. 8a |        and             |   8b]                                    | [Fig. 8c | and  | 8d]|
> |:------------------:|:-------------------------:|:-------------:|:-------------:|:--------------------------:|:-------------:|:-------------:|:--------------------------:|
> |                              |                                   | **$X_2$**            | **$X_4$** | **$X_1,\cdots,X_4$** | **$X_2$** | **$X_4$** | **$X_1,\cdots,X_4$** |
> | **Pre-trained**  | $\times$               | 0.115       | 0.296       | 0.566                    | 0.099       | 0.256       | 0.537                    |
> |                  | FABind- $X_4$         | 0.124       | 0.297       | **0.639**                | 0.115       | 0.263       | 0.540                    |
> |                  | CentroBind             | 0.131       | **0.363**   | 0.618                    | **0.116**   | 0.336       | 0.563                    |
> |                  | Weighted average       | 0.133       | 0.342       | 0.609                    | 0.102       | 0.338       | 0.574                    |
> |                  | Random + intra learning| 0.131       | 0.347       | 0.613                    | 0.105       | **0.353**   | 0.554                    |
> |                  | Random anchor          | **0.147**   | 0.359       | 0.619                    | 0.097       | 0.327       | **0.579**                |
> | **Random**       | $\times$               | 0.092       | 0.114       | 0.487                    | 0.067       | 0.176       | 0.465                    |
> |                  | FABind- $X_4$         | 0.131       | 0.143       | 0.575                    | 0.112       | 0.153       | 0.523                    |
> |                  | CentroBind             | 0.132       | **0.355**   | **0.626**                | **0.113**   | **0.336**   | **0.559**                |
> |                  | Weighted average       | 0.115       | 0.347       | 0.619                    | 0.109       | **0.336**   | 0.556                    |
> |                  | Random + intra learning| 0.132       | 0.333       | 0.602                    | 0.104       | 0.309       | 0.562                    |
> |                  | Random anchor          | **0.145**   | 0.354       | 0.618                    | 0.097       | 0.324       | 0.552                    |
>
>
>
> Table 4 presents the classification accuracy corresponding to the experiments illustrated in [Figure 8](https://anonymous.4open.science/api/repo/temp-C948/file/Comparison_dynamic_anchor.png?v=4844fb0a). In the experiments depicted in Figures 8a and 8b, the information level varies such that $X_1$, $X_2$, and $X_3$ are highly noisy, while $X_4$ is highly informative. Conversely, in Figures 8c and 8d, $X_1$ and $X_2$ are uninformative, whereas $X_3$ and $X_4$ are high-quality modalities. Each modality is selected as the fixed anchor modality in turn, and we compare CentroBind with FABind, weighted average, random anchor, and random anchor with intra-learning methods.
>
> When modality quality is imbalanced, the centroid-based approach may produce suboptimal dynamic anchor constructions, allowing other methods, such as weighted averages, to achieve better results in specific scenarios. Despite this, CentroBind consistently performs either better or on par with weighted average methods, showcasing its robustness to modality imbalance. Furthermore, the random anchor method also achieves competitive performance. These observations suggest that the choice of dynamic anchor generation method may have limited impact on final performance, as long as all encoders are appropriately trained during the process.

---

> ### Author Response · Authors · 2024-12-02
> **Thanks for reviewing our work!**
>
> Dear Reviewer 2XLP,
>
> Thank you for your effort in reviewing our work! We hope you have had the chance to review our responses and clarifications. As the discussion period is drawing to a close, we would greatly appreciate it if you could confirm whether our updates have fully addressed your concerns.
>
> Thank you again for your time and thoughtful review.
>
> Best regards,
> The Authors

---

### Author Response · Authors · 2024-11-23
**General response to reviewers**

We would like to thank the reviewers for their thoughtful feedback and suggestions.
Overall, the reviewers highlighted the following **strengths** of our paper:
> 1. Proposing an innovative approach to multimodal alignment that overcomes the limitations of existing fixed-anchor-bind (FABind) techniques. (Reviewers: 2XLP, BBKi, NCrk)
2. Demonstrating theoretical rigor and a solid foundation through comprehensive mathematical analyses and proofs supporting the effectiveness of CentroBind. (Reviewers: 2XLP, BBKi, NCrk)
3. Providing empirical validation and showing performance improvements on both synthetic and real-world datasets. (Reviewers: 2XLP, BBKi, NCrk)
4. Addressing real-world challenges with potential for broader impact by tackling the problem of misalignment among non-anchor modalities. (Reviewers: 2XLP, BBKi, NCrk, YKP7, aFcb)
5. Presenting a well-written manuscript characterized by clarity, organization, and ease of understanding, making complex theoretical concepts accessible through a concise writing style. (Reviewers: BBKi, YKP7, aFcb, NCrk)


The reviewers' main **criticisms** can be summarized as follows followed by our general response:
> 1. General concerns about computing centroid anchors by averaging, as it may render the method sensitive to outliers, especially with small batch sizes. Additionally, when one of the modalities is extremely “bad,” this may hurt performance. Reviewers suggest trying weighted averaging.

Thank you for the feedback; this is a valid point. In our original manuscript, we presented experiments with varying levels of modality quality in Figure 2 and observed that CentroBind demonstrated the best performance. Upon receiving this feedback, we conducted additional experiments where we pushed the modality qualities to the extreme. We compared CentroBind with weighted averaging based on prior knowledge, as well as with random anchoring both with and without intra-learning. Our new results show that CentroBind is robust to these outliers and that shared information among non-anchors—which is one of our contributions—appears to be more critical for performance in the extreme case where one modality contains most of semantic information (and hence other modalities contain partial information without unique information). For more details on these additional experiments, please refer to our response to Reviewer 2XLP.

> 2. The experimental evaluations primarily compare CentroBind with FABind methods, lacking comparisons with other recent and relevant multimodal alignment frameworks.

In this paper, we focus on multimodal "representation" learning via contrastive learning that does not require annotated labels used in supervised learning. We would like to point out that FABind methods are the current state-of-the-art and the most widely used approaches in multimodal representation learning. Many recent approaches are also variants of FABind methods, and one of our main contributions is formally categorizing these methods, providing theoretical analyses, and proposing CentroBind based on these analyses. Additionally, we are currently working on adding another baselines using the UniBind [1] and AudioCLIP [2], which is one of the most recent state-of-the-art approaches with an available codebase.


> 3. The experiments could be more extensive, exploring CentroBind's performance under varying conditions such as different data distributions, noise levels, and with more modalities.

In response to [2XLP, YKP7, aFcb, ioPy], we compared various baseline methods for dynamic anchor selection, including weighted average and random anchor (with intra and non-intra learning). A common concern with using centroids is their sensitivity to outliers and inability to capture diverse data distributions. However, our additional experiments under varying conditions of information density and modal dependency demonstrate that CentroBind performs robustly.

Intuitively, a weighted average should outperform centroids as it incorporates additional knowledge about the dataset (e.g., information density and quality). Yet, CentroBind achieves comparable performance, suggesting that our method is robust to modality imbalance. Furthermore, since dataset quality can vary even within a single modality in practice, and modality importance is task-dependent, we argue that centroids are a reasonable choice for dynamic anchor selection within a representation learning framework where the downstream task is unknown.

Nonetheless, it is worth exploring the modality imbalance problem beyond the current setting, such as in supervised representation learning, and we have included a discussion of the modality imbalance problem in the paper.

[1] Lyu et al., UniBind: LLM-Augmented Unified and Balanced Representation Space to Bind Them All.

[2] Guzhov at el., Audioclip: Extending clip to image, text and audio.

---

> ### Author Response · Authors · 2024-11-25
> **General response to reviewers [new updates!]**
>
> We would like to thank the reviewers for their valuable suggestions on our paper. To address the comments on including more baselines, we have added additional experimental results and comparisons with more baseline models, including AudioCLIP [1], UniBind [2], and ViT-Lens [3].
>
> Specifically, the following updates have been made:
>
> 1) We have added Table 2, which includes baseline methods (UniBind, AudioCLIP, and ViT-Lens) and reports their classification accuracy for cross-modal sarcasm and speaker classification tasks. As shown in Table 2, CentroBind significantly outperforms these additional baseline methods.
> 2) To improve the readability of Figure 2, we have included Table 3, which provides a clearer presentation of the associated results.
> 3) We have added Table 4 to enhance the readability of Figure 8. Table 4 presents a comparison of various dynamic anchor generation methods, including FABind (anchored on highly informative modalities), weighted average, random anchor with intra-learning, and random anchor.
> 4) We have introduced Table 5, which provides conventional classification results (not zero-shot or cross-modal evaluations) for FABind, UniBind, AudioCLIP, ViT-Lens, and CentroBind.
>
> [1] Guzhov et al., Audioclip: Extending Clip to Image, Text and Audio, ICASSP 2022
>
> [2] Lyu et al., UniBind: LLM-Augmented Unified and Balanced Representation Space to Bind Them All, CVPR 2024
>
> [3] Lei et al., ViT-Lens: Towards Omni-modal Representations, CVPR 2024

---

> ### Author Response · Authors · 2024-11-25
> **Updated Table 2**
>
> We have updated our paper and summarized the important changes related to the experimental results.
>
> ____
> >## Table 2
> | **Method**  | **Tr, (Ev)**   | **Sar-1** | **Spk-1** | **Spk-3** | **Spk-5** |
> |:-------------:|:----------------:|:-----------:|:-----------:|:-----------:|:-----------:|
> | FABind      | $\mathcal{V}$, ($\mathcal{T}$) | 0.706     | 0.378     | 0.614     | 0.730     |
> | UniBind     |                | 0.544     | 0.170     | 0.328     | 0.478     |
> | AudioCLIP*  |                | 0.501     | 0.096     | 0.258     | 0.388     |
> | ViT-Lens*   |                | 0.506     | 0.097     | 0.343     | 0.449     |
> | CentroBind  |                | **0.716** | **0.474** | **0.736** | **0.836** |
> |-------------|----------------|-----------|-----------|-----------|-----------|
> | FABind      | $\mathcal{A}$, ($\mathcal{T}$) | 0.648     | 0.186     | 0.455     | 0.577     |
> | UniBind     |                | 0.628     | 0.220     | 0.399     | 0.501     |
> | AudioCLIP*  |                | 0.486     | 0.094     | 0.214     | 0.322     |
> | ViT-Lens*   |                | 0.484     | 0.077     | 0.214     | 0.313     |
> | CentroBind  |                | **0.691** | **0.290** | **0.546** | **0.714** |
> |-------------|----------------|-----------|-----------|-----------|-----------|
> | FABind      | $\mathcal{T}$, ($\mathcal{V}$) | 0.572     | 0.243     | 0.445     | 0.630     |
> | UniBind     |                | 0.484     | 0.129     | 0.262     | 0.404     |
> | AudioCLIP*  |                | 0.506     | 0.158     | 0.345     | 0.461     |
> | ViT-Lens*   |                | 0.502     | 0.168     | 0.323     | 0.423     |
> | CentroBind  |                | **0.694** | **0.368** | **0.670** | **0.791** |
> |-------------|----------------|-----------|-----------|-----------|-----------|
> | FABind      | $\mathcal{A}$, ($\mathcal{V}$) | 0.623     | 0.228     | **0.484** | 0.628     |
> | UniBind     |                | 0.567     | 0.199     | 0.367     | 0.514     |
> | AudioCLIP*  |                | 0.503     | 0.209     | 0.384     | 0.496     |
> | ViT-Lens*   |                | 0.500     | 0.149     | 0.332     | 0.451     |
> | CentroBind  |                | **0.683** | **0.243** | 0.475     | **0.632** |
> |-------------|----------------|-----------|-----------|-----------|-----------|
> | FABind      | $\mathcal{V}$, ($\mathcal{A}$) | 0.604     | 0.255     | 0.472     | 0.636     |
> | UniBind     |                | 0.506     | 0.126     | 0.280     | 0.429     |
> | AudioCLIP*  |                | 0.501     | 0.080     | 0.199     | 0.326     |
> | ViT-Lens*   |                | 0.533     | 0.219     | 0.438     | 0.575     |
> | CentroBind  |                | **0.626** | **0.326** | **0.548** | **0.703** |
> |-------------|----------------|-----------|-----------|-----------|-----------|
> | FABind      | $\mathcal{T}$, ($\mathcal{A}$) | 0.534     | 0.241     | 0.509     | 0.635     |
> | UniBind     |                | 0.514     | 0.091     | 0.248     | 0.365     |
> | AudioCLIP*  |                | 0.477     | 0.088     | 0.309     | 0.439     |
> | ViT-Lens*   |                | 0.475     | 0.070     | 0.214     | 0.329     |
> | CentroBind  |                | **0.655** | **0.346** | **0.610** | **0.741** |
>
>
> Table 2 presents the classification accuracy for zero-shot cross-modal sarcasm detection and speaker classification tasks. CentroBind achieves significantly outperform top-1 accuracy compared to all baseline methods in both tasks. Our additional baselines, UniBind, AudioCLIP, and ViT-Lens, perform poorly in the zero-shot cross-modal experiment.

---

> ### Author Response · Authors · 2024-11-25
> **Updated Table 3**
>
> ## Table 3
> | **Backbone**     | **Method**      | Unimodal  |    Unimodal                 |         Unimodal           |         Unimodal           | Multimodal                  |
> |:------------------:|:----------------:|:-------------:|:-------------:|:-------------:|:-------------:|:-------------------------:|
> |                              |                           | **$X_1$** | **$X_2$** | **$X_3$** | **$X_4$** | **$X_1,\cdots,X_4$** |
> | **Pre-trained**  | $\times$       | 0.2166      | 0.2878      | 0.3536      | 0.3923      | 0.6985                  |
> |                  | FABind- $X_1$  | 0.2180      | 0.2736      | 0.3210      | 0.2999      | 0.5541                  |
> |                  | FABind- $X_4$  | 0.2483      | 0.3349      | **0.4207**  | 0.3896      | **0.7024**              |
> |                  | CentroBind     | **0.2540**  | **0.3433**  | 0.4162      | **0.4559**  | 0.6974
> |-------------|----------------|-----------|-----------|-----------|-----------|--------------|
> | **Random**       | $\times$       | 0.2109      | 0.2472      | 0.2597      | 0.2815      | 0.6648                  |
> |                  | FABind- $X_1$ | 0.2119      | 0.2587      | 0.3034      | 0.3081      | 0.5502                  |
> |                  | FABind- $X_4$ | 0.2447      | 0.3076      | 0.3826      | 0.2813      | 0.6742                  |
> |                  | CentroBind     | **0.2582**  | **0.3392**  | **0.4224**  | **0.4649**  | **0.7006**              |
>
>
> Table 3 compares classification accuracy between FABind and CentroBind in both unimodal and multimodal settings, with additional visualization in Figure 2. The experiments are conducted using both randomly initialized and pre-trained backbones while varying the anchor modality. For each scenario, we train five decoders: four for individual modalities and one for the concatenated embeddings of all modalities.  $\times$ denotes the experiment without binding method. The results in Table 3 highlight that CentroBind significantly outperforms FABind with randomly initialized backbones, demonstrating its robustness to low-quality backbones.

---

> ### Author Response · Authors · 2024-11-26
> **Updated Table 4**
>
> We have compared several dynamic anchor generation methods in the following Table 4.
>
> ## Table 4
> | **Backbone**     | **Method**              |    [Fig. 8a |        and             |   8b]                                    | [Fig. 8c | and  | 8d]|
> |:------------------:|:-------------------------:|:-------------:|:-------------:|:--------------------------:|:-------------:|:-------------:|:--------------------------:|
> |                              |                                   | **$X_2$**            | **$X_4$** | **$X_1,\cdots,X_4$** | **$X_2$** | **$X_4$** | **$X_1,\cdots,X_4$** |
> | **Pre-trained**  | $\times$               | 0.115       | 0.296       | 0.566                    | 0.099       | 0.256       | 0.537                    |
> |                  | FABind- $X_4$         | 0.124       | 0.297       | **0.639**                | 0.115       | 0.263       | 0.540                    |
> |                  | CentroBind             | 0.131       | **0.363**   | 0.618                    | **0.116**   | 0.336       | 0.563                    |
> |                  | Weighted average       | 0.133       | 0.342       | 0.609                    | 0.102       | 0.338       | 0.574                    |
> |                  | Random + intra learning| 0.131       | 0.347       | 0.613                    | 0.105       | **0.353**   | 0.554                    |
> |                  | Random anchor          | **0.147**   | 0.359       | 0.619                    | 0.097       | 0.327       | **0.579**                |
> | **Random**       | $\times$               | 0.092       | 0.114       | 0.487                    | 0.067       | 0.176       | 0.465                    |
> |                  | FABind- $X_4$         | 0.131       | 0.143       | 0.575                    | 0.112       | 0.153       | 0.523                    |
> |                  | CentroBind             | 0.132       | **0.355**   | **0.626**                | **0.113**   | **0.336**   | **0.559**                |
> |                  | Weighted average       | 0.115       | 0.347       | 0.619                    | 0.109       | **0.336**   | 0.556                    |
> |                  | Random + intra learning| 0.132       | 0.333       | 0.602                    | 0.104       | 0.309       | 0.562                    |
> |                  | Random anchor          | **0.145**   | 0.354       | 0.618                    | 0.097       | 0.324       | 0.552                    |
>
>
>
> Table 4 presents the classification accuracy corresponding to the experiments illustrated in [Figure 8](https://anonymous.4open.science/api/repo/temp-C948/file/Comparison_dynamic_anchor.png?v=4844fb0a). In the experiments depicted in Figures 8a and 8b, the information level varies such that $X_1$, $X_2$, and $X_3$ are highly noisy, while $X_4$ is highly informative. Conversely, in Figures 8c and 8d, $X_1$ and $X_2$ are uninformative, whereas $X_3$ and $X_4$ are high-quality modalities. Each modality is selected as the fixed anchor modality in turn, and we compare CentroBind with FABind, weighted average, random anchor, and random anchor with intra-learning methods.
>
> When modality quality is imbalanced, the centroid-based approach may produce suboptimal dynamic anchor constructions, allowing other methods, such as weighted averages, to achieve better results in specific scenarios. Despite this, CentroBind consistently performs either better or on par with weighted average methods, showcasing its robustness to modality imbalance. Furthermore, the random anchor method also achieves competitive performance. These observations suggest that the choice of dynamic anchor generation method may have limited impact on final performance, as long as all encoders are appropriately trained during the process.

---

> ### Author Response · Authors · 2024-11-26
> **Updated Table 5**
>
> ## Table 5
> | **Method**       | **Modality**                          | **Sar-1**  | **Spk-1**  | **Spk-3**  | **Spk-5**  |
> |:------------------:|:----------------------------------:|:------------:|:------------:|:------------:|:------------:|
> | FABind           | $\mathcal{T}$                                | 0.606      | 0.219      | 0.458      | 0.632      |
> | UniBind          |                                       | 0.600      | 0.214      | 0.412      | 0.569      |
> | AudioCLIP*       |                                       | 0.488      | 0.155      | 0.280      | 0.388      |
> | ViT-Lens*        |                                       | 0.543      | 0.172      | 0.342      | 0.472      |
> | CentroBind       |                                       | **0.667**  | **0.287**  | **0.507**  | **0.642**  |
> |----------------|-------------------------------|------------|------------|------------|------------|
> | FABind           | $\mathcal{V}$                                | 0.668      | 0.375      | 0.587      | 0.691      |
> | UniBind          |                                       | 0.658      | 0.381      | 0.641      | 0.770      |
> | AudioCLIP*       |                                       | 0.504      | 0.110      | 0.275      | 0.414      |
> | ViT-Lens*        |                                       | **0.697**  | **0.586**  | **0.738**  | **0.797**  |
> | CentroBind       |                                       | 0.670      | 0.380      | 0.609      | 0.726      |
> |----------------|-------------------------------|------------|------------|------------|------------|
> | FABind           | $\mathcal{A}$                                | 0.639      | 0.201      | 0.457      | 0.599      |
> | UniBind          |                                       | 0.633      | 0.272      | 0.528      | 0.691      |
> | AudioCLIP*       |                                       | 0.525      | 0.158      | 0.343      | 0.454      |
> | ViT-Lens*        |                                       | **0.686**  | **0.396**  | **0.664**  | **0.8**    |
> | CentroBind       |                                       | 0.616      | 0.234      | 0.461      | 0.609      |
> |----------------|-------------------------------|------------|------------|------------|------------|
> | FactorCL*        | $\substack{\mathcal{V}, \mathcal{A}, \mathcal{T} \\ (\mathcal{V}, \mathcal{A}, \mathcal{T})}$ | 0.699      | -          | -          | -          |
> | SimMMDG*         |                                       | 0.725      | -          | -          | -          |
> | FABind           |                                       | 0.678      | 0.343      | 0.554      | 0.677      |
> | UniBind          |                                       | 0.646      | 0.383      | 0.622      | 0.764      |
> | AudioCLIP*       |                                       | 0.530      | 0.119      | 0.261      | 0.378      |
> | ViT-Lens*        |                                       | **0.731**  | **0.506**  | **0.736**  | **0.812**  |
> | CentroBind       |                                       | 0.704      | 0.346      | 0.594      | 0.733      |
>
> Models with asterisk are evaluated in different experimental settings (encoder backbone, pretraining datasets).
>
> Table 5 summarizes the classification accuracy of FABind, UniBind, and CentroBind for both single-modal and multimodal settings, using embeddings extracted by the binding methods and a simple decoder for classification. CentroBind generally outperforms other methods in sarcasm detection, while UniBind shows better performance in speaker classification, likely due to its use of LLM-augmented descriptions as additional modality. Notably, CentroBind consistently surpasses FABind. Additionally, CentroBind can incorporate LLM-augmented descriptions as an extra modality, which could further enhance its performance.
>
> For reference, the table also includes sarcasm detection results from FactorCL, SimMMDG, AudioCLIP, and ViT-Lens. While ViT-Lens achieves higher accuracy due to larger backbone encoders and extensive pretraining, it can be considered a variant of FABind. Applying CentroBind's dynamic anchor method to ViT-Lens or integrating Vision Transformers into CentroBind could further boost performance.

---

### Meta-Review · Area_Chair_YcKP · 2024-12-16

**Metareview:**

This is a borderline paper, but the overall idea of this paper could meet the high quality of an ICLR Paper. This paper experienced a long discussion phase in both reviewer-author and reviewer-AC. The reviewers have clear diverse opinions about this paper, and the rating scores are 5, 8, 3, 6, and 8. Here is a summary of the discussion:
The key characteristic of this paper is its simplicity, and we all know that a simple but effective algorithm is our ideal idea. However, the primary concerns can be summarized as follows.

***Motivation Issue***
This paper primarily investigates learnable anchors for multimodal learning, in contrast to the claimed use of fixed anchors. The main comparisons are drawn against fixed-anchor methods, particularly ImageBind [1]. The stated limitations of ImageBind and similar approaches are outlined as follows:
P1: over-reliance on a single anchor modality;
P2: failure to capture intra information;
P3: absence of shared information among non-anchored modalities;

However, in my view, these motivations require further careful examination.
For P1, ImageBind utilizes images as anchors to generate or associate with other modalities. This approach is primarily based on the rich and complex features embedded in the visual space, which facilitate the inference of other modalities.
For P2, ImageBind incorporates intra-modal information in its modality alignment process.
For P3, the shared information is also utilized in modality alignment, which is a core research challenge in multimodal learning, rather than being absent. Furthermore, ImageBind offers greater flexibility in multimodal learning by enabling seamless transitions from one modality to others, followed by their fusion.

[1] Girdhar, Imagebind: One embedding space to bind them all. In CVPR 2023.

***Technical Issue***
The proposed method is straightforward yet potentially effective for large-scale inference tasks. ImageBind aligns with the principle: "One begets Two; Two begets Three; Three begets all things."
In contrast, the authors adopt a multi-view contrastive learning approach based on instance-level representations, focusing on the mean features of each instance. However, a deeper investigation into multi-view or multimodal learning using contrastive learning [2] and its subsequent studies [3-6] reveals numerous related works that appear to have been overlooked by the authors.

Furthermore, the authors claim that intra-modal and shared information are jointly considered using the mean features of each instance. However, in my view, ImageBind is also capable of addressing this scenario for multimodal alignment and fusion within a joint space.

[2] Tian et al, Contrastive Multiview Coding, ECCV 2020.
[3] Xing et al. Adaptive cross-modal few-shot learning, NeurIPS 2019.
[4] Zeng, et al. Pan: Prototype-based adaptive network for robust cross-modal retrieval.SIGIR 2021.
[5] Huang, et al., Clover: Towards a unified video-language alignment and fusion model, CVPR 2023.
[6] Wang, et al. Vlmixer: Unpaired vision-language pre-training via cross-modal cutmix, ICML 2022.

Experiments Issue For ImageBind, this work is based on scaling laws, and sufficient data must be collected to validate its effectiveness across multiple tasks.

As another reviewer has pointed out, the authors conduct their experiments on a synthetic dataset and a claimed real-world dataset (MUStARD), which raises concerns about fairness. At present, we are unable to fully evaluate the performance comparisons between the proposed algorithm and the open-source ImageBind across the six modalities—images, text, audio, depth, thermal, and IMU data.

***Theory Issue***
Proposition 1 states FABIND with sufficient anchors. The sufficiency is attributed to large-scale data, an InfoNCE-based encoder, and a transformer-based architecture, as also stated in the main paper. The use of information is clearly articulated, which is no need to use a Proposition to clarify.

Proposition 2 is ‘FABIND with insufficient anchor’. Yes, it is known that information bottleneck theory is always used to define or prove the representation bound Hence, all representation models would exhibit this behavior, as it is a commonly used theory to assess the capabilities of deep learning models.

For Theorem 1, while I have not thoroughly reviewed all the details, it appears to be based on the information bottleneck and bound theory derivation.

In summary, while the overall idea of this paper is straightforward, it is not fully convincing from both experimental and theoretical perspectives. Therefore, this paper will be rejected.

**Additional Comments On Reviewer Discussion:**

In the discussion phase, two reviewers voted to accept this paper, and only one reviewer raised concerns about the experimental issue. We all know that experimental validation is one of the key measurements for a good work, but not the final one. However, if you carefully check all the main document and the appendix, the main idea is simple but not fully convincing, just with learnable anchors, i.e., the mean of the corresponding instance-level features with contrastive learning (InfoNCE). No more challenges appear for the reviewer-AC discussion, and the overall technical contributions of this paper are NOT fully consistent with the high standard of ICLR.

---

### Decision · Program_Chairs · 2025-01-22

Reject